   

# Asymmetric envelope surface disposition of secreted protein YjbI controls bimodal antibiotic susceptibilities in *C. crescentus*

Jordan Costafrolaz[1], Laurence Degeorges[1], Gaël Panis[1], Simon-Ulysse Vallet[1], Manuel Velasco Gomariz [2], Fernando Teixeira Pinto Meireles[3], Matteo Dal Peraro[3], Kathrin S Fröhlich [2,4] & Patrick H Viollier [1✉]

## Abstract

Cytoplasmic pentapeptide repeat proteins (PRPs) protect bacterial DNA gyrase from quinolone antibiotics. While some secreted PRPs are essential upon quinolone exposure, their role in the regulation of antibiotic resistance remains to be fully characterized. We show that a YjbI-type secreted PRP regulates antibiotic sensitivity, bimodally for small or large molecules, via modulation of the *Caulobacter crescentus* outer membrane (OM). YjbI silences two converging envelope-stress pathways that globally reprogram the OM proteome via TonB-dependent receptors (TBDRs), periplasmic proteases, and AcrAB-NodT, a multidrug efflux pump whose induction by small molecules and antibiotics is lethal to *yjbI* mutant cells. Loss of YjbI also confers sensitivity to vancomycin and bacitracin, two large peptidoglycan-targeting and zinc-binding antibiotics that permeate the outer membrane via the previously uncharacterized TBDR BugA and its orthologs. Zinc stress triggers rapid proteolytic removal of YjbI, activates expression of TBDRs, including BugA, and ultimately leads to replenishment of YjbI. Molecular dynamics simulations and reactive thiol probing imply an asymmetric surface disposition of YjbI, explaining the differential accessibility of its conserved cysteine pairs that flank the quadrilateral β-helix. Taken together, our findings identify a role of YjbI as a cell surface-regulator of outer membrane composition and antibiotic sensitivity in a Gram-negative bacterium.

**Keywords** YjbI; Antibiotic Sensitivity; TonB-dependent Receptor; Two-component System; Secreted Quadrilateral β-helix
**Subject Categories** Microbiology, Virology & Host Pathogen Interaction; Signal Transduction

## Introduction

The outer membrane (OM) is the defining envelope structure of Gram-negative (diderm) bacteria. It confers mechanical rigidity and chemical surface diversity, while serving as a sensory hub for environmental fluctuations (Saxena et al, 2023; Sun et al, 2022). As the first physical contact point of bacteria in the host, the OM lipopolysaccharide (LPS) components lipid A and O-antigen are key recognition sites of host innate and adaptive immunity, respectively (Simpson and Trent, 2019b). Importantly, the OM also confers protection toward antibiotics as an intrinsic multi-resistance determinant that reduces antibiotic permeation into diderm bacteria (Manrique et al, 2023; Theuretzbacher et al, 2023). Unlike the symmetric phospholipid bilayer of the cytoplasmic membrane (CM), the OM layer is asymmetric, featuring an inner leaflet of phospholipids and an outer leaflet with the negatively charged glycolipid lipopolysaccharide (LPS) counter-stabilized by certain divalent cations (Simpson and Trent, 2019b). The tight packing of LPS prevents the entry of large hydrophilic antibiotics such as vancomycin (VAN) that blocks biosynthesis of the essential cell wall (also known as peptidoglycan, PG), while its chemical properties impair the passage of many hydrophobic molecules and detergents.

This OM permeability barrier also challenges the assimilation of essential nutrients from the environment. Such molecules can pass the OM via hydrophilic pores formed by proteins that typically adopt a β-barrel architecture. While small nutrients pass through ungated pores (porins), larger ones typically depend on active uptake by gated channels such as TonB-dependent receptors (TBDRs)(Nikaido, 2003; Silale and van den Berg, 2023). The gating and import of TBDRs is energized by the proton motive force, relayed from the CM to the OM via the ExbBD-TonB transducer complex. TBDRs can import diverse large molecules such as siderophores or other metallophores (Moeck and Coulton, 1998; Pawelek et al, 2006; Schauer et al, 2007), vitamins (Shultis et al, 2006), lignin derivatives (Fujita et al, 2019) or carbohydrate polymers (Blanvillain et al, 2007; Neugebauer et al, 2005). Because

[1]Department of Microbiology and Molecular Medicine, Faculty of Medicine/Centre Médical Universitaire (CMU), University of Geneva, Geneva, Switzerland. [2]Friedrich Schiller University, Institute of Microbiology, Jena, Germany. [3]Institute of Bioengineering, School of Life Science, École Polytechnique Fédérale de Lausanne (EPFL), Lausanne, Switzerland. [4]Microverse Cluster, Friedrich Schiller University, Jena, Germany. ✉E-mail: patrick.viollier@unige.ch

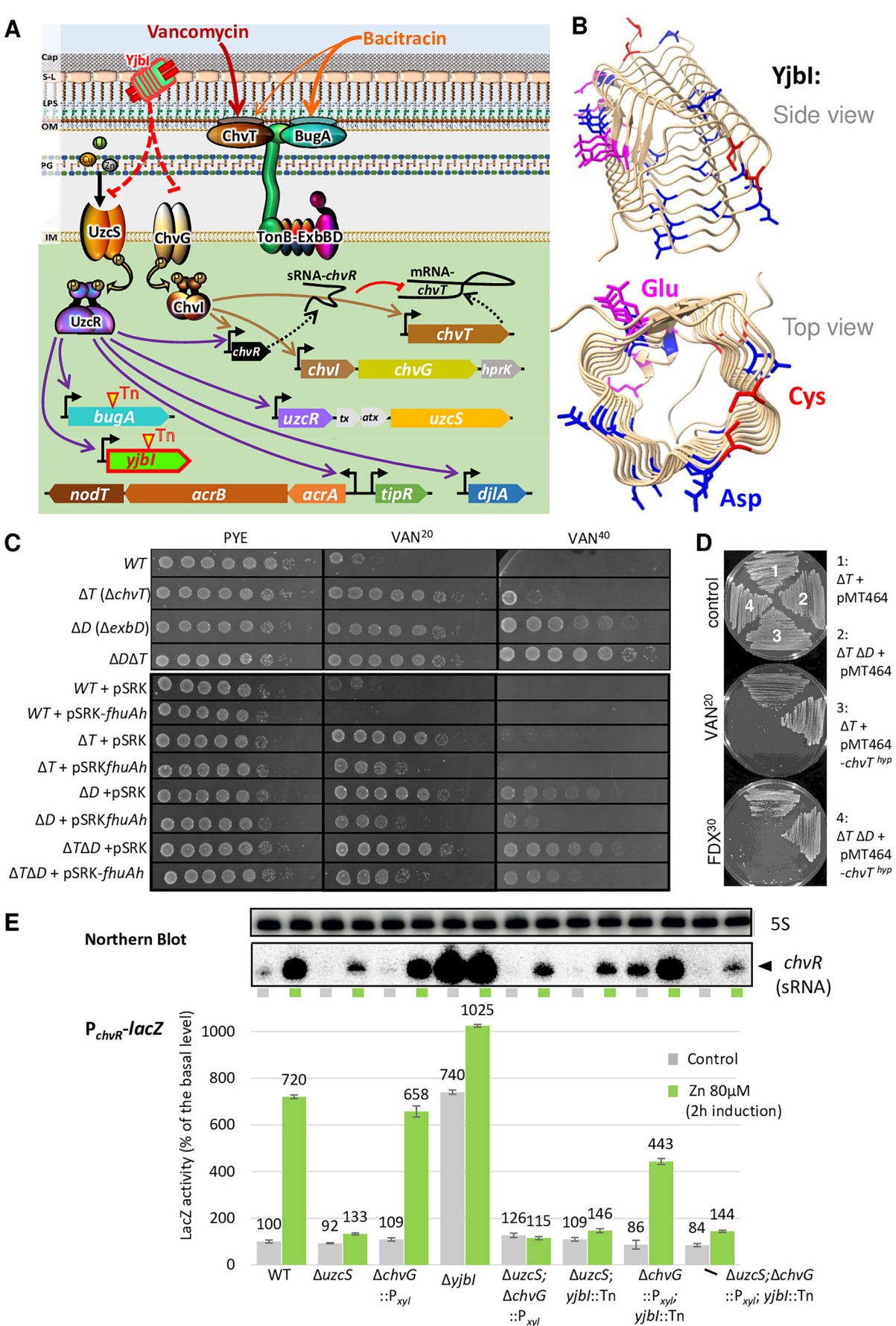

**Figure 1.  Model of YjbI-dependent signaling and identification of *yjbI*.**

(A) Proposed model of the TCSs-dependent regulation by YjbI across the *C. crescentus* envelope, including the capsule (cap), the S-layer (S-L), the outer membrane (OM) harboring lipopolysaccharide (LPS) in the outer leaflet, the cell wall (peptidoglycan, PG), and the inner or cytoplasmic membrane (IM). Inverted triangles (yellow) indicate transposon (Tn) mutants isolated in this work. Dashed lines indicate translation, and the red T-arrow depicts the negative translational regulation of *chvR* sRNA on the *chvT* mRNA. The dashed red T-arrow indicates the (presumed) indirect path by which YjbI inhibits UczRS and ChvGI signaling. The cartoon of YjbI indicates a quadrilateral β-helix (Jumper et al, 2021; Varadi et al, 2021) flanked by two cysteine pairs at each extremity. (B) Alpha-fold structure prediction of YjbI revealing a quadrilateral β-helix with charged acidic stretches containing glutamate (Glu, pink) and aspartate (Asp, blue) residues. Note the cysteine (Cys, red) pairs at the extremities. (C) Efficiency-of-plating (EOP) assay by tenfold serial dilutions (from left to right) of overnight cultures of *WT* and mutant cells on PYE plates with or without vancomycin 20 µg/mL (VAN[20]) or 40 µg/mL (VAN[40]) grown for 2 days at 30 °C. All plates containing strains harboring the pSRK-Gm or pSRK*fhuAh* plasmid contain IPTG (isopropyl β-D-1-thiogalactopyranoside) at a final concentration of 0.5 mM. (D) Sensitivity of *WT C. crescentus* containing pMT464 (conferring kanamycin resistance) or pMT464-*chvT*[hyp] streaked on PYE plates containing 10 µg/mL of kanamycin (KAN[10]) as a control, with or without VAN[20] or 30 µg/mL of fidaxomicin (FDX[30]). (E) β-galactosidase (LacZ) activity quantification from cells harboring the pP*chvR*-*lacZ* promoter probe plasmid. Inductions were performed in exponential phase cultures induced for 2 h with ZnSO$_4$ (80 µM) during growth at 30 °C in PYE (without xylose). LacZ activity is reported as percentage activity (reflecting 100% activity) of the *WT* (NA1000) in the uninduced condition. Error bars are defined as +/− standard deviation. The inset shows *chvR* mRNA levels as determined by Northern blotting using *chvR* as a probe against total RNA extractions from the same strains. The upper panel shows a blot of the same samples probed against the 5S rRNA as a loading control. The full blot with quantification is shown in Fig. EV3B. Source data are available online for this figure.

of their surface localization, they are also exploited as receptors for bacteriophages, anti-microbial peptides, or small toxins such as colicins (Braun et al, 1976; Salomon and Farias, 1993).

Binding of the substrate induces the opening (ungating) of the TBDR by dislodging its N-terminal plug domain of the TBDR that obstructs the large central pore (Braun, 2009; Hickman et al, 2017; Ratliff et al, 2022). When the plug domain is genetically deleted (along with surface-exposed loops), a large soluble channel remains that is constitutively open (also known as hyperpore). The diameter of this TBDR-derived hyperpore exceeds that of the ungated porins, permitting the unassisted transit of any large soluble antibiotic, for example VAN, across the OM (Killmann et al, 1993; Krishna-moorthy et al, 2016). The wide diameter of most TBDRs explains why large vitamins, nutrients, or bound metallophores use this import mechanism. However, TBDR channels can also represent an Achilles's heel of bacteria as entry portals for Trojan horse-style antibiotics, called sideromycins. Natural sideromycins are con-jugates with a molecular import signal that drives the internaliza-tion of a cytotoxic component across the OM, typically via a TBDR. In synthetic sideromycins, a siderophore-like uptake moiety is chemically joined to an antibiotic to facilitate TBDR-mediated internalization, a last resort antibiotic strategy used for treatment of infections with multidrug-resistant pathogens (Braun, 2009; Braun et al, 1976; Lin et al, 2019; Liu et al, 2018; Luna et al, 2020; Negash et al, 2019; Pugsley et al, 1987; Terra et al, 2021). Unfortunately, only few simple chemical moieties are known as suitable TBDR-directed uptake signals for antibiotic conjugates. Moreover, these signals are not universally recognized, presumably owing to the diversity of interactions required to engage TBDR types.

Some bacterial genomes encode a vast repertoire of TBDR paralogs. For example, the aquatic α-proteobacterium *Caulobacter crescentus* encodes 66 annotated TBDRs, but only two ExbD orthologs (Nierman et al, 2001), suggesting that the ExbBD-TonB complexes control multiple TBDRs. While the large TBDR repertoire presumably supports the oligotrophic lifestyle for the uptake of scarce nutrients in dilute environments, oligotrophic bacteria must also minimize the risk of promiscuous import of noxious molecules by TBDRs. Regulation of TBDR expression may reduce such unwanted internalization. Indeed, most of the *C. crescentus* TBDRs are not expressed in complex medium (PYE, Fig. EV1) (Siwach et al, 2021), but it is not well understood how the OM is reseeded with this plethora of TBDRs under different growth or stress states. While the ChvT TBDR (CCNA_03108) is abundant

in PYE and confers sensitivity to VAN (Vallet et al, 2020) (Fig. 1A), ChvT is negatively regulated by the small RNA ChvR at the post-transcriptional level (Frohlich et al, 2018; Vallet et al, 2020). Yet the sensory mechanism that induce downregulation of ChvT are poorly understood. The ChvR-ChvT regulatory interplay may respond to a physiological (for example, stress-induced) adaptation that results in reprogramming the OM proteomes with TBDRs to alter antibiotic susceptibility, potentially pervasively.

In the γ-proteobacterial model system *Escherichia coli*, sensory mechanisms mediated by OM proteins have been described (Cho et al, 2023a; Cho et al, 2023b; Mitchell and Silhavy, 2019; Simpson and Trent, 2019a), but orthologs of these sensors do not exist in *C. crescentus*. Moreover, as *E. coli* encodes few TBDRs, it is not an ideal model to explore switches in TBDR expression. By contrast, *C. crescentus* provides unique tractability and simple proxies to probe TBDR regulation and activity, for example, exploiting transcription of *chvR* as a genetic entry point to unravel how OM state changes are triggered. Prior evidence had revealed that the *chvR* promoter (P$_{chvR}$) is regulated by at least two distinct two-component signaling systems (TCSs): ChvGI (Bustamante et al, 2023; Green-wich et al, 2023; Stein et al, 2021) and UzcSR (Park et al, 2019; Park et al, 2017). Both TCSs are activated in response to different envelope stresses or challenges with toxic metals in *C. crescentus*, yet the molecular mechanism leading to activation remains obscure (Frohlich et al, 2018; Park et al, 2017; Quintero-Yanes et al, 2022; Vallet et al, 2020) (Fig. 1A).

Signaling by ChvGI involves phosphorylation of the ChvI DNA-binding response regulator by the ChvG histidine kinase, directing ChvI to target promoters like P$_{chvR}$ (Fig. 1A) (Aakre et al, 2013; Greenwich et al, 2023; Quintero-Yanes et al, 2022). P$_{chvR}$ is also a target of the DNA-binding response regulator UzcR that is phosphorylated by the UzcS histidine kinase (Fig. 1A) in response to excess zinc (Zn), copper (Cu), or uranium (U) (Park et al, 2017). Further studies of zinc stress revealed that UzcR also targets promoters of genes encoding OM proteins, TBDRs, several periplasmic metalloproteases, and a secreted pentapeptide repeat protein (sPRP) of the YjbI family [CCNA_01968, also known as PerA (García-Bayona et al, 2019)]. Importantly, UzcR also binds the P$_{acrA}$ promoter controlling expression of the *acrAB-nodT* operon, encoding a tripartite multidrug efflux pump of the RND (resistance-nodulation-division) superfamily. The assembled AcrAB-NodT machine spans all three envelope layers (Blair and Piddock, 2009; Kirkpatrick and Viollier, 2014; Nikaido and Pages,

2012; Siasat and Blair, 2023) and its assembly or activity is enhanced by the post-translational regulator DjlA (Costafrolaz et al, 2023), a DnaJ-like co-chaperone that is also expressed from a UzcR-target promoter ($P_{djlA}$).

$P_{acrA}$ and $P_{djlA}$ are also regulated by repression, and the addition of the quinolone antibiotic nalidixic acid (NAL) and other small molecules induces de-repression of both promoters (Costafrolaz et al, 2023). De-repression occurs by antagonistic action on the TipR repressor (Fig. 1A), by different mechanisms (Costafrolaz et al, 2023). Induction of $P_{acrA}$ (and therefore AcrAB-NodT) by NAL can be lethal in certain *C. crescentus* mutants with a fragile envelope (Delaby et al, 2021; Kirkpatrick and Viollier, 2014). The envelope-spanning nature of RND pumps may cause envelope stress upon massive and rapid upregulation, possibly through OM overcrowding. This adverse effect of AcrAB-NodT is less known since only the beneficial effects of efflux pump induction on cell survival through expulsion of antibiotics are typically reported (Friedman et al, 2001; Heeb et al, 2010; Jacoby, 2005).

Efflux by AcrAB-NodT is not the only source of quinolone resistance. Target modifications in the gyrase A (GyrA) subunit are known to confer a high level of resistance to many quinolones. Indeed, *C. crescentus* naturally harbors a GyrA variant that confers natural, high-level resistance to NAL (Hooper and Jacoby, 2016; Kirkpatrick et al, 2016; Kirkpatrick and Viollier, 2014). Interestingly, many multi-resistance plasmids carried by pathogens encode another type of quinolone resistance mechanism. In this case, protection is conferred by plasmid-encoded cytoplasmic PRPs that shield gyrase from the toxic action of quinolones (Hooper and Jacoby, 2016). Interestingly, PRPs can also be encoded on genomes, for example, members of a conserved class harboring an N-terminal secretion signal sequence (henceforth sPRPs), also known as YjbI orthologs (Pfam PF00805). Since these sPRPs/YjbIs are directed into the extracytoplasmic space, away from the compartment where DNA gyrase resides, their mechanism of action should differ from that of cytoplasmic PRPs. Here, we describe a genetic screen for negative regulators of $P_{chvR}$ in *C. crescentus* and follow-up studies that unveil YjbI as OM sentinel of stress responses (Fig. 1B). Removal of YjbI alters the OM proteome and, ultimately, intrinsic antibiotic susceptibility in at least two ways: reprogramming OM permeability via TBDRs and sensitivity to small molecules that induce de-repression of AcrAB-NodT. We find that silencing the UzcSR and ChvGI TCSs by YjbI occurs from an asymmetric surface disposition and that it constitutes a negative-autoregulatory loop to restore YjbI levels following stress-induced proteolysis by excess zinc.

# Results

## A TBDR system governs vancomycin (VAN) susceptibility

To unveil the determinants that render *C. crescentus* wild-type (WT) cells susceptible to VAN, we conducted transposon (Tn) insertion deep-sequencing (Tn-Seq) of VAN-resistant Tn mutants selected on complex (PYE) agar containing 20 μg/mL VAN (VAN[20], Fig. 1C; Dataset EV1). This Tn-Seq analysis uncovered three principal VAN-sensitivity determinants: *CCNA_00324* (encoding an ExbD family protein likely controlling TBDRs), CCNA_03052

(encoding a putative acetyltransferase), and CCNA_03108 encoding the TBDR ChvT (Fig. EV2A,B). As expected, the transcripts of these genes are abundant in cells grown in PYE, with *chvT* being the most highly expressed transcript among the three (Fig. EV1). The identification of *exbD* in the same selection regime suggested that *chvT* and *exbD* act in the same pathway, in which ExbD could potentially regulate ChvT and/or possibly other TBDRs. To explore this idea, we constructed strains with in-frame deletions in *chvT* and/or *exbD* (Δ*chvT*, Δ*exbD*, and Δ*chvT* Δ*exbD*) in *WT* cells to analyze the contribution of each deletion to VAN resistance. As readout in these experiments, we used efficiency of plating (EOP) on PYE containing VAN at 20 or 40 μg/mL (VAN[20] or VAN[40], Fig. 1C). Loss of ExbD or ChvT enhances colony formation on VAN[20], with an EOP differential of four orders of magnitude. Additionally, the EOP of Δ*exbD* Δ*chvT* double mutant cells is slightly increased compared to Δ*exbD* single mutant cells on VAN[40].

If ChvT/ExbD promote internalization of VAN, then the ChvT pore should be sufficiently wide to enable passage of VAN in the ungated state, even in the absence of ExbD. The ungated state of TBDRs has been mimicked by deletion of the N-terminal plug and surface-exposed loops of the *E. coli* TBDR FhuA, resulting in the well-characterized hyperpore derivative (FhuA[hyp], Fig. 1C). Indeed, expression of a comparable ChvT[hyp] derivative (lacking surface-exposed loops and the N-terminal plug) in Δ*exbD* or Δ*chvT* cells reduced the EOP on VAN[20]. Similarly, expression of FhuA[hyp] (Mohammad et al, 2011) renders *WT*, Δ*chvT* and Δ*exbD* single and double mutant cells more sensitive to VAN[20] and VAN[40] suggesting that VAN can now traverse the OM via FhuA[hyp] (Fig. 1C). Akin to FhuA[hyp], ChvT[hyp] renders cells susceptible to other large antibiotics that are typically excluded by an impermeable OM, for example fidaxomicin, a natural macrocycle antibiotic acting on RNA polymerase in the cytoplasm (Jung et al, 2023), at concentration of 30 μg/mL (FDX[30], Fig. 1D). Importantly, ChvT[hyp] expression even renders *E. coli* K12 cells sensitive to VAN and FDX (Fig. EV2C,D), indicating that the effect of ChvT[hyp] in permitting the passage of large antibiotics is species independent. Collectively, we conclude that the internalization of VAN in *C. crescentus* requires ChvT, ExbD, and possibly other TBDRs controlled by ExbD (on VAN[40]).

## YjbI negatively regulates transcription of the sRNA *chvR*

When ChvT is abundant, the levels of its translational inhibitor, the sRNA ChvR, are inversely proportional (Frohlich et al, 2018), due to poor activity of the *chvR* promoter ($P_{chvR}$) (Fig. 1E). Indeed, a transcriptional reporter in which $P_{chvR}$ drives expression of the NptII (*nptII*) neomycin phosphotransferase that confers resistance to kanamycin ($P_{chvR}$-*nptII*) is not active in *WT* cells (Fig. EV3A). To uncouple this negative regulation from $P_{chvR}$-*nptII*, we selected for kanamycin-resistant mutants following mutagenesis with a *himar1* Tn (conferring gentamycin resistance). We mapped the Tn insertion in two mutants with elevated $P_{chvR}$-*nptII* activity and found each mutant to harbor a Tn insertion in the poorly characterized *yjbI* gene (Fig. EV3A). The predicted YjbI gene product harbors 183 residues with abundant pentapeptide repeats and an N-terminal secretion signal. Alphafold analysis suggests that YjbI organizes into a quadrilateral β-helix (Buchko et al, 2006), featuring at least 3 patches of charged side chains lining the sides of the β-helix. Moreover, the organization of YjbI and orthologs is

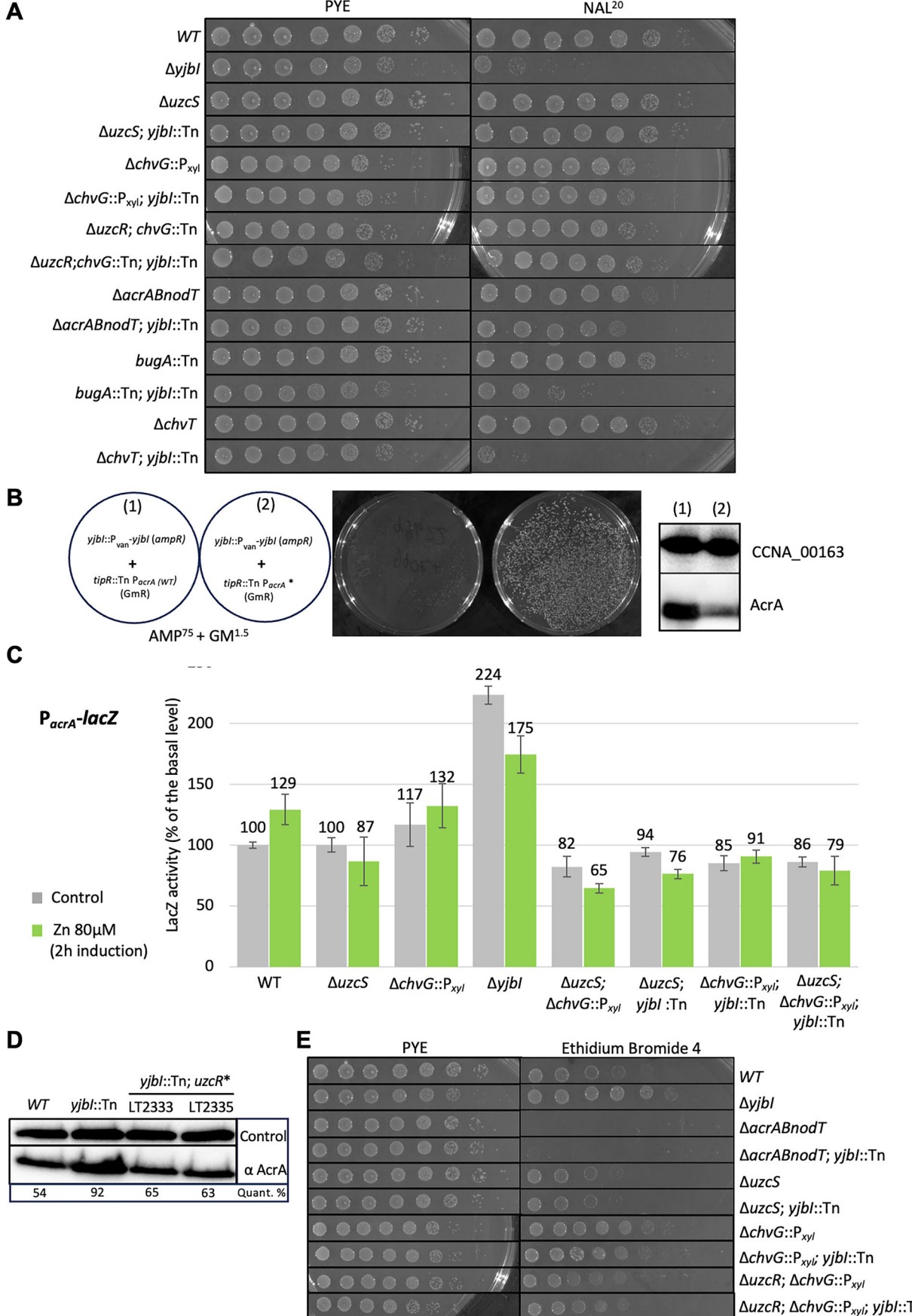

**Figure 2.  Genetic interplay of *yjbI*, *acrAB-nodT*, and *uzcR*.**

(A) Efficiency-of-plating (EOP) assay by tenfold serial dilutions (from left to right) of *WT* and mutant strains on PYE plates with or without 20 μg/mL of nalidixic acid (NAL[20]). Plates were incubated for 2 days at 30 °C. (B) Generalized transduction experiment of *tipR*::Tn (1) and *tipR*::Tn P$_{acrA}$* (2) into *yjbI*::P$_{van}$-*yjbI* cells selected on PYE plates containing 75 μg/mL of ampicillin (AMP[75], for maintenance of P$_{van}$-*yjbI*) and 1.5 μg/mL of gentamycin (GM[1.5]) for *tipR*::Tn and *tipR*::Tn P$_{acrA}$*. Plates were photographed after 3 days at 30 °C. The immunoblot on the right shows the AcrA steady-state levels in *tipR*::Tn (1) and *tipR*::Tn P$_{acrA}$* (2) cells as determined using polyclonal antibodies to AcrA. The blot was also probed with polyclonal antibodies to CCNA_00163 as a control for loading. (C) β-galactosidase (LacZ) activity measurements of *WT* and mutant cells that harbor the pP$_{acrA}$-*lacZ* promoter plasmid and were grown to exponential phase before ZnSO$_4$ (80 μM) induction (or not) at 30 °C in PYE (without xylose). All LacZ activity measurements are indicated as percentage of expression regarding the basal level of the uninduced *WT* (100%). Error bars are defined as +/− standard deviation. (D) Immunoblot analysis using antibodies to AcrA to probe extracts of exponential phase NA1000 (*WT*) and *yjbI* mutant cells. Extracts were separated by SDS-PAGE. Quantifications show the ratio of AcrA to loading control protein for each lane expressed as a percentage. (E) EOP assay to assess the efflux capacity by AcrAB-NodT in strains grown on PYE on plates with or without 4 μg/mL ethidium bromide (EtBr[4]). Plates were incubated for 48 h at 30 °C. The comparison is to growth on PYE without treatment as in (A). Source data are available online for this figure.

such that pairs of cysteines lie at each extremity of the β-helix (Fig. 1B). YjbI is localized to the OM (Cao et al, 2012; García-Bayona et al, 2019) (and see below), but it is unknown why point mutations in *yjbI* and *chvT* attenuate killing by the contact-dependent bacteriocin CdzC/D of *C. crescentus* (García-Bayona et al, 2019).

Next, we sought to substantiate the genetic regulatory link between *yjbI* and *chvT* (via ChvR). To this end, we conducted P$_{chvR}$ activity measurements in *WT* and Δ*yjbI* (with an in-frame deletion in *yjbI*) cells harboring the LacZ-based promoter probe plasmid pP$_{chvR}$-*lacZ* (Fig. 1E). Indeed, inactivation of *yjbI* resulted in a sevenfold increase in P$_{chvR}$ activity (740% relative to 100% *WT* activity, Fig. 1E). As P$_{chvR}$ is directly regulated by the ChvI response regulator that is phosphorylated by the ChvG kinase (Frohlich et al, 2018; Quintero-Yanes et al, 2022), we confirmed that ChvG or ChvI are required for P$_{chvR}$ induction upon inactivation of YjbI as evidenced by the finding that a *yjbI* mutation did not alter P$_{chvR}$-*lacZ* activity in Δ*chvI* or Δ*chvG*::P$_{xyl}$ cells (note that the Δ*chvG*::P$_{xyl}$ insertional mutation was designed to inactivate ChvG and maintain expression of the distal genes *hprK* in the operon using the leaky P$_{xyl}$, Figs. 1E and EV4D).

Two additional experiments indicate that loss of YjbI leads to hyperactivation of P$_{chvR}$ via ChvGI. First, Northern blotting confirmed that upregulation of the ChvR sRNA caused by loss of YjbI is ChvG dependent (Figs. 1E and EV3B), and chromatin-immunoprecipitation (ChIP) experiments described below revealed an increase of HA-tagged ChvI(-HA) at its target promoters in *yjbI* mutant cells versus *WT* cells.

## YjbI protects from lethal induction of the *acrAB-nodT* operon by NAL

Unexpectedly, our genetic analysis of *yjbI* mutants contrasts with the annotation of *yjbI* as an essential gene in *C. crescentus* based on high-density Tn-Seq analysis (Christen et al, 2011). However, in these Tn-Seq experiments, PYE plates containing NAL were used for the selection of viable Tn mutants, unlike our generation of *yjbI* mutants. Historically, NAL had been used as a selection for exconjugants after Tn mutagenesis introduced by intergeneric conjugation from NAL-sensitive *E. coli* (Christen et al, 2011). By contrast, *C. crescentus* is naturally resistant to NAL by way of a natural polymorphism (encoding F96D) in the only GyrA-encoding gene. While this mutation confers high-level resistance to NAL, the NAL-based transcriptional induction of AcrAB-NodT along with the co-chaperone DjlA protects cells from lower concentrations of

quinolones and to certain β-lactam antibiotics (Costafrolaz et al, 2023; Kirkpatrick and Viollier, 2014). In past work, we had observed that certain fragile envelope mutants can be NAL-sensitive, likely due to induction of the *acrAB-nodT* operon by NAL. For example, cells lacking the polarity factor TipN or the RNA polymerase-associated factor TrcR (Delaby et al, 2021; Kirkpatrick and Viollier, 2014) are sensitive to NAL. Consistent with their NAL-sensitivity, *tipN* and *trcR* are annotated as "essential" genes in the NAL-based Tn-Seq (Christen et al, 2011), but *tipN* or *trcR* in-frame deletion mutants are viable on PYE. It stands to reason that *yjbI* mutants are likely also NAL-sensitive, yet viable.

To probe for such NAL-sensitivity, we conducted EOP assays of *WT* and Δ*yjbI* cells on PYE plates with or without NAL (20 μg/mL, NAL[20]). We observed five orders of magnitude reduction in EOP of YjbI mutant cells in the presence of NAL (Fig. 2A). By contrast, no reduction in EOP was visible for *WT* cells on PYE with or without NAL[20] (Fig. 2A). In complementary experiments, we had conducted a negative selection screen for NAL[20]-sensitive (NAS) Tn mutants replica-plated from a library of mutants arrayed in individual wells of 96-well plates (Huitema et al, 2006) onto PYE plates with or without NAL[20]. This screen uncovered one NAS mutant with a Tn insertion in *yjbI*, another with a Tn in *trcR*, several mutants with a Tn insertion in the large TipN-encoding gene, and other, currently uncharacterized, mutants. The *yjbI*::Tn NAS mutant (NAS4, henceforth *yjbI*::Tn[NAS4]) harbors a Hyper-Mu Tn encoding kanamycin resistance within *yjbI*. We backcrossed *yjbI*::Tn[NAS4] into *WT* cells and found that the resulting mutants were indeed sensitive to NAL[20] as determined by EOP (Fig. EV3C). Moreover, the *yjbI*::Tn[NAS4] mutation also causes over-activation P$_{chvR}$-*lacZ* (Fig. EV4) and the NAL-sensitivity of Δ*yjbI* and *yjbI*::Tn[NAS4] cells is corrected upon expression of *WT* YjbI in trans from the *xylX* locus (Fig. EV3C) or from the replicative plasmid pMT335 (Thanbichler et al, 2007) (see Fig. EV5B).

Owing to the massive transcriptional induction of AcrAB-NodT by NAL (Costafrolaz et al, 2023; Kirkpatrick and Viollier, 2014), we speculated that AcrAB-NodT overexpression could be toxic to *yjbI* mutant cells, possibly by destabilizing an already fragile envelope. Indeed, deletion of the *acrAB-nodT* operon attenuated the NAL[20] sensitivity of *yjbI* mutant cells, as shown by a fourfold higher EOP in *yjbI*::Tn[NAS4]; Δ*acrAB-nodT* cells that lack AcrAB-NodT compared to *yjbI*::Tn[NAS4] cells (Fig. 2A). Since induction of AcrAB-NodT can also be achieved by inactivation of the *cis*-encoded repressor of the *acrAB-nodT* operon, TipR (Costafrolaz et al, 2023; Kirkpatrick and Viollier, 2014), *yjbI* mutant cells should be

synthetically lethal with a *tipR*::Tn mutation. The *tipR*::Tn (encoding gentamycin resistance) mutation can be transduced into cells expressing YjbI from the vanillate-inducible $P_{van}$ promoter (*yjbI*::P$_{van}$-*yjbI*), however the transduction is inefficient in the absence of vanillate, with few small colonies appearing after 3 days of incubation, and efficient propagation of these transductants requires vanillate (Figs. 2B and EV3D). However, when a *tipR*::Tn allele harboring a linked promoter-down mutation in P$_{acrA}$ (*tipR*::Tn P$_{acrA}$*) is transduced into *yjbI*::P$_{van}$-*yjbI* cells, then colonies appear after 3 days of incubation at 30 °C on plates lacking vanillate (Fig. 2B). Taken together, our findings indicate that de-repression of the *acrAB-nodT* operon is toxic to cells lacking YjbI.

## UzcSR mutations mitigate the NAL-sensitivity of Δ*yjbI* cells

The fact that the EOP of *yjbI* mutant cells lacking ArcAB-NodT is still inferior to that of *WT* cells on NAL[20] plates (Fig. 2A) indicates that determinants other than AcrAB-NodT can adversely affect *yjbI* mutant cells. To determine the genetic basis for this effect, we isolated three spontaneous *yjbI*::Tn suppressor mutants (LT2333, LT2335, and LT2336) growing on NAL[20] plates and sequenced their genomes. Each strain had a different mutation in *uzcR*, encoding the response regulator UzcR (Fig. 1A) (Park et al, 2017). While the nonsense mutation in *uzcR* of strain LT2333 is due to a CA dinucleotide insertion at codon 46, inducing a frameshift that terminates 36 codons later, mutant LT2336 has a C nucleotide insertion at codon 81 in *uzcR* that induces termination four codons later. By contrast, *uzcR* in mutant LT2335 harbors a missense mutation (C→A) at codon 188, encoding UzcR(T188N). To confirm the recessive nature of these mutations, we introduced the *yjbI*::Tn mutation into Δ*uzcR* cells and found a matching increase in EOP on NAL[20] that was reversed upon expression of a synthetic UzcR variant with a double HA-tag at the N-terminus (2xHA-UzcR, Fig. EV3C). To confirm that loss of UzcSR signaling abrogates the NAL[20] sensitivity of *yjbI* mutants, we transduced *yjbI*::Tn[NAS4] into Δ*uzcS* cells (lacking the UzcS histidine kinase) and found that the EOP of the resulting double mutant on NAL[20] is comparable to that of *WT* cells (Fig. 2A).

Previous ChIP-Seq experiments had shown that UzcR binds the *acrAB-nodT* promoter, P$_{acrA}$ [(Park et al, 2017) and below]. We therefore compared P$_{acrA}$ activities in *yjbI*::Tn single mutant and the three *yjbI*::Tn *uzcR** double mutant strains (LT2333, LT23335, and LT2336, Fig. EV4A) using the pP$_{acrA}$-*lacZ* promoter probe plasmid. We discovered a twofold increase in P$_{acrA}$-*lacZ* activity in *yjbI*::Tn cells that is lost in *yjbI*::Tn *uzcR** double mutant cells (Fig. EV4A), in *yjbI*::Tn Δ*uzcS* double mutant cells (Fig. 2C) or when Δ*yjbI* cells are complemented with the pMT335 plasmid expressing YjbI (Fig. EV4B). Moreover, immunoblotting using polyclonal antibodies to AcrA revealed an increase in the steady-state levels of AcrA in *yjbI*::Tn cells that is lost in *yjbI*::Tn *uzcR** double mutant cells (Fig. 2D). Because of the interplay between the UzcSR and ChvGI signaling systems on *chvR* as described below, we also tested the impact of the Δ*chvG* mutation on P$_{acrA}$-*lacZ* activity in *yjbI* mutant cells and found that it attenuates promoter firing to 91% *WT* activity (Fig. 2C), whereas the single Δ*yjbl* exhibits 175% activity relative to *WT*. Moreover, the Δ*chvG* mutation also improves the EOP of *yjbI* mutants on NAL[20]

(Fig. 2A), albeit to a lesser extent than the Δ*uzcS* mutation. While this explains why NAL[20]-resistant *yjbI*::Tn mutants preferentially inactivate UzcSR signaling, it also indicates that ChvGI can affect AcrAB-NodT expression (and activity, see below) in *yjbI* mutant cells.

To determine whether loss of YjbI also confers an increase in AcrAB-NodT activity, we assesed AcrAB-NodT-mediated efflux of the toxic DNA intercalant ethidium bromide (EtBr) by EOP assays on plates containing 4 μg/mL EtBr (EtBr[4], Fig. 2E). Whereas Δ*acrAB-nodT* cells cannot grow on EtBr[4] plates, *yjbI*::Tn cells show an increase in EOP of approximately three orders of magnitude relative to *WT* cells, an effect that is dependent on AcrAB-NodT (Fig. 2E). By contrast, the EOP of Δ*uzcS* single mutant or *yjbI*::Tn[NAS4] Δ*uzcS* double mutant cells is slightly reduced compared to that of *WT* cells, Δ*chvG* single mutant cells or *yjbI*::Tn[NAS4] Δ*chvG* double mutant cells. Thus, the efflux activity of EtBr by AcrAB-NodT in the different strains correlates well with the changes in P$_{acrA}$ firing.

## Stress-induced proteolysis of YjbI activates two intertwined TCSs

As previous ChIP experiments had indicated that Zn stress results in the recruitment of UzcR to its target promoters [including P$_{acrA}$ and P$_{chvR}$, (Park et al, 2017)], we set out to measure the induction of P$_{acrA}$ and P$_{chvR}$ in response to (Zn) stress or to loss of YjbI. We confirmed that P$_{chvR}$-*lacZ* and P$_{acrA}$-*lacZ* are both rapidly induced by Zn in a UzcS-dependent manner (Figs. 1E and 2C). Since P$_{acrA}$-*lacZ* and P$_{djlA}$-*lacZ* are both repressed by TipR (Costafrolaz et al, 2023) and activated by UzcR (Park et al, 2017), we probed for synergistic induction due to de-repression by NAL and activation by Zn and found this to be the case (Fig. EV4C). By contrast, YjbI overexpression from the vanillate-inducible $P_{van}$ promoter on pMT335-*yjbI* dampens the induction of P$_{acrA}$-*lacZ* by Zn (Fig. EV4B).

We noted that the Δ*chvG* mutation did not completely abolish the induction of P$_{chvR}$-*lacZ* in response to Zn in *WT* or *yjbI*::Tn cells compared to the Δ*uczS* mutation (Fig. 1E), suggesting that Zn induction of P$_{chvR}$ can occur independently of ChvG. However, measuring P$_{chvR}$-*lacZ* activity in Δ*chvI* and Δ*uzcR* single mutant cells and in Δ*chvI; ΔuzcR* double mutant cells, we found that each of the response regulators is required to induce P$_{chvR}$ by Zn or by the *yjbI*::Tn mutation (Fig. EV4D). Northern blotting (Figs. 1E, inset and EV3B) confirmed the induction of the *chvR* sRNA in response to Zn or to the *yjbI*::Tn mutation. Finally, we conducted ChIP-Seq experiments with N-terminally HA (hemagglutinin)-tagged versions of UzcR (HA-UzcR) or ChvI (HA-ChvI) expressed from the *xylX* locus of *yjbI* mutant or *WT* cells (Fig. 3A; Dataset EV2) and detected increased binding of HA-UzcR and HA-ChvI at 134 and 130 target promoters in *yjbI* mutant *versus WT* cells (Fig. 3A; Dataset EV2) (Park et al, 2017; Quintero-Yanes et al, 2022). Our analyses revealed an overlap of 30 YjbI-dependent promoters targeted by UzcR and ChvI, targeting distinct motifs (Fig. 3B; Dataset EV2) within the promoters of, for example, *chvR*, the TBDR-encoding genes (*chvT*, *CCNA_00974*, *CCNA_01051*, and *CCNA_03096*), and the *exbBD-tonB1* operon (*CCNA_02421-CCNA_02419*). Interestingly, the promoter of *yjbI* is itself a target of UzcR, suggesting negative autoregulation in the circuit mitigated in response to Zn stress or loss of YjbI.

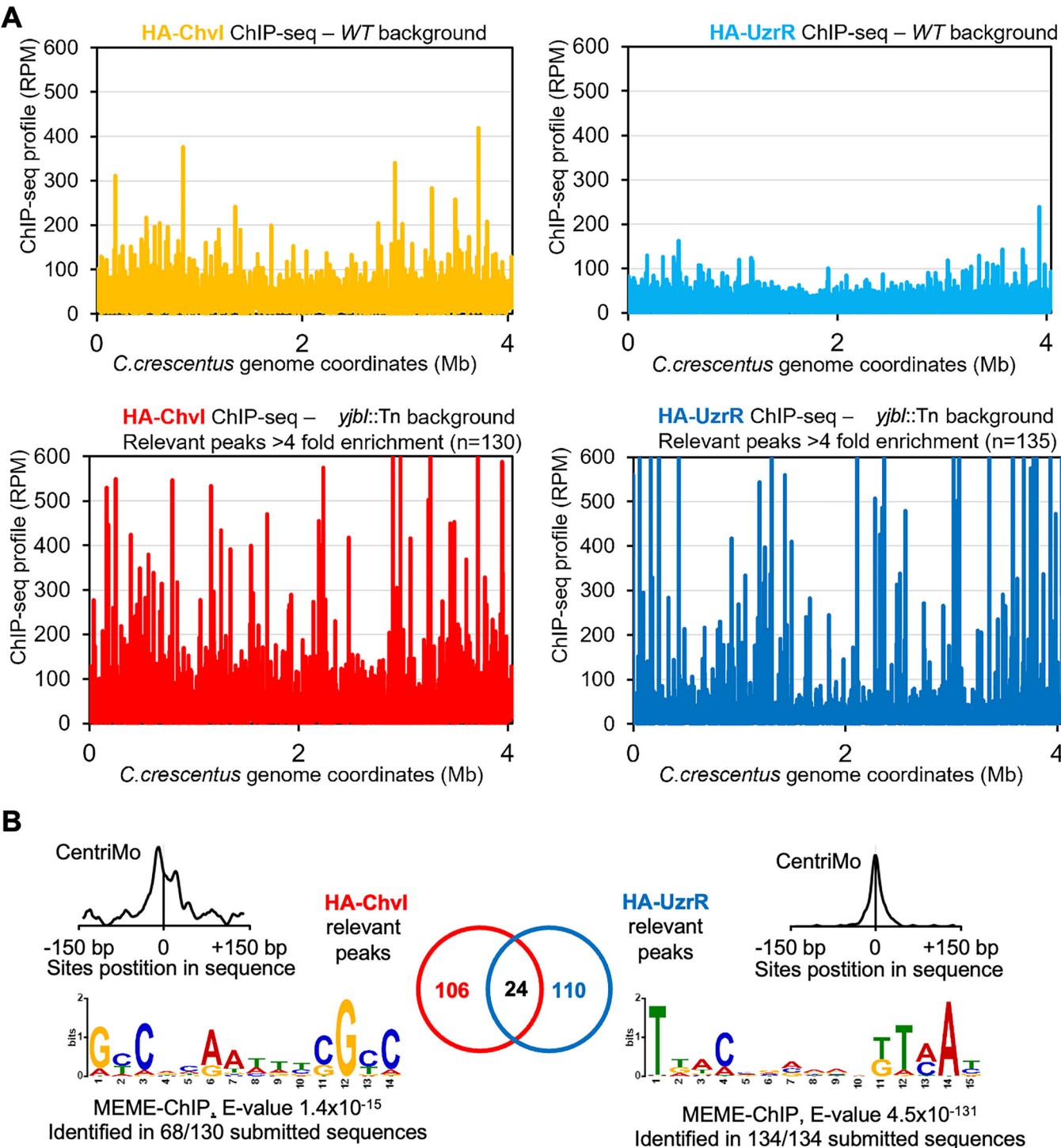

**Figure 3.   YjbI-dependent promoter occupancy of ChvI and UzcR.**

(A) Graphical representation of HA-UzcR or HA-ChvI (expressed from the *xylX* locus) occupancy on the chromosome of *WT* and *yjbI* mutant cells grown in PYE with xylose. The *X* axis is a linear representation of the genome starting at the *ori*, and the chart shows a fourfold increase in sequence representation compared to the total input as determined by ChIP-Seq analyses. The *Y* axis represents the abundance of sequence reads expressed as reads per million (RPM). (B) Venn diagram of the HA-UzcR and HA-ChvI regulons and the overlapping controlled genes. The MEME-derived sequence logos show the consensus sequence of binding of HA-UzcR and HA-ChvI obtained from, respectively, 130 and 134 binding sites detected by ChIP-Seq analysis. The CentriMo plots show that the binding sites overlap with the summit of each ChIP-Seq peak, centered between 150 bp upstream or downstream of the peak. Source data are available online for this figure.

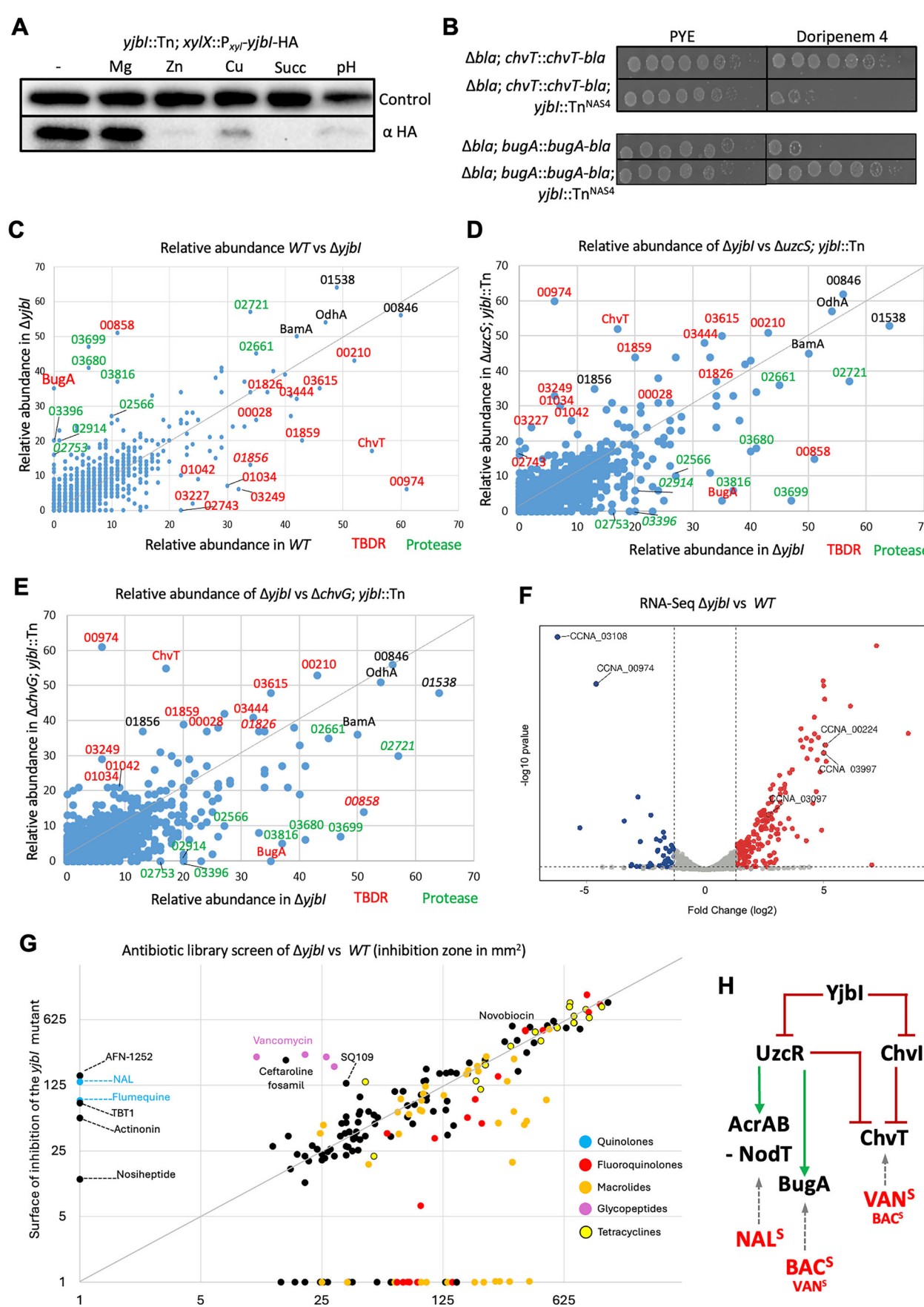

Figure 4. YjbI-dependent TBDR remodeling of the OM proteome.

(A) Immunoblot analysis using anti-HA antibodies on extracts from *yjbI*::Tn; *xylX*::P$_{xyl}$-*yjbI*-HA strain cells grown with xylose 0.03% for 4 h, with and without induction by MgSO$_4$, ZnSO$_4$, CuSO$_4$ (all at 80 μM), sucrose 6% or at pH 5.5. for 2 h. Extracts were separated by SDS-PAGE. The blot was probed with antibodies to CCNA_00163 as a control for loading. (B) EOP assay of strains expressing a ChvT or BugA variant fused to the Bla β-lactamase (ChvT-Bla or BugA-Bla) from the *chvT* or *bugA* promoter at the endogenous locus. EOP was assessed on PYE plates with or without 4 μg/mL of doripenem. Plates were incubated for 2 days at 30 °C. (C–E) Scatter plot representation of the relative abundance of outer membrane enriched protein fractions from *WT versus* Δ*yjbI*, or Δ*yjbI versus* Δ*uzcS*; *yjbI*::Tn$^{NAS4}$, or Δ*yjbI versus* Δ*chvG*::P$_{xyl}$; *yjbI*::Tn$^{Gm}$ cells as determined by LC-MS/MS. The TBDRs are colored in red, and the predicted proteases are colored in green. (F) Volcano plot representation of the RNA-Seq analysis of total RNA extracted from *WT* and Δ*yjbI* cells. Selected TBDR-encoding transcripts are indicated, including the downregulation (blue) of *chvT* (*CCNA_03108*) and *CCNA_00974* in Δ*yjbI versus WT* cells, and the corresponding upregulation (red) of *bugA* (*CCNA_00224*), *CCNA_03997*, and *CCNA_03097*. (G) Volcano plot representation of the growth inhibition area (in mm²) of cells achieved by Kirby-Bauer-type antibiotic diffusion assays upon spotting 3 μL of 1 mM solutions of various antibiotics onto *WT* or *yjbI* mutant indicator lawns embedded in soft agar layered on top of PYE plates and then grown for 24 h at 30 °C. The antibiotics were from two MedChemExpress libraries (HY-L033 and HY-L067). Data were from on two experiments with independent *WT* and two *yjbL* mutant strains. (right) The chart on the right illustrates the bimodal antibiotic sensitivity mechanisms induced by inactivation of Yjbl, including the sensitivity to NAL conferred by induction of the *acrAB-nodT* operon, but also the sensitivity to BAC and VAN conferred by the strong upregulation of BugA, despite the concomitant downregulation of ChvT by ChvR. (H) Model summarizing the bimodal antibiotic sensitivity of *yjbI* mutants, towards small anibitiotics such as NAL (NAL$^S$) that interfere with viability of *yjbI* cells via induction of *acrAB-nodT* operon, versus the sensitivity towards large antibiotics such as BAC or VAN that are internalized by induction of the TBDR BugA. Both sensitivities are due to the induction of UzcSR (and ChvGI) signaling in *yjbI* mutant cells (see Fig. 5). Source data are available online for this figure.

Our findings suggest that the rapid proteolysis of YjbI underpins the Zn stress response. Upon expression of a functional YjbI variant with a C-terminal HA tag (YjbI-HA) from the *xylX* locus of *yjbI* ::Tn cells (that restores NAL[20] resistance to *yjbI*::Tn cells, Fig. EV3C), we noted that exposure to 80 μM ZnSO$_4$ for 2 h led to a rapid loss of YjbI-HA as detected by immunoblotting using monoclonal antibodies to HA (Fig. 4A). While YjbI-HA was no longer detectable after ZnSO$_4$ exposure and barely detectable when CuSO$_4$ was used (80 μM), there was no difference in abundance upon the addition of MgSO$_4$ (80 μM). Zn and sucrose stress have been reported as triggers of UzcSR and ChvGI, respectively (Park et al, 2017; Quintero-Yanes et al, 2022), and we indeed observed the loss of YjbI-HA upon exposure to sucrose stress. Because acid stress (pH 5.5) has been reported to induce ChvR via ChvGI (Frohlich et al, 2018; Vallet et al, 2020), we tested whether low pH also affects YjbI-HA abundance and failed to detect the protein in response to low pH (Fig. 4A).

Taken together, our results show that YjbI is rapidly lost from cells upon stress challenges that trigger UzcSR and ChvGI signaling, profoundly inducing target promoters including the *yjbI* promoter in a homeostatic loop to replenish YjbI and thereby terminate the response once the stress subsides.

## YjbI controls the OM proteome and antibiotic uptake via TBDRs

Since our ChIP-Seq results revealed that many OM proteins and TBDRs are transcriptionally induced in *yjbI* mutant cells, the OM proteome should differ substantially between *yjbI* mutant and *WT* cells. Indeed, proteomic Liquid Chromatography-Mass Spectrometry (LC-MS/MS) analyses of OM protein-enriched samples obtained by differential solubilization from *WT* and Δ*yjbI* cells (Fig. 4C) revealed an abundance of highly expressed TBDRs in *WT* cells as determined by RNA-Seq (Fig. EV1). Several of these TBDRs are strongly downregulated in *yjbI* mutant cells (Fig. 4C), for example, only 2.3% of CCNA_00974 and 1.4% of ChvT (CCNA_03108) remain in Δ*yjbI* cells compared to *WT* cells. By contrast, two TBDRs were highly abundant in Δ*yjbI* cells compared to *WT* cells, with a 700-fold increase in the TBDR CCNA_00224 (see below) and a tenfold increase of the TBDR CCNA_00858, both

of which are expressed from an UzcR-target promoter (Park et al, 2017) (Fig. 3A; Dataset EV2).

Many putative extracytoplasmic proteases/peptidases are also more abundant in the OM-enriched fraction from Δ*yjbI* cells compared to *WT* cells, including CCNA_03396, CCNA_03699, CCNA_02914, and CCNA_02933. Again, most of the upregulated proteins are expressed from UzcR- or ChvI-target promoters, and this imbalance in abundance is largely mitigated when UzcS or ChvG is inactivated, as revealed by LC-MS/MS analyses of Δ*uzcS yjbI*::Tn and Δ*chvG yjbI*::Tn double mutant cells (Fig. 4D,E; Dataset EV3). Finally, RNA-seq analysis of total RNA extracted from *WT* and *yjbI* cells provided matching results to the proteomics dataset (Fig. 4F; Dataset EV4). For example, several transcripts encoding TBDRs such as CCNA_00224, CCNA_03097, and CCNA_03997 are highly upregulated compared in *yjbI* mutant cells *versus WT* cells, whereas for the inverse pattern was seen for other transcripts such as the ones encoding ChvT (CCNA_03108) or CCNA_00974.

To confirm that the *yjbI*::Tn mutation results in strong downregulation of total ChvT protein levels, we introduced a plasmid to express a ChvT-Bla translational fusion in which ChvT bearing a C-terminal translational fusion to the Bla enzyme, the endogenous metallo-β-lactamase from *C. crescentus* (Docquier et al, 2002; West et al, 2002), from the *chvT* locus in Δ*bla C. crescentus* cells (Fig. 4B). These Δ*bla chvT*::*chvT-bla* reporter cells were then transduced with the *yjbI*::Tn mutation for EOP assays on plates with or without 4 μg/mL of doripenem (DOR[4]), a β-lactam antibiotic of the carbapenem subclass that is efficiently inactivated by Bla (Docquier et al, 2002). Our EOP assays revealed that the *yjbI*::Tn derivative grows poorly on DOR[4] plates compared to the *yjbI*[+] parent. To estimate the level of resistance (MIC) in both strains, we used doripenem E-strips (harboring a defined step-gradient of doripenem) and determined an MIC of >32 μg/mL in the parent, while the *yjbI*::Tn mutation reduces the estimated MIC to 2 μg/mL (Fig. EV5C). This result in conjunction with the converse experiment described below for the BugA-Bla fusion (Figs. 4B and EV5C), confirms that *yjbI*::Tn cells lack ChvT.

Reasoning that OM remodeling should impact antibiotic permeability, we determined the antibiotic sensitivity spectrum of *yjbI* mutant cells versus *WT* cells using a library of ca. 800

**A**

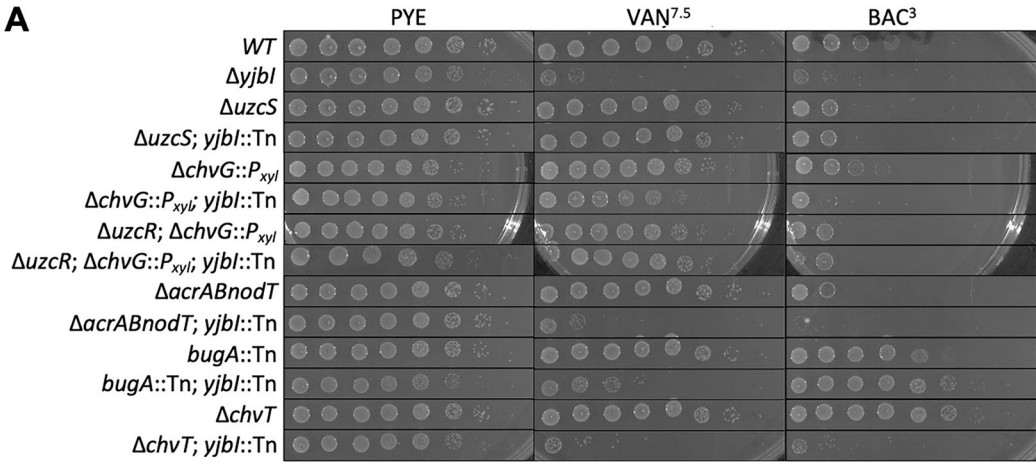

**B**

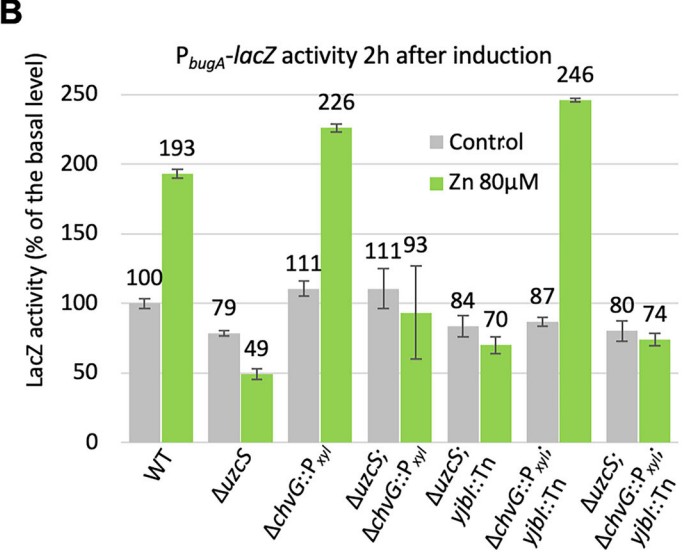

P$_{bugA}$-lacZ activity 2h after induction

**C**

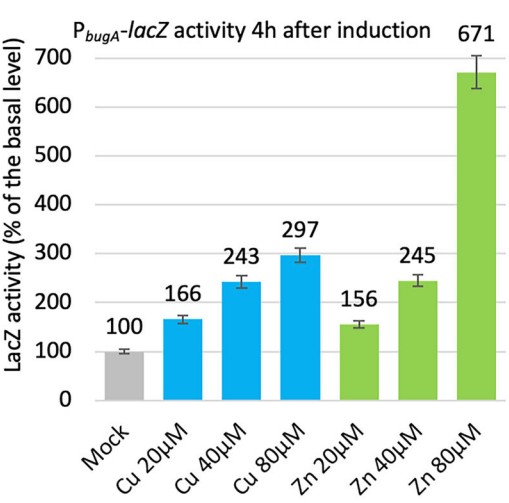

P$_{bugA}$-lacZ activity 4h after induction

**D**

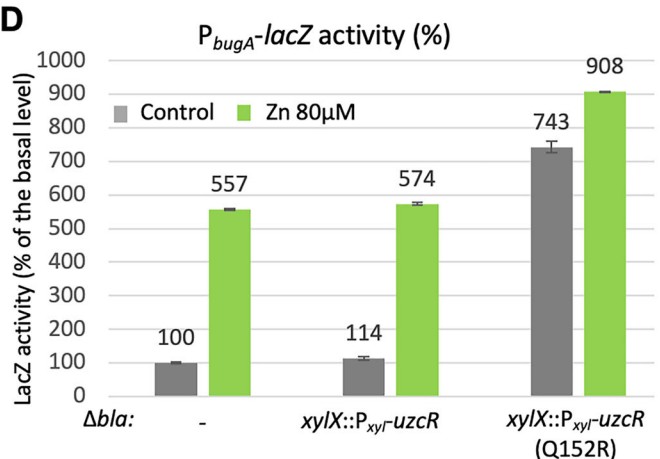

P$_{bugA}$-lacZ activity (%)

**E**

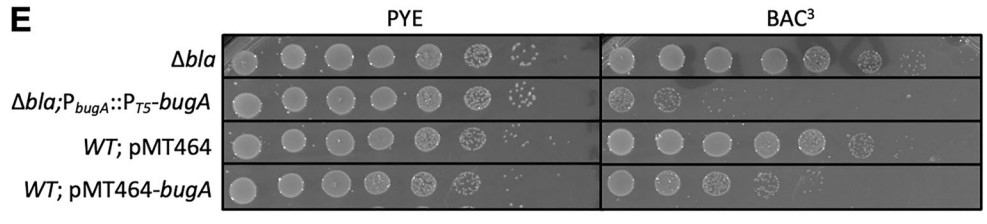

**Figure 5.  Ectopic expression of BugA confers sensitivity to bacitracin.**

(A) EOP assay of *WT* and *yjbI* mutant cells on PYE with vancomycin at 7.5 (VAN[7.5]), 20 (VAN[20]), or bacitracin (3 µg/mL, BAC[3]). Plates were incubated for 2 days at 30 °C. The comparison is to growth on PYE without treatment, as in Fig. 2A. (B–D) β-galactosidase assay measurement from *WT* and mutant cells harboring the pP*bugA*-*lacZ* reporter plasmid with or without inductions for 2 h (B, D) or 4 h (C) with CuSO4 or ZnSO4 (80 µM). All β-galactosidase measurements are reported relative to the 100% activity level of the uninduced state in *WT* cells. Error bars are defined as $+/-$ standard deviation. For (D), cells were grown in PYE containing 0.3% xylose, and for (B, C) without xylose. (E) EOP assay of *WT* and cells ectopically expressing BugA from the constitutive *E. coli* phage T5 promoter at the *bugA* locus (*bugA*:: T5-*bugA*) or from a replicative plasmid (pMT464-*bugA*). Cells were spotted on PYE plates with or without BAC[3] and incubated for 2 days at 30 °C. The comparison is to growth on PYE without treatment, as in Fig. 2A. Source data are available online for this figure.

antibiotics (Dataset EV5), individually spotted on a soft-agar indicator lawn containing either *WT* or *yjbI* mutant cells (Fig. 4G). Among the antibiotics that inhibit growth of *yjbI* cells (relative to *WT* cells) are smaller molecules known to induce the efflux pump via de-repression of TipR, including the first-generation quinolone antibiotics NAL and flumequine. *yjbI* mutant cells are also more sensitive to the FabI inhibitor AFN-1252, the first-generation MsbA-inhibitor TBT-1, and the peptide deformylase inhibitor actinonin. Our finding that TBT-1 is an inducer of P*acrA*-*lacZ* reporter (227% activity at 5 µM relative to 100% activity of uninduced cells, Fig. EV5D), suggests that TBT-1 is an efflux substrate of AcrAB-NodT. Indeed, in *E. coli* AcrAB-TolC protects against MsbA-inhibitors such as TBT-1 (Zhu et al, 2024), Hence, we suspect that our small molecule hits likely inhibit growth of *yjbI* mutant cells via AcrAB-NodT induction.

Interestingly, our antibiotic profiling also revealed several large soluble antibiotics that inhibit *yjbI* cells more potently than *WT* cells, for example, the Gyrase B inhibitor and LptB activator novobiocin (May et al, 2017) and the two glycopeptide antibiotics ristomycin and vancomycin (VAN, and its derivatives, see below). The finding of the VAN sensitivity of Δ*yjbI* cells is unexpected since VAN is not an efflux substrate. Therefore, we hypothesized that differential expression of specific OM proteins in *yjbI* mutant cells might permit the internalization of large antibiotics like VAN (see below). By contrast, our screen also revealed that the *yjbI* mutation protects against other types of large antibiotics, for example, the macrolides gamithromycin, erythromycin, spiramycin, roxithromycin, azithromycin, and solithromycin, and the phosphoglycolipid moenomycin (Fig. EV5E,F). This finding might be reconciled by downregulation of other TBDRs such as ChvT (CCNA_03108) or CCNA_000974 in *yjbI* mutant cells *versus WT* cells (see above) or upregulation of efflux. Finally, we also observed that several 3rd or 4th generation fluoroquinolones, such as levofloxacin, enrofloxacin, ciprofloxacin, difloxacin, and trovafloxacin, that are small enough to enter through porins (Delcour, 2009; Mach et al, 2008), are less active against Δ*yjbI versus WT* cells, possibly owing to a change in efflux activity.

In summary, YjbI plays a multifactorial role in envelope stress adaptation, likely through control of the OM proteome and efflux, manifested as a profound change in sensitivity (directly or indirectly) towards functionally and structurally diverse antibiotic groups.

## YjbI silences BugA, a conserved TBDR that confers bacitracin sensitivity

Since *yjbI* mutant cells do not express ChvT, a TBDR that confers sensitivity to VAN (Figs. 1C and 4A), we predicted that *yjbI* mutant

cells should be more resistant to VAN than *WT* cells. However, the screening above suggested the opposite to be true. Indeed, Δ*yjbI* cells exhibit a drastic decrease in EOP on VAN[7.5] compared to *WT* cells (Fig. 5A). However, we also noted a reduced EOP of Δ*yjbI* cells on plates containing 3 µg/mL of the control antibiotic bacitracin (BAC) compared to *WT* cells (Fig. 5A). BAC is a large PG-targeting peptide antibiotic that binds zinc, and that is excluded by an impermeable OM barrier. The BAC sensitivity is abolished upon expression of YjbI from plasmid pMT335-*yjbI* in *yjbI* mutant cells (Fig. EV5B). To determine the basis for this BAC sensitivity, we mutagenized *yjbI* mutant cells with a *himar1* Tn and selected for survivors on BAC[3] plates. We isolated two mutants (LT2399 and LT2402), each harboring a Tn insertion in the TBDR-encoding gene CCNA_00224 (henceforth called *bugA*, for BAC uptake gene A). EOP assays revealed that the *bugA*::Tn mutation reversed the growth defect of *yjbI* mutant cells on BAC[3] plates and on VAN[7.5] plates (Fig. 5A). Thus, *bugA* confers both VAN and BAC sensitivity in *yjbI* mutant cells.

While BugA is poorly expressed in *WT* cells, it is among the most abundant proteins detected by LC-MS/MS in the OM-enriched fraction prepared from Δ*yjbI* cells (Fig. 4C), but no longer detectable in Δ*uzcS yjbI*::Tn and Δ*chvG yjbI*::Tn cells (Fig. 4D,E). Similarly, 2 h after exposure of *WT* cells to Zn, BugA was detectable by LC-MS/MS analysis of OM-enriched fractions (Fig. EV5A). The *bugA* transcript is poorly expressed in unstressed cells (Fig. EV1), but the *bugA* promoter (P*bugA*) is among the top targets of UzcR (Dataset EV2) (Park et al, 2019; Park et al, 2017) and using a pP*bugA*-*lacZ* promoter probe reporter, we confirmed that P*bugA* fires when cells are stressed with 80 µM Zn or Cu (Fig. 5C). This induction no longer occurs in Δ*uzcS* single mutant or in Δ*uzcS yjbI*::Tn double mutant cells (Fig. 5B). We were unable to transform the pP*bugA*-*lacZ* plasmid into Δ*yjbI* mutant cells, presumably due to accumulation of toxic levels of LacZ from P*bugA* hyperactivity when YjbI is genetically (constitutively) inactivated. However, we engineered an Δ*bla bugA*::*bugA-bla* strain in which Bla is translationally fused to the C-terminus of BugA, encoded at the *bugA* locus. Upon transduction of the *yjbI*::Tn mutation into Δ*bla bugA*::*bugA-bla* cells, we observed an increase in EOP by five orders of magnitude on plates harboring DOR[4] compared to the parental reporter. We estimated an MIC of > 32 µg/mL in cells harboring *yjbI*::Tn versus 4 µg/mL in the parental reporter strain using DOR E-strips (Figs. 4A and EV5B).

Since UzcS is required for the induction of P*bugA*-*lacZ* and UzcR binds P*bugA*, we anticipated that UzcR would be sufficient for the induction of P*bugA*-*lacZ*. Indeed, when the gain-of-function mutant UzcR(Q152R) was expressed from the *xylX* locus (*xylX*::P*xyl*-*uzcR*-*Q152R*) in *WT* cells harboring pP*bugA*-*lacZ*, 743% relative LacZ activity was measured *versus* to the 100% P*bugA*-*lacZ* activity of

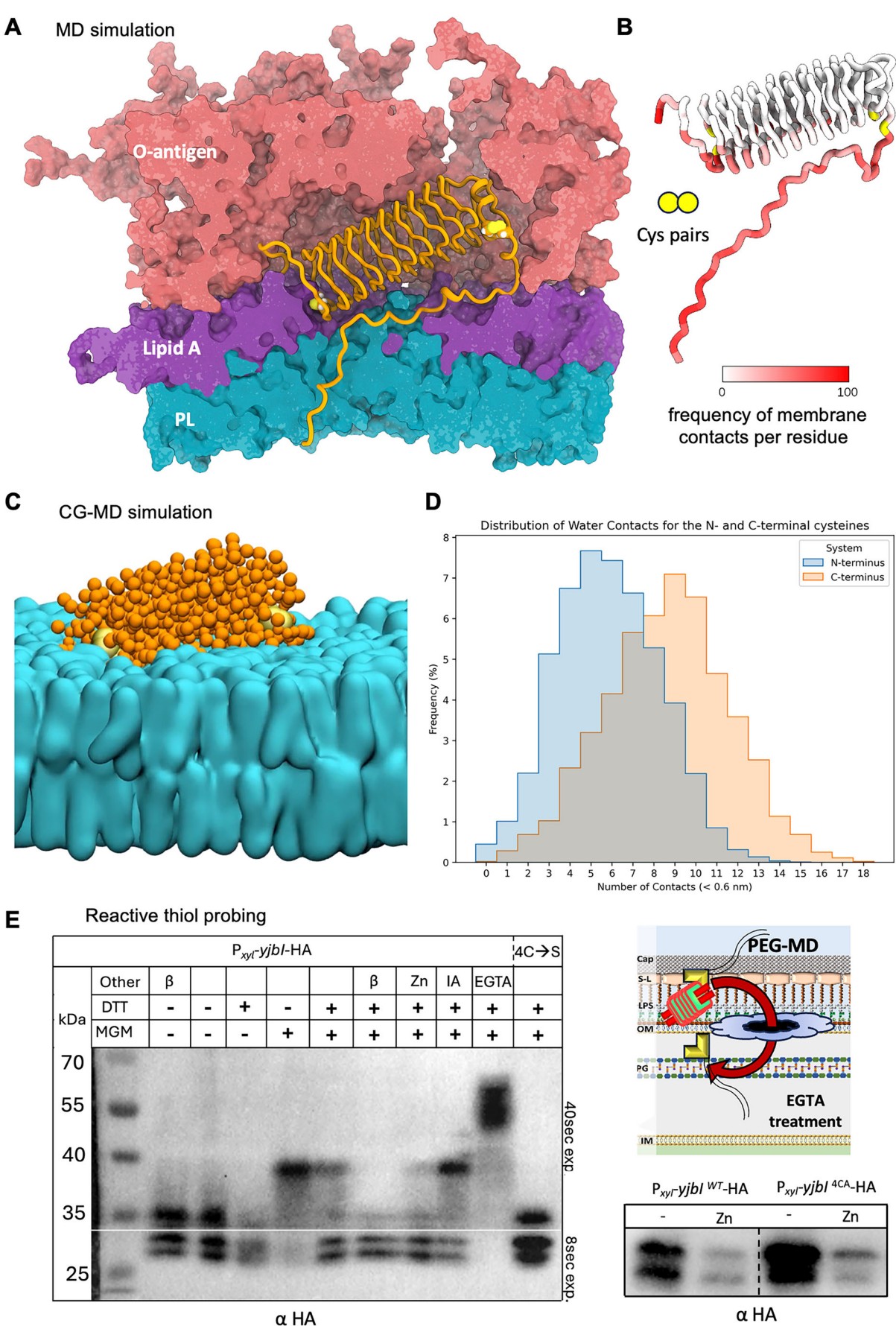

**A** MD simulation

O-antigen

Lipid A

PL

**B**

Cys pairs

0          100

frequency of membrane
contacts per residue

**C** CG-MD simulation

**D** Distribution of Water Contacts for the N- and C-terminal cysteines

**E** Reactive thiol probing

α HA

P$_{xyl}$-yjbl$^{WT}$-HA    P$_{xyl}$-yjbl$^{4CA}$-HA

α HA

◄ **Figure 6.   Asymmetric surface disposition of YjbI.**

(**A**) Cartoon model of a cross-section of the YjbI structure (orange) embedded within the OM as revealed by molecular dynamics (MD) simulations. The OM components include: the O-antigen (pink) and lipid A (purple) components of LPS, and the phospholipid-based inner leaflet (cyan) composed of cardiolipin and DOPG (Dioleoylphosphatidylglycerol). (**B, C**) Model of the contact sites of YjbI's residues with the OM (**B**) as determined by the coarse-grained (CG) simulation. In (**B**), each residue is colored, from white to red, by the frequency (%) with which they were in contact with the membrane (distance <0.6 nm). The N-terminal and C-terminal cysteine pairs are colored in yellow. Panel (**C**) shows a still image of the dynamic CG simulation. Color code as in (**A**). (**D**) Distribution of water contacts to YjbI to reveal the exposure of cysteine residues to the aqueous phase. Frequency (%) in which each pair of cysteines was in contact with a specific number of water beads (distance <0.6 nm) during the simulations. The distribution for the C-terminal pair of cysteines is colored in orange, and the one for the N-terminal pair, in blue. (**E**) Reactive cysteine probing on the cell surface using methoxy-polyethylene glycol maleimide (MGM), revealing free and masked cysteines YjbI-HA. Blots containing cells were probed with anti-HA antibodies on the strain *xylX*::P*xyl*-*yjbI*-2xHA *WT* or the 4CS mutant derivative (shown on the right) before or after zinc induction. In the 4CS derivative of YjbI-HA, all four cysteine residues of YjbI have been changed to serine. Intact cells were exposed to 500 µM MGM, adding 5 kDa PEG to the protein for each cysteine linked to MGM. Some samples were boiled in 1× Laemmli buffer (β) to break disulfide bonds or treated with 20 mM dithiothreitol (DTT) prior to MGM treatment to reduce cysteine residues without boiling. During the MGM treatment, cells were treated with 80 µM ZnSO4 (Zn), 1 mM iodoacetamide (an alkylation agent) to compete with the reaction to MGM (IA) or 10 mM EGTA (EGTA) to perforate the OM and S-layer of *C. crescentus* (according to the scheme on the right). All samples (except the samples labeled as β) were resuspended in non-reducing buffer. Source data are available online for this figure.

(uninduced) *WT* cells (Fig. 5D). By comparison, 80 µM Zn induction of *WT* cells yielded 557% relative P*bugA*-*lacZ* activity (Fig. 5D). For P*chvR*-*lacZ*, we found that UzcR(Q152R) also suffices for its induction to 183% activity (Fig. 5D), whereas phosphomimetic ChvI(D52E) did not significantly improve P*chvR*-*lacZ* activity in PYE (Fig. EV5G), unless cells are grown in M2G to activate ChvGI (Frohlich et al, 2018).

The result that expression of UzcR(Q152R) induces P*bugA*, even in the absence of Zn, and that it renders cells sensitive to BAC[3] (Fig. EV6A), suggests that BugA expression is sufficient to internalize BAC[3]. Three lines of evidence support this conclusion. First, a *bugA*::Tn mutation mitigates the BAC[3] sensitivity of *xylX*::P*xyl*-*uzcR-Q152R* cells (Fig. EV6A). Second, BugA expression is sufficient to confer BAC[3] sensitivity to *WT* and Δ*chvT* Δ*exbD* cells. Using a strain constitutively expressing BugA from the *E. coli* phage T5 promoter (Δ*bla bugA*::P*T5*-*bugA*), we observed a substantial reduction in EOP on plates containing BAC[3] compared to control cells (Fig. 5E). Similarly, expression of BugA from a replicative plasmid (pMT464-BugA) in *WT* cells induced BAC[3] sensitivity compared to cells with the empty vector (pMT464, Figs. 5E and EV6B–D). The pMT464-*bugA* plasmid also causes a reduction in EOP on VAN[20] and VAN[40] to *WT*, Δ*exbD*, and Δ*chvT* Δ*exbD* double mutant cells, indicating that BugA can function independently of ExbD (CCNA_00324) (Fig. EV6D). Lastly, while none of five other TBDRs tested (expressed from pMT464) had this effect on BAC[3], the BugA ortholog from *Brevundimonas subvibrioides* (BugA[Bs]) expressed from pMT464 also confers BAC[3] and VAN[10] (Fig. EV6D), indicating that BugA function is conserved. Based on these results, we conclude that YjbI silences UzcSR signaling and thus BugA expression.

## Asymmetric surface disposition of YjbI in the LPS layer

To explore how YjbI fulfils its regulatory function as OM sentinel, we first verified that the predicted Sec-dependent secretion signal at the N-terminus of YjbI (residues 1–24) suffices to direct synthetic metallo-beta-lactamase NDM1 or Bla (both lacking their own secretion signal) into the extracytoplasmic space to confer doripenem resistance to *C. crescentus* Δ*bla* cells (Fig. EV6E). Next, we used molecular dynamics (MD) simulations to determine if YjbI resides in the lipopolysaccharide (LPS)-containing layer of the OM. For these simulations, we used an OM composition of 100% lipopolysaccharide (LPS) in the outer leaflet of the OM, a 9:1 ratio

of phosphatidylglycerol:cardiolipin (PG:CL) in the inner leaflet and the AlphaFold model of YjbI (retrieved from the AlphaFold Protein Structure Database, AFDB) from which the secretion signal peptide had been removed. The simulation revealed that the surrounding LPS provides a stabilizing environment for YjbI, enabling YjbI's β-helix to align laterally with LPS (Fig. 6A; Movie EV1). The charged surface of YjbI may facilitate interactions with lipid A or with the cations that neutralize charge repulsion between lipid A molecules. Moreover, MD showed the charged C-terminal tail of YjbI to be embedded inside the bilayer as a transmembrane segment in a hook-like configuration (Fig. 6B), reaching across the outer leaflet to the inner leaflet, a configuration that was maintained during the entire simulation. Since the quadrilateral β-helix of YjbI predicts two flanking cysteine pairs, one at the N-terminal end and the other near the C-terminus, we paid particular attention to the disposition of these cysteines within the membrane layer. Strikingly, the N-terminal cysteine pair is deeply buried within the alkyl chains of the lipid A moiety of LPS, while the C-terminal pair is mostly solvent exposed (Fig. 6A).

To ensure that the initial placement of YjbI at the outset of the modeling does not bias the outcome, we also performed coarse-grained MD (CG-MD) simulations to assess i) whether YjbI spontaneously interacts peripherally with a lipid bilayer starting from bulk solvent and ii) whether the C-terminal cysteines are indeed more solvent accessible than the N-terminal ones (Fig. 6C,D; Movie EV2). In these simulations, YjbI diffused towards the lipid bilayer and always interacted peripherally and stably with it during the remainder of the runs. Importantly, the interaction surface of YjbI with the bilayer was the same in all replicates, matching atomistic MD. To estimate the placement depth of the four cysteine residues relative to the LPS, we computed the number of water beads in contact with each pair of cysteines (N-terminal *versus* C-terminal) for each snapshot in the CG-MD simulation (Fig. 6D). Again, the C-terminal pair was more frequently exposed to water than the N-terminal pair.

To complement these MD simulations experimentally, we resorted to reactive thiol probing of the cysteines within YjbI-HA, which recapitulated the asymmetric disposition of YjbI from MD. To this end, we used the OM-impermeable probe methoxy-polyethylene glycol maleimide (MGM) that covalently joins a 5 kDa polyethylene glycol (PEG) group to each accessible (reduced) cysteine (Fig. 6E, left). MGM was added to cells before or after the addition of the calcium chelator EGTA [(ethylene glycol

bis(2-aminoethyl ether)-N,N,N',N'-tetraacetic acid)] to disrupt the OM organization. Immunoblotting revealed a PEG-shifted YjbI-HA with an apparent mass increase of 10 kDa that was further shifted by an additional (apparent) mass of 10 kDa after EGTA treatment. Evidently, two reactive cysteine residues are surface-exposed, while two additional cysteines are shielded by calcium and/or an intact OM structure. No MGM reactivity was apparent when YjbI-(4CS)-HA was expressed in which the four cysteines of YjbI are converted to serine residues (Fig. 6E, left panel) and none of the cysteines are required for Zn-induced turnover of YjbI-HA (Fig. 6E, right panel). However, replacing three terminal lysine residues with alanines impaired YjbI's ability to support growth on NAL[10] or BAC[2.5], indicating that the C-terminus plays an important role in YjbI function (Fig. EV6F).

# Discussion

## Bimodal antibiotic sensitivity regulated by YjbI

The bimodal nature by which the secreted β-helix protein YjbI can signal adaptive responses from the cell surface to alter antibiotic sensitivity is unprecedented. We identified YjbI mutants in two independent genetic selections, one based on the regulation of TBDR expression and another using a negative selection for sensitivity to NAL, aimed at uncovering mutants that do not tolerate induction of the *acrAB-nodT* efflux pump operon. After having discovered the NAL-sensitivity of *yjbI* mutants, we proceeded to survey its antibiotic sensitivity on a larger scale by surveying an 800-compound antibiotic library. The fact that the *yjbI* is also more resistant to different classes of antibiotics (Fig. 4G), such as moenomycin and gamithromycin (Fig. EV5E,F), argues against a general permeability defect as known for the permeability mutants of *E. coli* in which OM barrier function is compromised due to mutations in LptD, LptA, or BamB (Bos et al, 2004; Falchi et al, 2018; Krishnamoorthy et al, 2016; Ruiz et al, 2005). We found that *yjbI* mutant cells are sensitive toward two groups of antibiotics, a group of small molecules and a group of large compounds (Fig. 4H). The two underlying mechanisms were not previously known to be jointly regulated. In the latter, we discovered a genetically controlled switch in TBDR expression (Fig. 4H), relying on the strong induction of BugA, an uncharacterized TBDR that mediates the internalization of large antibiotics such as VAN and BAC. The former path is governed by small antibiotics such as the quinolones NAL and flumequine, or other molecules such as TBT-1 (tetrahydrobenzothiophene 1, Fig. EV5D), that penetrate the OM efficiently to cause toxicity of *yjbI* mutant cells by inducing de-repression of the *acrAB-nodT* operon. Such de-repression triggers a surge in AcrAB-NodT abundance that could curb growth of *yjbI* mutant cells, possibly by OM crowding or bottlenecks in the BAM system required for insertion of β-barrel proteins in the OM. Consistent with this notion, we also found that (i) inactivation of the *tipR* gene, encoding the repressor of the *acrAB-nodT* operon, phenocopies the toxic effect of NAL on *yjbI* mutant cells and (ii) the toxicity of the *tipR* mutation is mitigated in the presence of a genetically linked promoter-down mutation in P$_{acrA}$. Indeed, previous electron-cryo tomography (ECT) provided evidence for envelope deformations arising from massive expression of *acrAB-nodT* in *C. crescentus* WT

cells from a high-copy plasmid in the presence of NAL (Costafrolaz et al, 2023). Therefore, we speculate that the envelope of *yjbI* mutant cells is in a partially crippled or stressed state, rendering it less tolerant toward additional perturbations (for example, induction of *acrAB-nodT* operon) than that of *WT* cells. Having indeed confirmed that the *yjbI* mutant OM proteome is massively reprogrammed versus the *WT* OM proteome, it is conceivable that *yjbI* cells are predisposed to envelope stress caused by OM proteins.

A priori, the notion that induction of the *acrAB-nodT* operon can be toxic to cells is surprising and counterintuitive, since the induction of efflux pump expression is normally considered a beneficial trait for bacteria, protecting them against antibiotics. However, RND-pump induction also brings about a massive consummation of proton motive force energy to fuel efflux activity, a drainage that occurs on the heels of the costly insertion of a large trans-envelope machine. It is therefore not surprising that RND-pump expression is typically tightly controlled and restricted to a transient burst of expression until the antibiotic (or toxic molecule) is expelled. The expulsion enables the rapid return to the repressed (ground) state, thereby avoiding a state of constitutive AcrAB-NodT expression that may be disadvantageous to *WT* cells, let alone for mutant cells with a crippled envelope. Yet, many multidrug-resistant clinical isolates have become genetically locked in a constitutive RND-pump expression state to deal with constant antibiotic challenge during treatment. This raises the possibility that such constitutive RND-pump producers could be sensitive to molecules that are not efflux substrates but capable of inducing envelope stress in these mutants whereas they have negligable effects in *WT* cells.

While the bulk of RND-pump induction is typically caused by de-repression of negative regulators such as TipR, a TetR-family repressor, we found that induction of AcrAB-NodT in *C. crescentus* can also be bolstered or fine-tuned by transcriptional activation, for example, in response to envelope stress. We detected an elevated basal (uninduced) P$_{acrA}$ activity in *yjbI* mutant cells, even in the absence of de-repression, owing to transcriptional activation by the response regulator UzcR. Interestingly, DjlA, a DnaJ-like co-chaperone that enhances AcrAB-NodT function (Costafrolaz et al, 2023), is also expressed from a promoter (P$_{djlA}$) that is repressed by TipR and activated by UzcR (Costafrolaz et al, 2023; Park et al, 2017), akin to P$_{acrA}$. We showed that P$_{acrA}$ and P$_{djlA}$ firing is magnified in the presence of zinc (to activate UzcR) and NAL (to induce de-repression of TipR). As zinc stress induces the proteolytic removal of YjbI and NAL counteracts TipR, joint induction with NAL and zinc augments firing of both P$_{acrA}$ and P$_{djlA}$, (Fig. EV4C), resulting in maximal AcrAB-NodT assembly in the OM, which is further enhanced by DjlA at the post-translational level and may potentially aggravate OM stress or crowding. Evidently, *yjbI* mutant cells still benefit from high AcrAB-NodT-mediated efflux of toxic molecules such as ethidium bromide, a less potent inducer of P$_{acrA}$. compared to NAL (Costafrolaz et al, 2023) (Fig. 2E). Loss of AcrAB-NodT attenuates the toxicity of NAL in *yjbI* mutant cells, and it does not substantially affect NAL resistance in *WT C. crescentus* cells because of the natural polymorphism in GyrA that confers resistance towards NAL (Kirkpatrick and Viollier, 2014). The *yjbI* mutation could cause crowding stress in the OM by TBDRs such as BugA that are massively expressed, limiting available space for insertion of NAL-induced RND-pumps or of additional OM

proteins. Consistent with this notion, the sensitivity to NAL can be mitigated by mutations in *uzcR* or in *uzcS* that will simultaneously prevent BugA induction, while also dampening *acrAB-nodT* production in the presence of NAL (Fig. EV4A). Even though MD simulations and reactive thiol probing reveal YjbI's unusual arrangement on the OM surface, our findings overall indicate that the principal defect caused by loss of YjbI is primarily due to mis-regulation of gene expression rather than a severe structural defect in the OM.

## TBDRs, oligotrophy, and antibiotic permeability of the outer membrane

A manifestation of the multifactorial role of YjbI as OM sentinel is that it inversely regulates TBDRs: while the loss of YjbI silences expression of ChvT and the uncharacterized TBDR CCNA_00974, it induces other TBDRs, including BugA and the uncharacterized CCNA_00858 TBDR. Our genetic identification of the conserved BugA and our demonstration that its expression suffices to confer sensitivity to VAN and BAC broaden the evidence that TBDRs can also internalize large PG-targeting (glyco)peptides (Fig. 5A,E). Since VAN and BAC sequester essential PG precursors in the periplasmic space, it is conceivable that they are mistakenly imported by TBDRs as PG precursors or recycling products. Moreover, we show here that BugA expression is strongly induced by zinc stress. Knowing that VAN and BAC both bind zinc (Economou et al, 2013; Zarkan et al, 2017; Zarkan et al, 2016) and the effect of zinc on PG homeostasis (discussed below), the induction of BugA by zinc may reflect a PG adaptive response. As zinc stress is a common protective response by eukaryotic host cells to combat bacteria, it may be possible to leverage TBDR-dependent internalization of BAC or VAN (derivatives) as Trojan-Horse antibiotics (sideromycins) to kill diderm bacteria that express a suitable importing TBDR. In fact, BAC can also enter *Mycobacterium tuberculosis* (*Mtb*) cells through a multi-solute transporter that also allows the recently discovered gyrase-targeting antibiotic evybactin to traverse the OM (mycomembrane) of *Mtb* (Imai et al, 2022).

In *C. crescentus*, BAC and VAN also engage the constitutively expressed TBDR ChvT, but perhaps less efficiently since expression of BugA substantially increases the sensitivity to these antibiotics, regardless of whether cells express ChvT or not (Fig. 5A,B). Since zinc-stressed cells execute a TBDR switch from ChvT to BugA, the reprogrammed OM may contain a TBDR with improved uptake kinetics of PG-like (or related) products. The TBDR-dependent permeability of the *C. crescentus* OM towards large antibiotics likely underlies the oligotrophic lifestyle and habitat that necessitates nutrient assimilation by active uptake in a nutrient-sparse environment. As most free-living bacteria encode a multiplicity of TBDRs, differential expression of TBDRs may represent a common regulatory concept that could also influence antibiotic permeability of the OM in other Gram-negative bacteria, even in human pathogens, under stress conditions. If such conditions can be defined, then sideromycins (Wencewicz and Miller, 2018) may serve as a narrow-spectrum antibiotic treatment regime achieved in other ways (Muñoz et al, 2024), since TBDR-mediated import of antibiotics is likely genus- or class-specific. Thus, treatment with TBDR-imported antibiotics (Luna et al, 2020) would also reduce the risk of pervasive dissemination of antibiotic resistance to co-inhabiting microbes.

The current challenge is also understanding how TBDR expression is regulated and enlarging our repertoire of how chemical "trigger"-groups engage TBDRs to induce uptake, for example in suitably conjugated antibiotics. Insight into differential TBDR expression could also lead to new (combined) options to treat mixed infections by Gram-negative and Gram-positive bacteria. Based on our findings, we also suggest that antibiotics inducing *acrAB-nodT* operons or related tripartite efflux systems may sensitize bacteria towards antibiotics that target the envelope, including the BAM system (Imai et al, 2019; Kaur et al, 2021; Miller et al, 2022), which inserts NodT and TBDRs in the OM.

## Structure–function relationship of YjbI proteins

The conserved features of the YjbI and its orthologs include (i) the secretion signal directing YjbI into the extracytoplasmic space, (ii) the two conserved cysteine pairs that are differentially embedded in the LPS environment, and (iii) the charged C-terminal tail that permeates the OM as a transmembrane segment and is required for function. Whereas many β-helical proteins exist in the extracyto-plasmic environment, including the LptD bridge or phage tail-spike proteins, YjbI is unusual in its predicted quadrilateral β-helix architecture. Intriguingly, the crystal structure of the YjbI ortholog from *Crocosphaera subtropica* (PDB ID: 2G0Y) (Buchko et al, 2006) reveals three calcium ions bound along one face of the quadrilateral β-helix protein. Calcium ions are abundant in the OM and needed to balance charge repulsion between negatively charged LPS molecules that also surround YjbI. Since zinc stress acts as a trigger of the YjbI-dependent transcriptional response and YjbI instability, it is possible that forced exchange of calcium with zinc could alter the LPS organization and the embedding of YjbI. Zinc stress could also result, directly or indirectly, in structural deformations of YjbI to render it susceptible to proteolysis by specific extracytoplasmic proteases, for example, CtpA-type carboxyl-terminal processing proteases (Sommerfield and Darwin, 2022), possibly in the periplasm via its C-terminal tail that penetrates the OM. Interestingly, there is also precedence for another β-helix protein, GlmU from *Streptococcus pneumoniae*, to bind Zn ions (Brazel et al, 2022). GlmU is an essential and bifunctional cell wall biosynthetic metalloenzyme in the cytoplasm that synthesizes UDP-linked N-acetyl glucosamine (UDP-GlcNAc) through a globular uridyl-transferase domain in the N-terminal region, but it also possesses a β-helical acetyltransferase domain in the C-terminal region whose activity is impaired when it binds Zn ions (Brazel et al, 2022). Excess zinc could also affect cell wall remodeling or recycling enzymes (Cook et al, 2023; Micelli et al, 2023). The β-lactamase of *C. crescentus* (Bla) is also a zinc-dependent metalloenzyme whose activity could potentially be affected by zinc levels (Brem et al, 2016; Docquier et al, 2002; West et al, 2002). Thus, mis-metallation by zinc could alter the substrate spectrum of a metal-dependent periplasmic protease that destroys YjbI to jointly unleash UzcSR and ChvGI signaling.

It remains to be determined whether the *Sinorhizobium meliloti* YjbI ortholog (SM2011_c04201) plays a role in envelope stress sensing. In the rhizobia, proteolytic destruction of the negative regulator ExoR underlies ChvGI activation (Reed et al, 1991). ExoR is a small, soluble α-helical protein located in the periplasm that binds and inhibits the ChvG kinase (Chen et al, 2008). While it is conceivable that stress perception in rhizobia also occurs at the OM

and is then relayed to a periplasmic connector, *C. crescentus* does not encode a detectable ExoR ortholog (Nierman et al, 2001), but rhizobial genomes feature genes encoding UzcSR homologs. The simultaneous activation of *C. crescentus* UczSR and ChvGI upon removal of YjbI, not underlie the bimodal induction of envelope stress response that alters antibiotic sensitivity, but it also likely plays a key role in terminating the response after the stress subsides or the stress has been relieved. With YjbI expression responding positively to activated UzcR, the genetic circuitry with YjbI is ideally designed to replenish YjbI, favoring a reset to the ground (unstressed) state.

# Methods

### Reagents and tools table

| Reagent/resource | Reference or source | Identifier or catalog number |
|---|---|---|
| **Experimental models** | | |
| NA1000 | PMID: 334726 | *C. crescentus* WT |
| Δ*chvT* | PMID: 32371598 | NA1000 Δ*chvT* |
| Δ*exbD* | This study | NA1000 Δ*exbD* |
| Δ*chvT*Δ*exbD* | This study | NA1000 Δ*chvT*Δ*exbD* |
| Δ*uzcS* | PMID: 28035693 | NA1000 Δ*uzcS* |
| Δ*uzcR* | PMID: 28035693 | NA1000 Δ*uzcR::tet* |
| Δ*chvG*::P$_{xyl}$ | This study | Integration of pUCIDT-Km$^R$ *chvG*::P$_{xyl}$ into NA1000 |
| *yjbI*::Tn$^{NAS4}$ | This study | *yjbI*::Tn (HyperMu, Km$^R$), aka *NAS4* |
| *yjbI*::Tn (*LT1982*) | This study | *yjbI*::Tn (Mar2xT7, Gm$^R$), aka *LT1982* |
| *yjbI*::Tn (*LT1983*) | This study | *yjbI*::Tn (Mar2xT7, Gm$^R$), aka *LT1983* |
| Δ*yjbI* | This study | NA1000 Δ*yjbI* |
| Δ*uzcS*; *yjbI*::Tn | This study | Transduction from LT1982 into Δ*uzcS* |
| Δ*chvG*::P$_{xyl}$; *yjbI*::Tn | This study | Transduction from LT1982 into Δ*chvG*::P$_{xyl}$ |
| Δ*uzcS*; Δ*chvG*::P$_{xyl}$ | This study | Transduction from Δ*chvG*::P$_{xyl}$ into Δ*uzcS* |
| Δ*uzcS*;Δ*chvG*::P$_{xyl}$; *yjbI*::Tn | This study | Transduction from LT1982 into Δ*uzcS* Δ*chvG*::P$_{xyl}$ |
| Δ*acrAB-nodT* | PMID: 24726830 | |
| Δ*acrAB-nodT*; *yjbI*::Tn$^{NAS4}$ | This study | Transduction from NAS4 into Δ*acrAB-nodT* |
| *bugA*::Tn | This study | *bugA*::Tn (Mar2xT7, Gm$^R$) aka *LT2399* |
| *bugA*::Tn | This study | *bugA*::Tn (Mar2xT7, GmR) aka *LT2402* |
| *bugA*::Tn; *yjbI*::Tn$^{NAS4}$ | This study | Transduction from NAS4 into *bugA*::Tn |
| Δ*chvT*; *yjbI*::Tn$^{NAS4}$ | This study | Transduction from NAS4 into Δ*chvT* |
| *yjbI*::Tn; *xylX*::P$_{xyl}$-*yjbI-HA* | This study | Transduction from LT1982 into *xylX*::P$_{xyl}$-*yjbI-HA* |
| *xylX*::P$_{xyl}$-*yjbI-HA* | This study | Integration of pUCIDT-Km$^R$ P$_{xyl}$-*yjbI*-HA into NA1000 |
| Δ*yjbI*; *xylX*:: P$_{xyl}$-*yjbI-HA* | This study | Transduction of *xylX*::P$_{xyl}$-*yjbI-HA into* Δ*yjbI* |
| Δ*uzcR*; *xylX*::P$_{xyl}$-*HA-uzcR* | This study | Integration of pUCIDT-Km$^R$ P$_{xyl}$-*HA-uzcR* into Δ*uzcR* |
| Δ*uzcR*; *xylX*::P$_{xyl}$-*HA-uzcR*; *yjbI*::Tn | This study | Transduction from LT1982 into Δ*uzcR* *xylX*::P$_{xyl}$-*HA-uzcR* |
| Δ*bla*; *xylX*::P$_{xyl}$-*HA-chvI* | This study | Integration of pUCIDT-Km$^R$ P$_{xyl}$::*HA-chvI* into Δ*bla* |
| Δ*bla*; *xylX*::P$_{xyl}$-*HA-chvI*; *yjbI*::Tn | This study | Transduction from LT1982 into Δ*bla*; *xylX*::P$_{xyl}$-*HA-chvI* |
| Δ*bla* | PMID: 11914347 | NA1000 Δ*bla* |
| Δ*bla* Δ*lacA* | PMID: 20190087 | NA1000 Δ*bla* Δ*lacA* |
| Δ*bla*; *yjbI*::Tn$^{NAS4}$ | This study | Transduction from NAS4 into Δ*bla* |
| Δ*bla* Δ*lacA*; *yjbI*::Tn | This study | Transduction from LT1982 into Δ*bla* Δ*lacA* |
| Δ*bla* Δ*lacA*; *yjbI*::Tn; *uzcR** (*LT2333*) | This study | NAL20R isolate of Δ*bla* Δ*lacA*; *yjbI*::Tn |
| Δ*bla* Δ*lacA*; *yjbI*::Tn; *uzcR** (*LT2335*) | This study | NAL20R isolate of Δ*bla* Δ*lacA*; *yjbI*::Tn |
| Δ*bla*; *xylX*::P$_{xyl}$-*uzcR* | This study | Integration of pUC-GW-Amp$^R$ P$_{van}$-*uzcR* into Δ*bla* |
| Δ*bla*; *xylX*::P$_{xyl}$-*uzcR-Q152R* | This study | Integration of pUC-GW-Amp$^R$ P$_{van}$-*uzcR-Q152R* into Δ*bla* |
| Δ*bla*; *xylX*::P$_{xyl}$-*chvI* | This study | Integration of pUCIDT-Km$^R$ P$_{xyl}$-*chvI* into Δ*bla* |
| Δ*bla*; *xylX*::P$_{xyl}$-*chvI-D52E* | This study | Integration of pUCIDT-Km$^R$ P$_{xyl}$-*chvI-D52E* into Δ*bla* |
| Δ*bla*; *vanA*::T5-*uzcR* | This study | Integration of pUC-GW-Amp$^R$ P$_{van}$-T5-*uzcR* into Δ*bla* |
| Δ*bla*; *xylX*::P$_{xyl}$-*chvI-D52E*; *vanA*::T5-*uzcR* | This study | Transduction of *vanA*::T5-*uzcR* into Δ*bla*; *xylX*::P$_{xyl}$-*chvI-D52E* |
| Δ*bla*; *vanA*::T5-*uzcR-Q152R* | This study | Integration of pUC-GW-Amp$^R$ P$_{van}$-*uzcR-Q152R* into Δ*bla* |
| Δ*bla*; *xylX*::P$_{xyl}$-*chvI-D52E*; *vanA*::T5-*uzcR-Q152R* | This study | Transduction of *vanA*::T5-*uzcR-Q152R* into Δ*bla*; *xylX*::P$_{xyl}$-*chvI-D52E* |
| Δ*uzcR* | This study | In-frame deletion of *uzcR* |

| Reagent/resource | Reference or source | Identifier or catalog number |
|---|---|---|
| Δ*chvI* | PMID: 36480504 | In-frame deletion of *chvI* |
| Δ*bla*; *bugA*::T5-*bugA* | This study | Integration of pUCIDT-Km^R P*xyl*-HA-*uzcR* into Δ*uzcR* |
| *tipR*::Tn | PMID: 38051727 | *tipR*::Tn (Mar2xT7, Gm^R), aka UG7660 |
| *tipR*::Tn P*acrA** | Lab collection | *tipR*::Tn (Mar2xT7, Gm^R) with linked P*acrA**, aka UG24028 |
| Δ*bla*; *yjbI*::P*van*-*yjbI* | This study | Integration of pUC-GW-AmpR P*van*-*yjbI* into Δ*bla* |
| Δ*bla*; *yjbI*::P*van*-*yjbI*; *tipR*::Tn | This study | Transduction of UG7660 into Δ*bla*; *yjbI*::P*van*-*yjbI* |
| Δ*bla*; *yjbI*::P*van*-*yjbI*; *tipR*::Tn P*acrA** | This study | Transduction of UG24028 into Δ*bla*; *yjbI*::P*van*-*yjbI* |
| Δ*bla*; *bugA*::*bugA*-*bla* | This study | Integration of pUC-GW-Amp^R *bugA*-*bla* into Δ*bla* |
| Δ*bla*; *bugA*::*bugA*-*bla* *yjbI*::Tn^NAS4 | This study | Transduction of NAS4 into Δ*bla*; *bugA*::*bugA*-*bla* |
| Δ*bla*; *chvT*::*chvT*-*bla* | This study | Integration of pUC-GW-Amp^R *chvT*-*bla* into Δ*bla* |
| Δ*bla*; *chvT*::*chvT*-*bla* *yjbI*::Tn^NAS4 | This study | Transduction of NAS4 into Δ*bla*; *chvT*::*chvT*-*bla* |
| Δ*bla*; *xylX*::T5-*NDM1*-*SSyjbI* | This study | Integration of pUC-GW-AmpR-T5-*NDM1*-*SSyjbI* into Δ*bla* |
| Δ*bla*; *xylX*::T5-*bla*-*SSyjbI* | This study | Integration of pUC-GW-AmpR-T5-*bla*-*SSyjbI* into Δ*bla* |
| Δ*bla*; *xylX*::P*xyl*-*chvT*^hyp | This study | Integration of pUCIDT-KmR-chvThyp into Δ*bla* |
| Δ*bla*; *xylX*::P*xyl*-*yjbI* | This study | Integration of pUC-GW-AmpR-P*xyl*-*yjbI* into Δ*bla* |
| Δ*bla*; *xylX*::P*xyl*-*yjbI*-*mut* | This study | Integration of pUC-GW-AmpR-P*xyl*-*yjbI*-*mut* into Δ*bla* |
| Δ*bla*; *xylX*::P*xyl*-*yjbI* *yjbI*::Tn^NAS4 | This study | Transduction *yjbI*::Tn^NAS4 into Δ*bla*; *xylX*::P*xyl*-*yjbI* |
| Δ*bla*; *xylX*::P*xyl*-*yjbI*-*mut* *yjbI*::Tn^NAS4 | This study | Transduction *yjbI*::Tn^NAS4 into Δ*bla*; *xylX*::P*xyl*-*yjbI*-*mut* |
| EC100D | PMID: 334726 | *E. coli* laboratory strain |
| BW25113 Δ*tolC* | PMID: 32371598 | *E. coli tolC* mutant |
| **Recombinant DNA** | | |
| *pSRK-Gm* | PMID: 18606801 | |
| *pSRK-fhuAh* | PMID: 37728121 | pSRK Gm harboring *fhuA*^hyp |

| Reagent/resource | Reference or source | Identifier or catalog number |
|---|---|---|
| *pMT464* | PMID: 17959646 | |
| *pMT464-bugA* | This study | *bugA* cloned into pMT464 |
| *pLac290-P*chvR* | PMID: 32371598 | P*chvR* cloned into pLac290 |
| *pLac290-P*acrA* | PMID: 24726830 | P*acrA* cloned into pLac290 |
| *pLac290-P*bugA* | This study | P*bugA* cloned into pLac290 |
| *pMT335* | PMID: 17959646 | |
| *pMT335-yjbI* | This study | *yjbI* cloned into pMT335 |
| *pMT464-TBDR-CCNA_00171* | This study | *CCNA_00171* cloned into pMT464 |
| *pMT464-TBDR-CCNA_00185* | This study | *CCNA_00185* cloned into pMT464 |
| *pMT464-TBDR-CCNA_00214* | This study | *CCNA_00214* cloned into pMT464 |
| *pMT464-TBDR-CCNA_00486* | This study | *CCNA_00486* cloned into pMT464 |
| *pMT464-TBDR-CCNA_01194* | This study | *CCNA_01194* cloned into pMT464 |
| pUCIDT-KmR-P*xyl*-*chvT*^hyp | IDT/Dataset EV6 | ChvT^hyp expressed from P*xyl* |
| *pMT464-chvT*^hyp | This study | *chvT*^hyp cloned into pMT464 |
| *pNPTS138-ΔuzcR* | This study | *uzcR* deletion plasmid |
| *pNPTS138-Δyjbl* | This study | *yjbl* deletion plasmid |
| pUC-GW-Amp^R T5-*bugA* | Genewiz/Dataset EV6 | 5'-fragment of *bugA* fused to T5 promoter |
| pUCIDT-Km^R *chvG*::P*xyl* | IDT/Dataset EV6 | *chvG* disruption plasmid |
| pUCIDT-Km^R P*xyl*-HA-*uzcR* | IDT/Dataset EV6 | HA-UzcR expressed from P*xyl* |
| pUCIDT-Km^R P*xyl*-HA-*chvI* | IDT/Dataset EV6 | HA-ChvI expressed from P*xyl* |
| pUCIDT-Km^R P*xyl*-*yjbI*-HA | IDT/Dataset EV6 | YjbI-HA expressed from Pxyl |
| pUC-GW-Amp^R P*van*-T5-*uzcR* | Genewiz/Dataset EV6 | UzcR expressed from T5 promoter |
| pUC-GW-Amp^R P*van*-T5-*uzcR*-Q152R | Genewiz/Dataset EV6 | UzcR-Q152R expressed from T5 promoter |
| pUCIDT-Km^R P*xyl*-*chvI* | IDT/Dataset EV6 | ChvI expressed from P*xyl* |
| pUCIDT-Km^R P*xyl*-*chvI*-D52E | IDT/Dataset EV6 | ChvI-D52E expressed from P*xyl* |
| *pMT335-bugA*^Bs | This study | *bugA* from *Brevundimonas subvibrioides* cloned into pMT335 |
| pUC-GW-AmpR P*van*-*yjbI* | Genewiz/Dataset EV6 | 5'-fragment of *yjbI* fused to P*van* |
| *pLac290-P*chvR*-*nptII* | This study | P*chvR*-*nptII* cloned into pLac290 |

| Reagent/resource | Reference or source | Identifier or catalog number |
|---|---|---|
| *pUC-GW-Amp^R bugA-bla* | Genewiz/Dataset EV6 | 3′ end of *bugA* translationally fused to *bla* lacking signal sequence |
| *pUC-GW-Amp^R chvT-bla* | Genewiz/Dataset EV6 | 3′ end of *chvT* translationally fused to *chvT* lacking signal sequence |
| *pUC-GW-AmpR-T5-NDM1-SSyjbI* | Genewiz/Dataset EV6 | T5 promoter expressing synthetic NDM1 with signal sequence from YjbI |
| *pUC-GW-AmpR-T5-bla-SSyjbI* | Genewiz/Dataset EV6 | T5 promoter expressing synthetic Bla with signal sequence from YjbI |
| *pUC-GW-AmpR-P_{xyl}-yjbI* | Genewiz/Dataset EV6 | YjbI expressed from $P_{xyl}$ |
| *pUC-GW-AmpR-P_{xyl}-yjbI-mut3* | Genewiz/Dataset EV6 | YjbI-mut3 expressed from $P_{xyl}$ |
| **Antibodies** | | |
| AcrA | Vallet et al, 2020 | |
| CCNA_00163 | Ardissone et al, 2014 | |
| HA Tag | EMD Milipore Corp | 05-902 R |
| **Oligonucleotides and other sequence-based reagents** | | |
| Gene fragments | Genewiz, IDT | |
| **Chemicals, enzymes, and other reagents** | | |
| Yeast extract | AppliChem | A1552.0500 |
| Peptone | Gibco | 4022026 |
| Magnesium sulfate heptahydrate | Sigma-Aldrich | 10034-99-8 |
| Calcium Chloride | Sigma-Aldrich | 10043-52-4 |
| Ortho-nitrophénýl-β-galactoside | Bio-solve-Chemical | 367-93-1 |
| Calcium Carbonate | Sigma-Aldrich | 471-34-1 |
| Potassium chloride | Sigma-Aldrich | 7447-40-7 |
| Sodium phosphate monobasic monohydrate | Sigma-Aldrich | 10028-24-7 |
| Sodium phosphate dibasic dihydrate | Sigma-Aldrich | 10049-21-5 |
| methoxypolyethylene glycol maleimide | Sigma-Aldrich | 63187-1 G |
| Roti-Hybri-Quick hybridization solution | Roth | |
| Nalidixic acid sodium salt | Sigma-Aldrich | 222-159-7 |
| Vancomycin hydrocholide | Sigma-Aldrich | 1404-93-9 |
| Doripenem hydrate | Thermofischer | 1820954-21-9 |
| Bacitracin | Sigma-Aldrich | 1405-87-4 |
| Ethylene glycol-bis(2-aminoethyl ether)-N,N,N′,N′-tetraacetic acid (EGTA) | Sigma-Aldrich | 67-42-5 |

| Reagent/resource | Reference or source | Identifier or catalog number |
|---|---|---|
| DTT | itwreagents | 3483-12-3 |
| Gamithromycin | MCE | HY-108365 |
| Moenomycin | Cayman | 15506 |
| TBT-1 | MCE | HY-124789 |
| Antibiotics Compound Library | MCE | HY-L067 |
| Peptidomimetic library | MCE | HY-L033 |
| **Software** | | |
| ImageJ 1.54 g | Rasband, W.S., ImageJ, U.S. National Institutes of Health, Bethesda, Maryland, USA, https://imagej.net/ij/, 1997-2018 | |
| Alphafold V3 | https://alphafoldserver.com/ | |
| Microsoft Office Suite | Microsoft | |
| CHARMM-GUI web server | Jo et al, 2008 | |
| **Other** | | |

## Methods and protocols

### Strains and growth conditions

*Caulobacter crescentus* NA1000 and derivatives were cultivated in peptone-yeast extract (PYE) at 30 °C in an incubator. Liquid cultures were grown on a rotary wheel, and ΦCr30-mediated generalized transductions were done as described (Ely, 1991). For electroporations into *C. crescentus*, exponential phase cells were washed twice in sterile $H_2O$ and electroporated using cooled 1 mm cuvettes at 1.8 kV using an Eppendorf Eporator. *E. coli* was grown in LB (Lysogeny Broth) with antibiotics at 30 °C under agitation in the media 0.3 mM meso-diaminopimelic acid (mDAP) was supplemented to enable growth for WM3064 derivatives (Saltikov and Newman, 2003). Plates contained 1.5% agar medium assay, and soft agar overlays were done by mixing ¼ solid medium with ¾ of liquid PYE. For the selection of *C. crescentus* on plates, antibiotics at the indicated concentrations (in µg/mL) were used: kanamycin (10), gentamycin (1), ampicillin (75), tetracycline (1), and spectinomycin/streptomycin (30/5). For liquid, when mentioned, the media was supplemented with xylose at 0.03% concentration, 0.5 µM IPTG, and 100 µM vanillate (final concentration) for inducing the expression of corresponding promoters. Transposon delivery plasmids were pHVPV414 (Viollier et al, 2004) or pMAr2xT7 (Liberati et al, 2006). Strains and plasmids are summarized in Dataset EV6.

### Efficiency of plating (EOP) assays

Petri dishes were prepared with the mentioned antibiotic(s). Overnight cultures were normalized at $OD_{600} = 0.5$. Tenfold serial dilutions were performed by transferring 20 µL of the culture in 180 µL of fresh media in 96-well plates. This step was repeated until the eight dilutions were done. Five µL of each dilution was dropped on the selected Petri dishes and incubated for 48 h at 30 °C.

### Genome-wide transposon mutagenesis coupled to deep-sequencing (Tn-Seq)

Overnight cultures of the *E. coli* donor cells S17-1 λ*pir* strain containing the *himar1*-delivery plasmid pHPV414 with a kanamycin resistance cassette (Viollier et al, 2004), and overnight cultures of the *C. crescentus* recipient cells were mixed at a ratio of 250 μL of the *E. coli* donor cells and 1 mL of *C. crescentus* recipient cells. After centrifugation, cells were resuspended in 40 μL PYE and incubated for 5 h at 30 °C on PYE agar for mating. Cells were plated either on PYE containing kanamycin 20 μg/mL and NAL[20] (control) or PYE with kanamycin 20 μg/mL, NAL[20], and VAN[20] incubated at 30 °C for 5 days. An average of 100,000 Tn mutants on the former plate and polled, while a few thousand were isolated from the latter plate and pooled. Chromosomal DNA was extracted from the pools using Ready-Lyse Lysozyme (Epicentre Lucigen) and DNAzol (ThermoFischer) according to the manufacturers' recommendations. This harvested DNA was used to generate a Tn-Seq library and submitted to the Illumina HiSeq 2500 sequencer (Fasteris, Geneva, Switzerland). The Tn insertion specific reads were sequenced using the *himar1*-based Tn-seq primer (5'-AGACCGGGGACTTATCAGCCAACCTGTTA-3') to create the single-end sequence reads (50 bp), which were mapped against the genome of the reference genome of *Caulobacter crescentus* NA1000 (NC_011916) (Nierman et al, 2001). Data was processed using the Galaxy software (https://usegalaxy.org/) and the SeqMonk software (http://www.bioinformatics.babraham.ac.uk/projects/seqmonk/) to build sequence read profiles. Using SeqMonk, the genome of *Caulobacter* was divided into 50 bp probes, and a calculated value that represents a normalized read number per million has been determined for every probe. A ratio of the reads obtained in the control condition versus the VAN treatment was calculated for each 50 bp position. Tn-Seq reads sequencing and alignment statistics are summarized in Dataset EV1.

### Chromatin immunoprecipitation coupled to deep sequencing (ChIP-Seq) analyses

Cultures of exponentially growing (OD660nm of 0.5, 80 mL per sample of culture in PYE supplemented with xylose 0.3%) *C. crescentus* cultures Δ*uzcR*; *xylX*::P$_{xyl}$-HA-*uzcR, the yjbI* mutant derivative Δ*uzcR*; *xylX*::P$_{xyl}$-HA-*uzcR; yjbI*::Tn (Gm^R), as well as Δ*bla*; *xylX*::P$_{xyl}$-HA-chvI* and the *yjbI* mutant derivative Δ*bla*; *xylX*::P$_{xyl}$.HA-chvI; yjbI*::Tn (Gm^R) were respectively supplemented with 10 μM sodium phosphate buffer (pH 7.6) and then treated with formaldehyde (1% final concentration) at room temperature for 10 min to achieve crosslinking. Subsequently, the cultures were incubated for an additional 30 min on ice and washed three times in phosphate-buffered saline (PBS, pH 7.4). The resulting cell pellets were stored at -80 °C. After resuspension of the cells in TES buffer (10 mM Tris-HCl pH 7.5, 1 mM EDTA, 100 mM NaCl) containing 10 mM of DTT, the cell resuspensions were incubated in the presence of Ready-Lyse lysozyme solution (Epicentre, Madison, WI) for 10 min at 37 °C, according to the manufacturer's instructions. Lysates were sonicated (Bioruptor Pico) at 4 °C using 15 bursts of 30 s to shear DNA fragments to an average length of 0.3–0.5 kbp and cleared by centrifugation at 14,000 rpm for 2 min at 4 °C. The volume of the lysates was then adjusted (relative to the protein concentration) to 1 mL using ChIP buffer (0.01% SDS, 1.1% Triton X-84 100, 1.2 mM EDTA, 16.7 mM Tris-HCl [pH 8.1],

167 mM NaCl) containing protease inhibitors (Roche) and pre-cleared with 80 μL of Protein-A agarose (Roche, www.roche.com) and 100 μg BSA. Five percent of the pre-cleared lysates were kept as total input samples (negative control samples). The rest of the pre-cleared lysates were then incubated overnight at 4 °C with monoclonal rabbit Anti-HA Tag antibodies at a 1:400 dilution. Immuno-complexes were captured by incubation with Protein-A agarose beads (pre-saturated with BSA) during a 2 h period at 4 °C and then, washed with low salt washing buffer (0.1% SDS, 1% Triton X-100, 2 mM EDTA, 20 mM Tris-HCl pH 8.1, 150 mM NaCl), with high salt washing buffer (0.1% SDS, 1% Triton X-100, 2 mM EDTA, 20 mM Tris-HCl pH 8.1, 500 mM NaCl), with LiCl washing buffer (0.25 M LiCl, 1% NP-40, 1% deoxycholate, 1 mM EDTA, 10 mM Tris-HCl pH 8.1) and finally twice with TE buffer (10 mM Tris-HCl pH 8.1, 1 mM EDTA). Immuno-complexes were then eluted from the Protein-A beads with two times 250 μL elution buffer (SDS 1%, 0.1 M NaHCO$_3$, freshly prepared) and then, just like the total input samples, incubated overnight with 300 mM NaCl at 65 °C to reverse the crosslinks. The samples were then treated with 2 μg of Proteinase K for 2 h at 45 °C in 40 mM EDTA and 40 mM Tris-HCl (pH 6.5). DNA was extracted using phenol:chloroform:isoamyl alcohol (25:24:1), ethanol-precipitated using 20 μg of glycogen as a carrier, and resuspended in 30 μL of DNAse/RNAse-free water.

The immunoprecipitated chromatin was used to prepare sample libraries used for deep sequencing by Fasteris SA (Geneva, Switzerland). ChIP-Seq libraries were prepared using the DNA Sample Prep Kit (Illumina) following manufacturer instructions. Single-end run was performed on an Illumina Next-Generation DNA sequencing instrument (NextSeq High), 50 cycles (for *HA-uzcR*) and 75 cycles (*HA-chvI*), respectively, were performed and yielded several million reads per sequenced sample. The single-end sequence reads stored in FastQ files were mapped against the genome of *C. crescentus* NA1000 (NC_011916.1) using Bowtie2 Version 2.4.5+galaxy1 available on the web-based analysis platform Galaxy (https://usegalaxy.org) to generate the standard genomic position format files (BAM). ChIP-Seq reads sequencing and alignment statistics are summarized in Dataset EV2. Then, BAM files were imported into SeqMonk version 1.47.2 (http://www.bioinformatics.babraham.ac.uk/projects/seqmonk/) to build ChIP-Seq normalized sequence read profiles. Briefly, the genome was subdivided into 50 bp, and for every probe, we calculated the number of reads per probe as a function of the total number of reads (per million, using the Read Count Quantitation option). Analyzed data illustrated in Fig. 3A,B are provided in Table S2. Using the web-based analysis platform Galaxy (https://usegalaxy.org), HA-ChvI and 2xHA-UzcR ChIP-Seq peaks were called using MACS2 Version 2.2.7.1+galaxy0 (No broad regions option) relative to the total input DNA samples. The q-value (false discovery rate, FDR) cut-off for called peaks was 0.05. Peaks were rank-ordered according to their fold-enrichment values (Dataset EV2, Peaks with a fold-enrichment values >4 were retained for further analysis). Consensus sequences common to the top 130 HA-ChvI peaks and to the top 134 HA-UzcR peaks (identified in *yjbI*::*Tn* backgrounds) were respectively identified by scanning peak sequences (+ or −150 bp relative to the coordinates of the peak summit) for conserved motifs using MEME-ChIP (Version 5.5.0; http://meme-suite.org/)(Bailey et al, 2006). The data were deposited at GEO under accession number GSE309018.

### β-galactosidase assays

β-galactosidase assays were done using freshly electroporated strains carrying the pLac290 plasmid with transcriptional fusions between the indicated promoters and *lacZ*. Briefly, 100 μL of cells ($OD_{600}nm = 0.3$-0.7) were lysed in 30 μL of chloroform and vigorously mixed with Z buffer (60 mM $Na_2HPO_4$; 40 mM $NaH_2PO_4$; 10 mM KCl and 1 mM $MgSO_4$; pH 7) to obtain a final volume of 800 μL. To begin the reaction 200 μL of ONPG were added (o-nitrophenyl-b-D-galactopyranoside, at 4 mg/mL in 0.1 M potassium phosphate, pH 7). Assays were stopped with 500 μL of 1 M $Na_2CO_3$ when the solution turned light yellow. All assays were done at room temperature. The $OD_{420nm}$ of the supernatant and the $OD_{600nm}$ of the culture were collected and used to calculate the Miller units as follows: $U = (OD_{420}*1000)/(OD_{600}*t(min)*v(ml))$. The standard deviation is represented as error bars. All data are from a minimum of three biological replicates.

### Immunoblots

Strains were grown for 2 h at 30 °C under constant agitation up to $OD_{600nm}$ between 0.2 and 0.4, prior to the addition of the inducer, and the cells were grown for an additional 2 h. Samples were harvested by centrifugation and resuspended in 1× SDS sample buffer (50 mM Tris-HCl pH 6.8, 2,5% SDS, 10% glycerol, 1% β-mercaptoethanol, 12.5 mM EDTA, 0.05% bromophenol blue). Protein samples were analyzed on an SDS–polyacrylamide (37.5:1) gel electrophoresis and blotted on 0.45 μm pore PolyVinyliDenFluoride (PVDF) membranes (Immobilon-P from Sigma-Aldrich). Membranes were blocked 2 h with 1× Tris-buffered saline (TBS) (50 mM Tris-HCl, 150 mM NaCl [pH 8]) that contain 0.1% Tween-20 and 8% powdered milk. After 1 h of incubation, the primary antibodies were used overnight at 4 °C. The polyclonal antisera to AcrA (1:15,000), HA (1:20,000), and CCNA_00164 (1:20,000) were used (Ardissone et al, 2014; Vallet et al, 2020). The detection of primary antibodies was done using HRP-conjugated donkey anti-rabbit antibody (Jackson ImmunoResearch) with Western Blotting Detection System (Immobilon from Millipore). Images were acquired on a Bio-Rad illuminator (Chemidoc MP, Bio-Rad).

### Cell fractionation and LC-MS/MS analysis

Fifty mL of an exponential culture of *Caulobacter* at $OD_{600}nm = 0.4$ to 0.6 was harvested by centrifugation for 15 min at $8000 \times g$ at 4 °C. The cells were washed twice with 20 mL cold PBS, then resuspended in 1 mL resuspension buffer [20 mM Tris-HCl pH 7.5, 300 mM NaCl, 12500 U Ready-lyse (Epicentre Technologies), and one tablet of EDTA-free protease inhibitor cocktail]. After 5 min of incubation, 10 μL of DNase 1 μg/mL, 5 μL of RNaseA 20 μg/mL were added prior sonication in an ice-water bath (15 cycles 30 s ON; 30 s OFF). Sonicated samples are centrifuged for 40 min at $20,000 \times g$ at 4 °C, the supernatant was removed. The pellet was resuspended in 300 μL of 20 mM Tris-HCl pH 7.5, 300 mM NaCl, 2% Triton X-100, and incubated at RT for 20 min. Centrifugation was performed for 30 min at $20,000 \times g$ at 4 °C. The final pellet was solubilized in 2× SDS sample buffer and boiled for 10 min, then centrifuged for 2 min at $13,000 \times g$. The supernatant was harvested and sent for MS identification at the Taplin Mass Spectrometry Facility at the Harvard Medical School (Boston, USA).

### Northern blot analysis and RNA-seq

Northern blots were done using total RNA extracted from *C. crescentus WT* and mutants and probed for *chvR* as previously

described (Frohlich et al, 2018). For RNA-Seq, *C. crescentus WT* and Δ*yjbI* mutant cells were grown in triplicate in PYE medium to an $OD_{660}$ of 0.4. Bacterial samples were collected 30 min after incubation with/without 80 μM ZnSO4 in 0.2 volumes of stop-mix (95% ethanol and 5% phenol, vol/vol) and snap-frozen in liquid nitrogen. Total RNA was purified with the Hot-Phenol method followed by DNase treatment with the TurboDNAse (Thermofisher) and purification with Phenol:Chloroform:Isoamyl Alcohol (PCi). RNA integrity was confirmed using a Bioanalyzer (Agilent), and rRNA was depleted from the samples as described previously (Culviner et al, 2020). Purified mRNA was subjected to cDNA library preparation with the NEBNext® Ultra™ II Directional RNA Library Prep Kit for Illumina® (NEB, #E7760) according to the manufacturer's instructions, adjusting library size with AMPure XP beads (Beckmann Coulter). Libraries were pooled and sequenced on an Illumina NextSeq1000/2000 system (Illumina) in single-read mode (P200 cycles cartridge). Read files in FASTQ format were imported into CLC Genomics Workbench (version 22.0.2, Qiagen), trimmed, and mapped to the *Caulobacter crescentus* NA1000 reference genome (NC_011916.1) using the "RNA-Seq Analysis" tool with standard parameters. Read counts were normalized (CPM) and transformed (log2). Differential expression was tested using the built-in tool corresponding to edgeR in exact mode with tagwise dispersions. Genes with a fold change ≥2.0 and an FDR-adjusted $P$ value ≤ 0.05 were considered as differentially expressed. The RNA-Seq dataset has been deposited in GEO under accession GSE269833.

### Maleimide modification of surface-exposed cysteines

Two mL of exponential phase cultures of *C. crescentus* ($OD_{600}nm = 0.4$–0.6) were harvested by centrifugation for 10 min at $8000 \times g$ at 4 °C. Cells were washed once with soft buffer (20 mM Tris-HCL pH 7.5, 20 mM NaCl), prior to resuspension in 1 mL of soft buffer containing protease inhibitors (Roche). Next, DTT (dithiothreitol) was added when indicated to a final concentration of 20 mM and incubated for 10 min at room temperature. Following three washes with 1 mL in soft buffer, cells were resuspended in 1 mL of soft buffer containing protease inhibitors (Roche) supplemented with 500 μM methoxypolyethylene glycol maleimide (MGM) and incubated 20 min at room temperature. Cells were then harvested by centrifugation and resuspended in 150 μL of soft buffer, supplemented with 150 units of Ready-Lyse (LGC Biosearch Technologies), 1 μg DNAse, 1 mM $MgCl_2$, and incubated 2 min at room temperature. Finally, 50 μL of 4× SDS-PAGE sample buffer without reducing agent (250 mM Tris-HCl, pH 6.8, 10% SDS, 25% glycerol, bromophenol blue) was added, and samples were analyzed by SDS-PAGE and immunoblotting using monoclonal antibodies to HA. Control samples were boiled in regular (reducing) 1× SDS sample buffer (50 mM Tris-HCl pH 6.8, 2.5% SDS, 10% glycerol, 1% β-mercaptoethanol, 12.5 mM EDTA, 0.05% bromophenol blue) to show the loss of MGM modification on YjbI-HA by immunoblotting using monoclonal antibodies to HA.

### Antibiotic library screening

Two libraries from MedChemExpress (HY-L067 and HL-L033) were purchased and stored at −80 °C in 96-well plates. The compounds were diluted to a final concentration of 1 mM in DMSO, water, or ethanol according to manufacturer instructions in

new 96-well plates. For each 96-well plate and each strain tested, a volume of 50 mL of PYE 1.5% agar was poured into two 14-cm round Petri dishes. After polymerization, a top layer of 12 mL PYE soft (0.3%) agar mixed with 400 μL of overnight culture of the strain of interest was poured on each Petri dish. Using a multichannel pipette, 3 μL of each compound from odd-numbered columns was dropped on one plate, and from even-numbered columns on the other one (to minimize chemical interactions between molecules). After 24 h of incubation at 30 °C, pictures (with a reference scale) were taken using Bio-Rad illuminator with the Image Lab 4.1 software. The area of inhibition was measured using ImageJ software. The presented results correspond to the mean of the area from two independent replicates.

### Atomistic and CG MD simulations

We used the CHARMM-GUI web server (Jo et al, 2008) to prepare the systems and input files for the atomistic MD simulation; we specifically used the CHARMM-GUI Membrane Builder tool (Jo et al, 2009) with the YjbI model as input. We used the Positioning of Proteins in Membrane (PPM) tool from the Orientation of Proteins and Membranes (OPM) database (Lomize et al, 2012) to place YjbI on a hypothetical membrane layer. We generated a tetragonal simulation box with YjbI and an asymmetric lipid bilayer with an OM composition of 100% LPS with Lipid A type 2 from *Salmonella enterica*, with Core R1 and O-antigen 1,3,19, the closest choice CHARMM-GUI known for the molecular structure of *C. crescentus* LPS (Jones et al, 2015). For the inner leaflet, we built a layer with 90% 1,2-dioleoyl-sn-glycero-3-phosphoglycerol (DOPG) and 10% cardiolipin (charge -2, TOCL2). The resulting membrane had an area of $119 \times 119$ Å$^2$. We solvated the simulation box with TIP3P water and neutralized the charges of the system with counterions corresponding to a 1 mM CaCl$_2$ solution. We then generated all files necessary to run the simulations using the CHARMM36m forcefield (Huang et al, 2017). GROMACS (Bekker et al, 1993) version 2024.3 was then used to compute the MD simulations. We set the cut-off radius for short-range electrostatic and van der Waals interactions at 12 Å; we used the particle mesh Ewald algorithm (PME) (Darden et al, 1993) to calculate long-range electrostatic interactions. We restrained the length of covalent bonds involving hydrogen atoms using the LINCS algorithm (Hess et al, 1997). We applied periodic boundary conditions in all directions.

In the first stage of the simulation protocol, we minimized the potential energy of the system using the steepest descent algorithm, setting a threshold of a maximum 1000 kJ mol$^{-1}$ nm$^{-1}$ of force acting on the atoms as a convergence criterion. We then subjected the system to thermal equilibration in the canonical ensemble (NVT)(Nosé, 1984a). Initial atomic velocities were determined following a Maxwell-Boltzmann distribution corresponding to a temperature of 0.03 K. The system was linearly heated during 10 ns up to 300 K, using simulated annealing, and we equilibrated the system under these conditions for a further 10 ns with a 1-femtosecond time step using the Berendsen thermostat. During this procedure, we restrained protein heavy-atom positions using a harmonic potential, with force constants of 4000 (backbone) and 2000 (sidechain) kJ mol$^{-1}$ nm$^{-2}$, respectively. The phosphorus atoms of the lipid molecules were also restrained on the Z-axis (direction of the normal vector at the membrane surface; only one

of the phosphorus atoms, in the case of Lipid A), as well as the dihedral angles of the double bonds in the fatty acid chains, with force constants of 1000 kJ mol$^{-1}$ nm$^{-2}$ and 1000 kJ mol$^{-1}$ rad$^{-2}$, respectively.

We then proceeded to stabilize the pressure and density of the system in the isothermal-isobaric ensemble (NPT)(Andersen, 1980). We kept the pressure at 1 bar using the Berendsen barostat with semi-isotropic pressure coupling and a 5 ps time constant. We equilibrated the system under these conditions for 35 ns, and the position restraints were gradually removed. We launched the production dynamics in the NPT ensemble, changing the thermostat to Nosé-Hoover (Hoover, 1985; Nosé, 1984b) and the barostat to Parrinello-Rahman (Parrinello and Rahman, 1981). This stage lasted 2 μs.

For CG simulations, we used the martinize2 tool (Kroon et al, 2025) to obtain a CG representation of YjbI using its structural model as input. We generated an elastic bond network between backbone beads to keep the tertiary structure of the CG representation of YjbI, with a force constant of 500 kJ mol$^{-1}$ nm$^{-2}$. The upper cutoff for creating bonds between beads was 9 Å. We did not impose elastic networks on the C-terminus loop of YjbI, leaving it flexible (residues 256–294).

We generated the simulation system with the Insane Python script (19). We created a symmetric lipid bilayer composed of 90% 1-palmitoyl-2-oleoyl-sn-*glycero*-3-phosphocholine (POPC) and 10% cardiolipin (charge -2, CDL2). YjbI was positioned in a random orientation with respect to the membrane, and its center of mass (COM) was shifted 7 nm away from that of the membrane, effectively placing the protein in bulk solvent. We neutralized the negative charges of the system with Na$^+$ counterions. GROMACS 2024.3 was used with the MARTINI 3 forcefield (20). We set the cut-off radius for short-range electrostatic and van der Waals interactions at 11 Å; Coulomb interactions were corrected with a reaction-field of dielectric constant 15. We also applied periodic boundary conditions in all directions. We minimized the potential energy of the system in two steps using the steepest descent algorithm. In the first step, we used a soft-core potential for non-bonded interactions (with alpha set to 4 and power to 2) for 5000 steps. This was done to avoid infinite energies arising from possible overlaps between MARTINI beads after building the system. We then proceeded with a further 5000 minimization steps with a normal hard-core potential. We equilibrated the system in the NPT ensemble for 162.50 ns, while gradually removing position restraints from the protein backbone beads (harmonic force constant set to 400 kJ mol$^{-1}$ nm$^{-2}$) and the lipid head group beads (harmonic force constant set to 100 kJ mol$^{-1}$ nm$^{-2}$). Atomic velocities were determined following a Maxwell-Boltzmann distribution corresponding to a temperature of 300 K, and the temperature was maintained with a V-rescale thermostat. We stabilized the pressure at 1 bar using the Berendsen barostat with semi-isotropic pressure coupling and a 5 ps time constant.

We launched production dynamics in the NPT ensemble, keeping the thermostat and changing the barostat to Parrinello-Rahman. This stage lasted 4 μs. We ran the entire procedure in five replicates. We performed all lipid interaction analyses of the CG simulations with the Python package PyLipID (21), with the distance cutoff for lipid interactions set to 0.6 nm. We used the Visual Molecular Dynamics (VMD) (22) software for visual inspection of the simulations and recording of the movies. All protein and membrane images were generated with ChimeraX (23).

The water beads count distribution plot was generated with the Python programming language in Visual Studio Code.

## Data availability

RNA-Seq and ChIP-Seq data is deposited in GEO under accession GSE269833 and GSE309018, respectively.

The source data of this paper are collected in the following database record: biostudies:S-SCDT-10_1038-S44318-025-00668-x.

## Peer review information

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

## Acknowledgements

We thank Yves Mattenberger for valuable intellectual input and for constructing pSRK-*fhuA*$^{hyp}$. We also thank Sean Crosson, Clare Kirkpatrick, Dan Park, and Yongqin Jiao for strains, materials and/or discussions. We also thank Klaas Martinus Pos for providing *E. coli* K12 BW25113 Δ*tolC*. This work was funded by grants from the Swiss National Science Foundation (CRSII5_198737 to MDP and PHV; 310030_219565 to PHV), the Canton de Genève, and the Deutsche Forschungsgemeinschaft (FR 3502/2-1 to KSF). JC owes special thanks to Sabine Quindou, Jamy Gourmaud, and Frédéric Courant for nurturing an entire generation's curiosity.

## Author contributions

**Jordan Costafrolaz**: Conceptualization; Validation; Investigation; Visualization; Methodology; Writing—original draft; Writing—review and editing. **Laurence Degeorges**: Formal analysis; Investigation; Methodology. **Gaël Panis**: Resources; Formal analysis; Investigation; Methodology. **Simon-Ulysse Vallet**: Formal analysis; Investigation; Methodology. **Manuel Velasco Gomariz**: Formal analysis; Investigation; Methodology; Writing—review and editing. **Fernando Teixeira Pinto Meireles**: Investigation; Methodology; Writing—review and editing. **Matteo Dal Peraro**: Formal analysis; Funding acquisition; Methodology; Writing—review and editing. **Kathrin S Fröhlich**: Funding acquisition; Investigation; Methodology; Writing—original draft; Project administration; Writing—review and editing. **Patrick H Viollier**: Conceptualization; Formal analysis; Funding acquisition; Investigation; Writing—original draft; Writing—review and editing.

Source data underlying figure panels in this paper may have individual authorship assigned. Where available, figure panel/source data authorship is listed in the following database record: biostudies:S-SCDT-10_1038-S44318-025-00668-x.

## Disclosure and competing interests statement

The authors declare no competing interests.

# Expanded View Figures

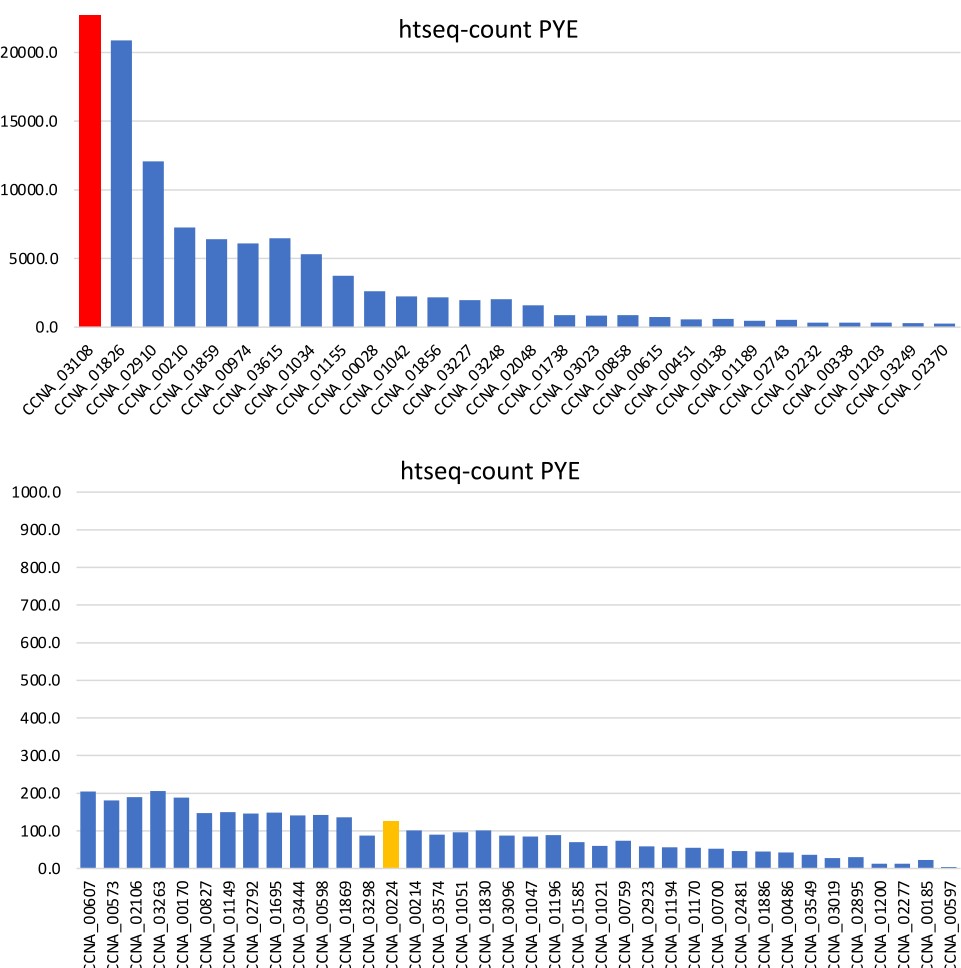

**Figure EV1. Expression of TBDRs as determined by RNA-Seq.**

Graphical representation of the htseq transcript count of RNA-Seq data from (Siwach et al, 2021) showing the expression profiles of all annotated TBDR genes from *Caulobcater crescentus* NA1000 grown in PYE. Note that the *chvT* transcript (*CCNA_03108*, in red) and the *bugA* transcript (*CCNA_00224* in yellow) are at opposite ends of the abundance spectrum of TBDR-encoding transcripts, with *chvT* being the most abundant, whilst the *bugA* transcript is among the low abundance transcripts.

## A

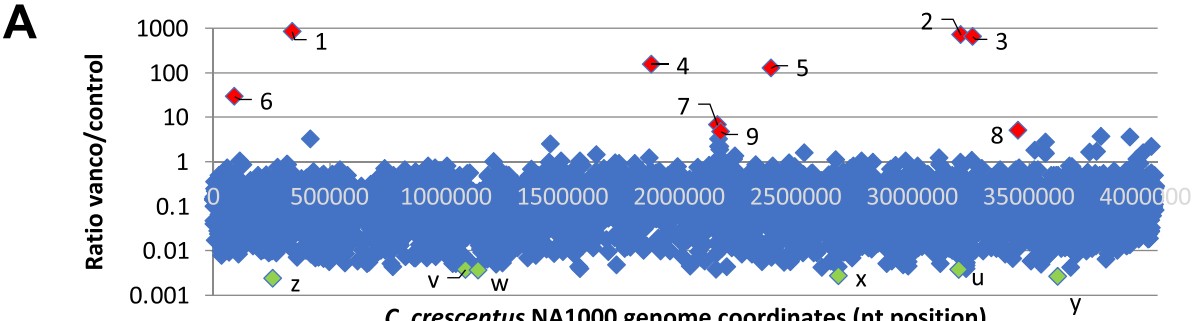

## B

| # | ID (feature) | Description | Ratio |
|---|---|---|---|
| 1 | CCNA_00324 (exbD) | ExbD-family protein | 832.31 |
| 2 | CCNA_03052 | acyltransferase | 702.05 |
| 3 | CCNA_03108 (chvT) | TonB-dependent outer membrane receptor | 643.50 |
| 4 | CCNA_01751 (mltG) | endolytic murein transglycosylase | 156.10 |
| 5 | CCNA_02243 (crbA) | D-alanyl-D-alanine serine-type carboxypeptidase | 125.41 |
| 6 | CCNA_00086 (gdhZ) | NAD-specific glutamate dehydrogenase | 28.71 |
| 7 | CCNA_02016 (nuoM) | NADH-quinone oxidoreductase chain M | 6.64 |
| 8 | CCNA_03281 | Lrp-family transcriptional regulator | 5.03 |
| 9 | CCNA_02033 (nuoA) | NADH-quinone oxidoreductase chain A | 4.70 |
| u | CCNA_03043 (pilA) | type IV pilin protein | 0.00377 |
| v | CCNA_00998 (mucR2) | ROS/MUCR transcriptional regulator | 0.00372 |
| w | CCNA_01041 | hypothetical protein | 0.00367 |
| x | CCNA_02533 | hypothetical protein | 0.00266 |
| y | CCNA_03992 | UDP-glucose 6-dehydrogenase-related protein | 0.00261 |
| z | CCNA_03917 | hypothetical protein | 0.00235 |

## C

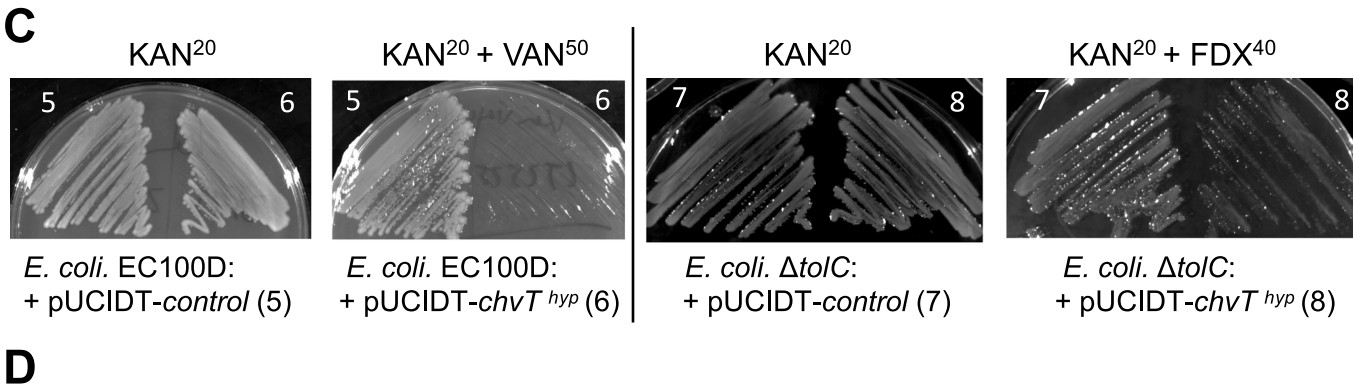

## D

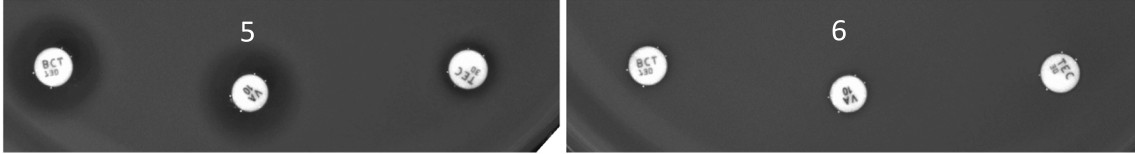

◀  **Figure EV2.  Vancomycin resistance determinants identified by Tn-Seq.**

(A) Plot representation of the Tn-Seq on PYE plates containing 20 µg/mL of vancomycin (VAN[20]). (B) Table showing the top nine genes ("1–9") whose disruption favors growth on plates with VAN[20] and are thus overrepresented as Tn insertion sites, while the six lowest (underrepresented genes) are also shown in the table indicated as positions "u–z". (C) Growth of *E. coli* EC100D or efflux-defective Δ*tolC* cells containing pUCIDT- control or pUCIDT-*chvT* [hyp] plasmids streaked on LB-plates containing 20 µg/mL of kanamycin (KAN[20]) with or without or 50 µg/mL of vancomycin (VAN[50]) or 40 µg/mL fidaxomicin (FDX[40]). Since FDX is a substrate of the *E. coli* AcrAB-TolC multidrug efflux pump, the permeability of FDX conferred by ChvT[hyp] expression is best detectable in Δ*tolC* cells. (D) Growth inhibition assays by antibiotics on discs placed on *E. coli* EC100D cells embedded on LB-soft agar on LB plates. Discs from left to right containing 130 units bacitracin (BCT130), 10 µg/mL of vancomycin (VA10) or 30 µg/mL of teicoplanin (TEC30). Plates were incubated for 2 days at 30 °C.

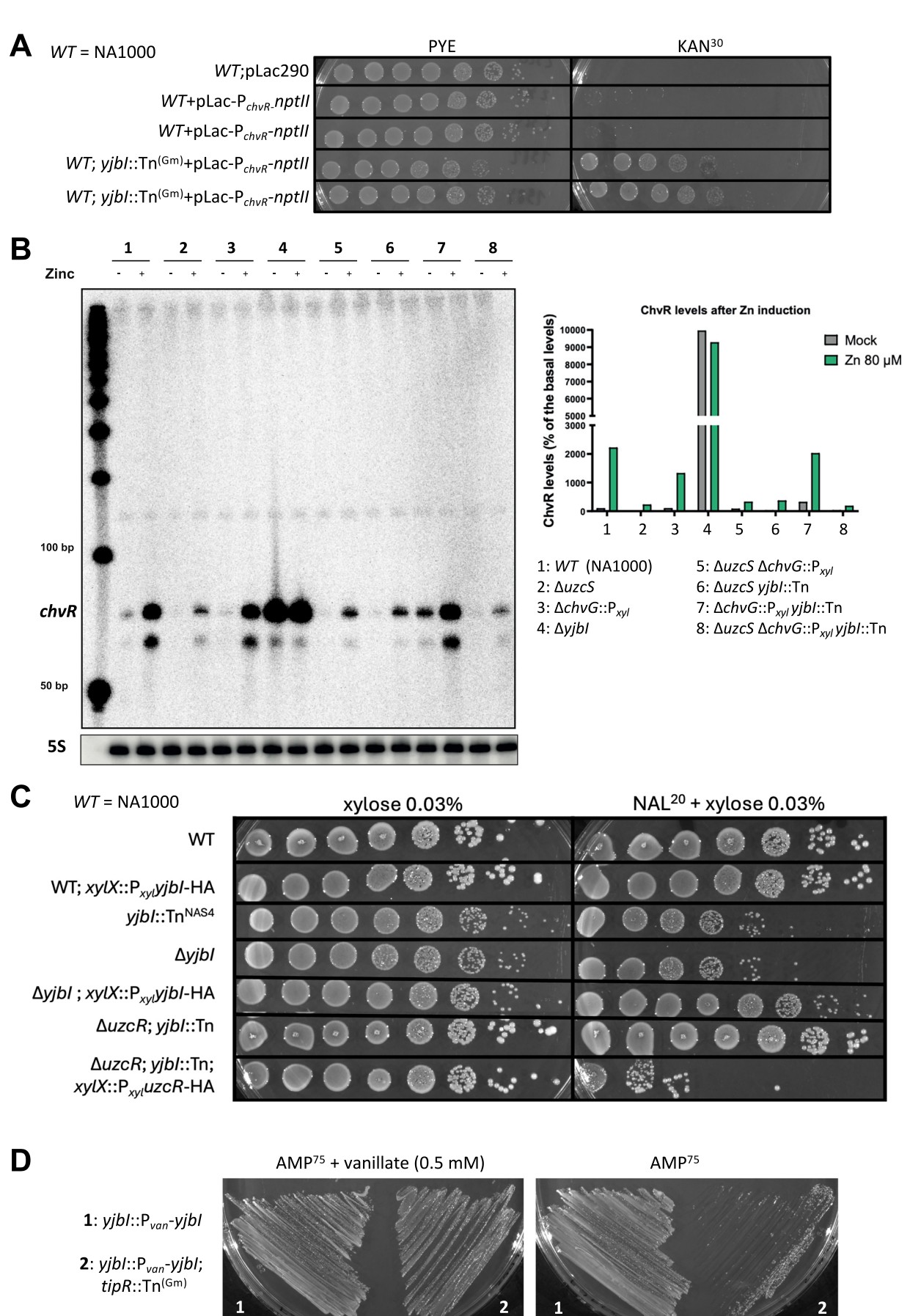

◀ **Figure EV3. Phenotype of cells lacking YjbI.**

(A) EOP assay of strains carrying the pLac290 empty vector or the pLac290 derivative harboring the P$_{chvR}$ - *nptII* reporter plated PYE with or without 30 µg/mL kanamycin (KAN[30]). Plates were incubated for 2 days at 30 °C. (B) Northern blots are shown as described in the inset of Fig. 1D. This figure shows the full blot and size markers. The graph on the right is a quantification of the transcripts detected by Northern blot. (C) EOP assay of *yjbI* mutant cells complemented with YjbI-HA or *uzcR* mutant complemented with HA-UzcR on PYE containing or not 20 µg/mL of nalidixic acid (NAL[20]). Both plates contain 0.03% of xylose and were incubated for 48 h at 30 °C. (D) Growth of *yjbI*::P$_{van}$-*yjbI* cells with or without *tipR*::Tn on plates containing 75 µg/mL of ampicillin (AMP75) with or without 0.5 mM vanillate. Plates were incubated for 60 h at 30 °C.

**A** P*acrA*-*lacZ* activity 2 hours after NAL induction

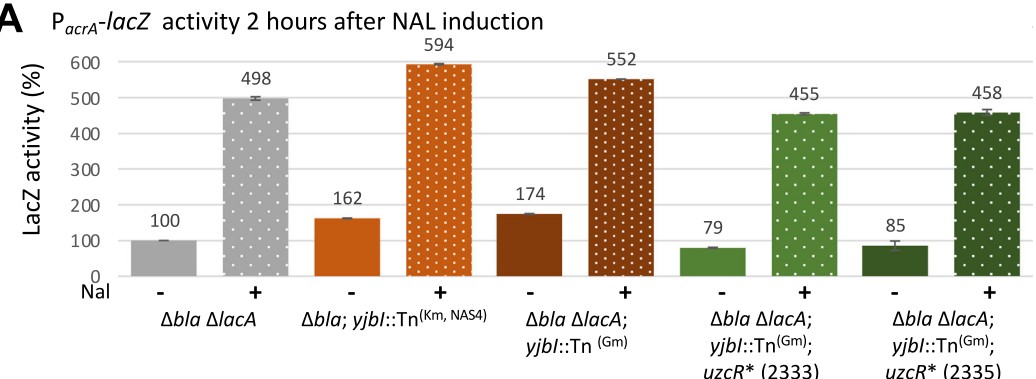

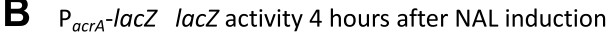

**B** P*acrA*-*lacZ* *lacZ* activity 4 hours after NAL induction

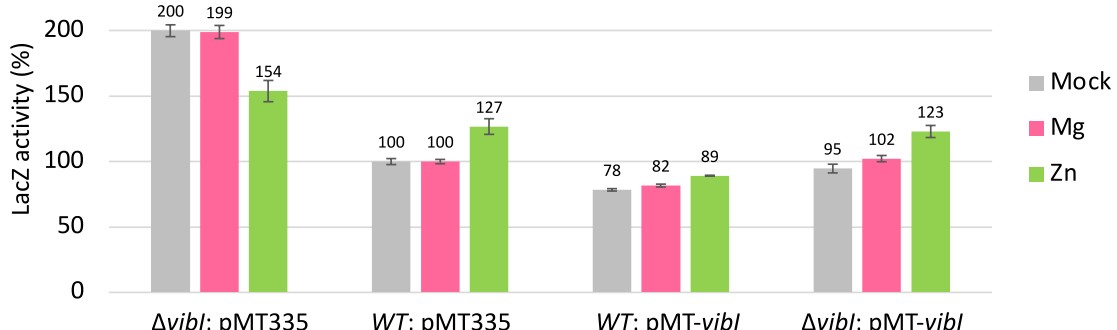

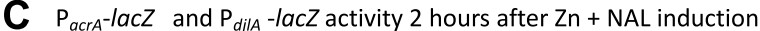

**C** P*acrA*-*lacZ* and P*djlA*-*lacZ* activity 2 hours after Zn + NAL induction

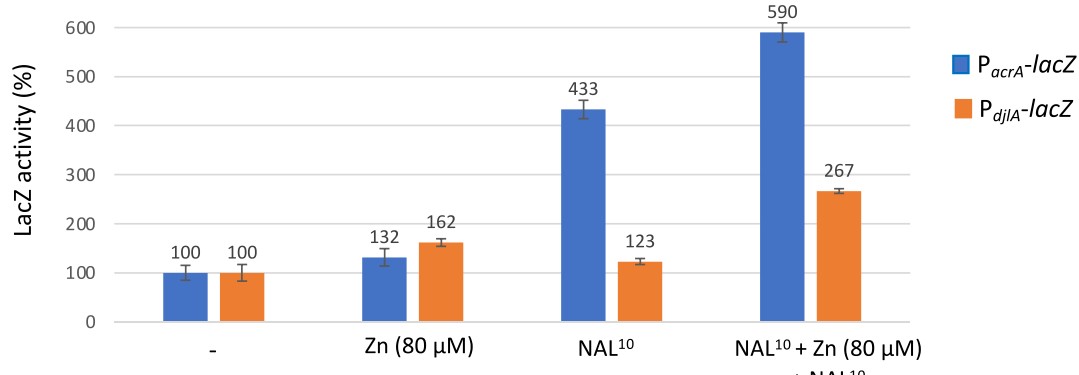

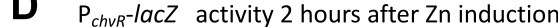

**D** P*chvR*-*lacZ* activity 2 hours after Zn induction

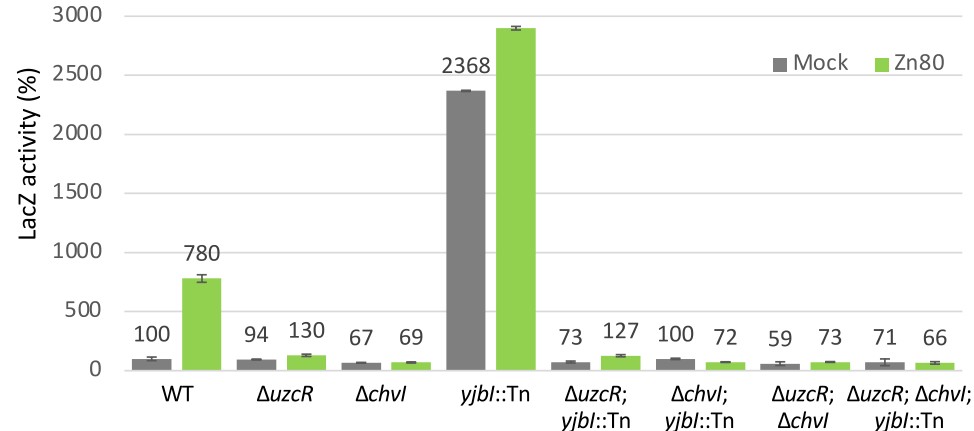

◀ **Figure EV4. NAL- and Zn induction in cells lacking YjbI.**

(A–D) β-galactosidase assay using the pP$_{acrA}$-*lacZ* (**A–C**), pP$_{djlA}$-*lacZ* (**C**) and pP$_{chvR}$-*lacZ* (**D**) promoter probe plasmids in various mutants performed for two (**A, B, D**) or 4 h (**C**) with MgSO$_4$, ZnSO$_4$ (80 μM) or NAL[10] (+) at 30 °C in PYE. All levels are indicated in percentage of expression relative to the basal level of the *WT* in the uninduced condition. Error bars are defined as +/− standard deviation. All experiments done with strains carrying the pMT335 (and derivatives) were done in the presence of vanillate (100 μM).

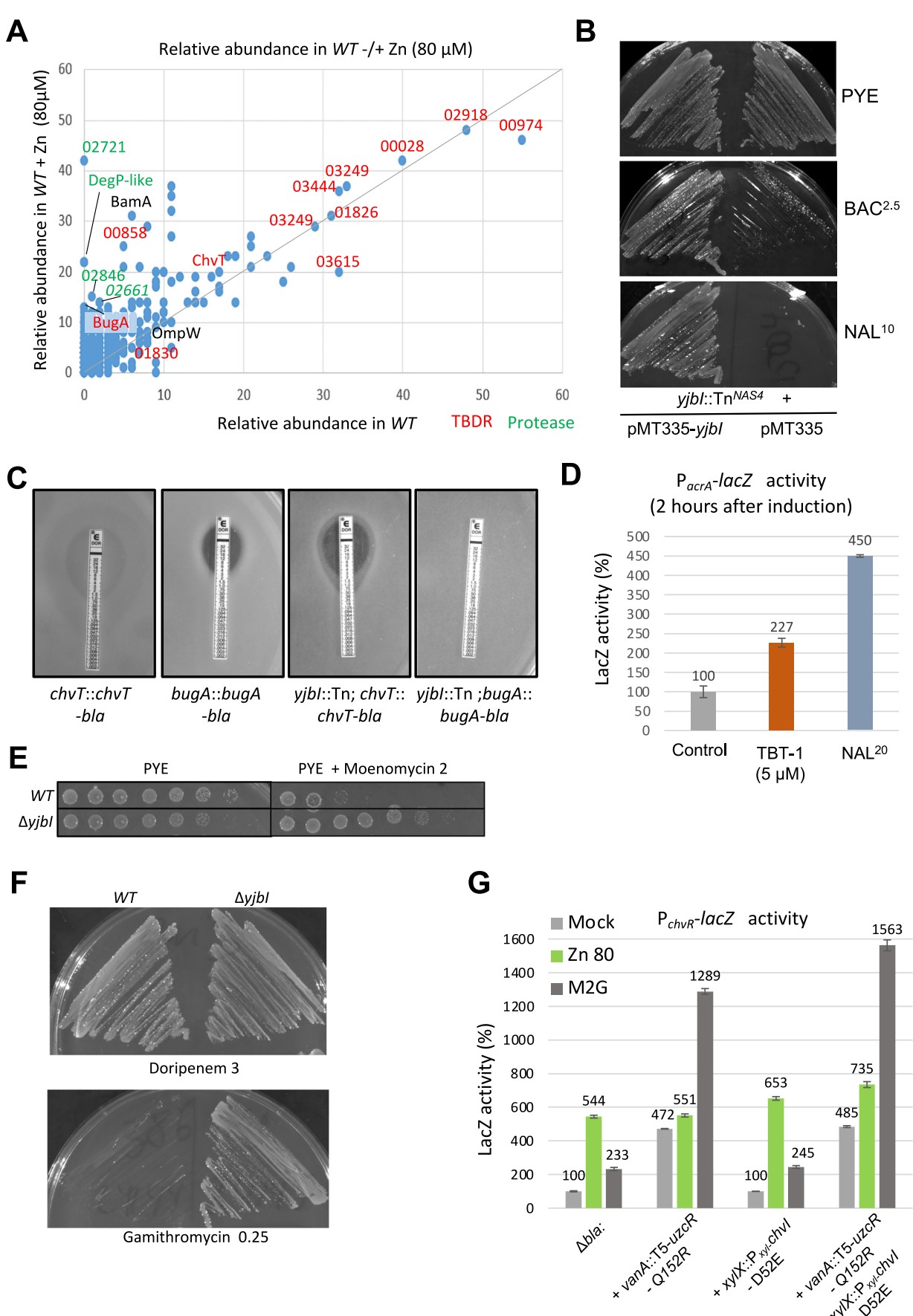

◄ **Figure EV5. OM proteome, ChvT and BugA expression in *WT* and *yjbI* mutant cells.**

(A) Scatter plot graphic representing the relative abundancy of proteins from outer membrane enriched samples of WT cells after 2 h of growth with or without ZnSO$_4$ (80 μM). The TBDR are indicated in red while the predicted proteases are in green. (B) Complementation tests of *yjbI*::Tn$^{NAS4}$ mutants harboring pMT335 or pMT335-*yjbI* streaked on PYE plates, plates with NAL[10] or plates with BAC[2.5]. (C) Doripenem MIC estimation by E-test, a strip containing various concentrations of doripenem. The E-test was placed on soft agar seeded with Δ*bla chvT::chvT-bla* and Δ*bla bugA::bugA-bla* cells with or without the *yjbI*::Tn$^{NAS4}$ mutation. The plates were incubated for 24 h at 30 °C. (D) β-galactosidase assay using the pP$_{acrA}$-*lacZ* promoter probe plasmid in *WT* cells following 2 h of induction with TBT-1 or with NAL at 30 °C in PYE. Error bars are defined as +/- standard deviation. (E) EOP assay of *WT* and Δ*yjbI* cells on PYE plates with or without moenomycin (2 μg/mL). The plates were incubated for 2 days at 30 °C. (F) *WT* and Δ*yjbI* cells were streaked on agar plates containing either doripenem (3 μg/mL) or gamithromycin (0.25 μg/mL) and grown for 2 days at 30 °C. (G) β-galactosidase assay using the pP$_{chvR}$-*lacZ* promoter probe plasmid in cells expressing UzR-Q152R from the strong *E. coli* T5 promoter at the *vanA* locus or ChvI-D52E from the P$_{xyl}$ promoter at the *xylX* locus. Cells were grown in PYE containing 0.3% xylose, exposed to ZnSO$_4$ (80 μM) for 2 h or, switched and grown in M2G minimal medium for 5 h before measurements. Error bars are defined as +/− standard deviation.

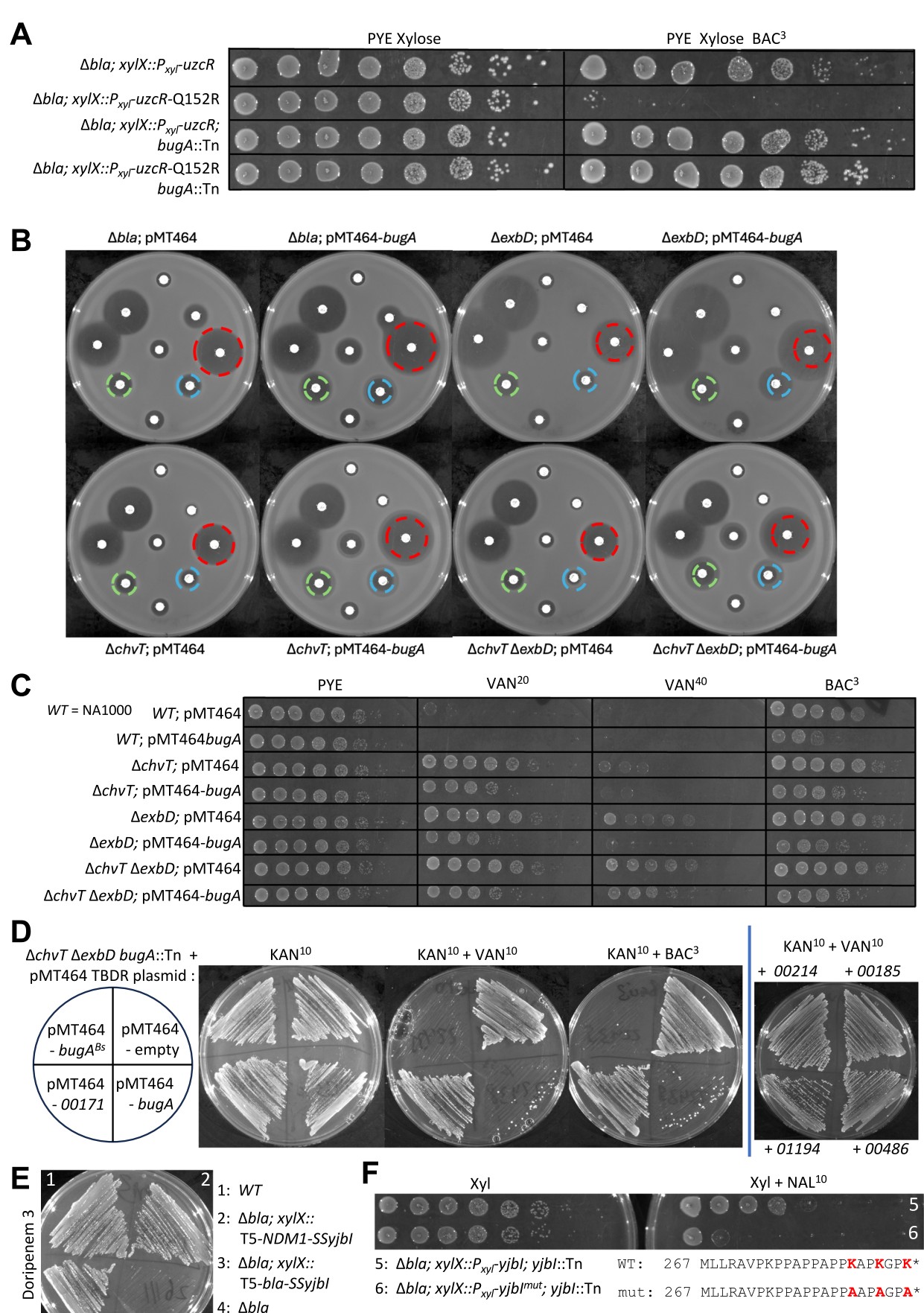

**A**

Δ*bla*; *xylX::P*$_{xyl}$*-uzcR*

Δ*bla*; *xylX::P*$_{xyl}$*-uzcR*-Q152R

Δ*bla*; *xylX::P*$_{xyl}$*-uzcR*; *bugA*::Tn

Δ*bla*; *xylX::P*$_{xyl}$*-uzcR*-Q152R *bugA*::Tn

PYE Xylose

PYE Xylose BAC[3]

**B**

Δ*bla*; pMT464  Δ*bla*; pMT464-*bugA*  Δ*exbD*; pMT464  Δ*exbD*; pMT464-*bugA*

Δ*chvT*; pMT464  Δ*chvT*; pMT464-*bugA*  Δ*chvT* Δ*exbD*; pMT464  Δ*chvT* Δ*exbD*; pMT464-*bugA*

**C**

*WT* = NA1000

*WT*; pMT464

*WT*; pMT464*bugA*

Δ*chvT*; pMT464

Δ*chvT*; pMT464-*bugA*

Δ*exbD*; pMT464

Δ*exbD*; pMT464-*bugA*

Δ*chvT* Δ*exbD*; pMT464

Δ*chvT* Δ*exbD*; pMT464-*bugA*

PYE  VAN[20]  VAN[40]  BAC[3]

**D**

Δ*chvT* Δ*exbD* *bugA*::Tn + pMT464 TBDR plasmid :

| pMT464 - *bugA*[Bs] | pMT464 - empty |
| pMT464 - 00171 | pMT464 - *bugA* |

KAN[10]  KAN[10] + VAN[10]  KAN[10] + BAC[3]

KAN[10] + VAN[10]
+ 00214   + 00185

+ 01194   + 00486

**E**

Doripenem 3

| 1 | 2 |
| 3 | 4 |

1: *WT*
2: Δ*bla*; *xylX::T5-NDM1-SSyjbI*
3: Δ*bla*; *xylX::T5-bla-SSyjbI*
4: Δ*bla*

**F**

Xyl  Xyl + NAL[10]

5
6

5: Δ*bla*; *xylX::P*$_{xyl}$*-yjbI*; *yjbI*::Tn
6: Δ*bla*; *xylX::P*$_{xyl}$*-yjbI*$^{mut}$; *yjbI*::Tn

WT:  267 MLLRAVPKPPAPPAPP**K**AP**K**GP**K**\*
mut: 267 MLLRAVPKPPAPPAPP**A**AP**A**GP**A**\*

◀

**Figure EV6.  Antibiotic sensitivity analysis of cells expressing TBDRs.**

(A) EOP assay of Δ*bla* cells harboring *xyl*::P$_{xyl}$-*uzcR* or *xyl*::P$_{xyl}$-*uzcR-Q152R* with or without the *bugA*::Tn mutation. Cells were plated on PYE plates containing 0.3% xylose, with or without BAC3 and the plates were incubated for 2 days at 30 °C. (B) Kirby-Bauer type antibiograms of various *C. crescentus* strains containing pMT464-*bugA* or the empty vector. The indicator strains were embedded in PYE soft agar and antibiotic discs were placed on soft agar from top left to bottom right: colistin (50 µg), fosfomycin (50 µg), piperacillin (100 µg), rifampicin (30 µg), moenomycin (5 µg), bacitracin (130 µg), teicoplanin (30 µg), vancomycin (30 µg) and ramoplanin (40 µg). Plates were incubated for 24 h at 30 °C. (C) EOP assay of *C. crescentus* strains harboring pMT464 or pMT464-*bugA* diluted on PYE plates with or without VAN$^{20}$, VAN$^{40}$ or BAC$^3$. Plates were incubated for 2 days at 30 °C. (D) *C. crescentus* Δ*chvT* Δ*exbD bugA*::Tn cells harboring pMT464, pMT464-*bugA*, pMT464-*bugA*$^{Bs}$ or other TBDR expression plasmids including pMT464-*CCNA_00171*, pMT464-*CCNA_00185*, pMT464-*CCNA_00214*, pMT464-*CCNA_00486* or pMT464-*CCNA_01194* were plated on PYE with KAN$^{10}$, KAN$^{10}$ + VAN$^{10}$ or KAN$^{10}$ + BAC$^3$. Plates were incubated for 3 days at 30 °C. (E) *WT*, Δ*bla* and derivatives expressing either an NDM1 or a Bla fusion proteins harboring the N-terminal signal sequence of YjbI in lieu of their native signal sequences, were streaked on agar plates containing doripenem (3 µg/mL) and grown for 48 h at 30 °C. The fusion proteins were expressed from the *xylX* locus using the *E. coli* phage T5 promoter. (F) Complementation analysis of *WT* and mutant (*mut*) YjbI expressed from P$_{xyl}$ at the *xylX* locus in *yjbI*::Tn cells. Cells were streaked on plates containing 0.03% xylose (Xyl) and bacitracin (2.5 µg/mL, BAC$^{2.5}$) or nalidixic acid (10 µg/mL, NAL$^{10}$). Pates were incubated for 3 days at 30 °C. The mutant protein harbors the K284A/K287A double mutation and lacks the terminal lysine.

