## [Peer Review File · The EMBO Journal]

Asymmetric surface disposition of β -helix protein Yjbl controls bimodal antibiotic susceptibilities

Jordan Costafrolaz, Laurence Degeorges, Gael Panis, Simon-Ulysse Vallet, Manuel Velasco-Gomariz, Fernando Meireles, Matteo Dal Peraro, Kathrin Froehlich, and Patrick Viollier

Corresponding author: Patrick Viollier (patrick.viollier@unige.ch)

Review Timeline:

Submission Date:	25th Jul 24
Editorial Decision:	1st Oct 24
Revision Received:	7th Sep 25
Editorial Decision:	13th Nov 25
Revision Received:	25th Nov 25
Accepted:	7th Dec 25

Editor: Ieva Gailite

Transaction Report:

Dear Patrick,

Thank you for submitting your manuscript for consideration by the EMBO Journal. We have now received comments from three reviewers, which are included below for your information.

As you can see, reviewers find the presented role of Yjbl in regulation of antibiotic sensitivity per se of interest. However, the reviewers, and in particular reviewer #3, also raise a range of concerns regarding the experimental approach, the lack of appropriate controls and the presentation of the results that would need to be addressed before they can support publication. Additionally, reviewers #1 and #2 ask for further analysis of Yjbl mode of action (reviewer #1, points 1 and 2, reviewer #2, point #1), while reviewer #2 indicates that the potential conservation of Yjbl function in other Gram-negative bacteria remains currently unclear. While the reviewer proposes to address this in the discussion, I think that further experimental exploration of this question would significantly increase the impact of the study. Finally, reviewers #1 and #3 indicate that the textual clarity and the logical flow of the manuscript should be improved.

Based on the general interest expressed by the reviewers, I therefore invite you to address these concerns in a revised manuscript. I think that it would be useful to discuss the revision in more detail via email or phone/videoconferencing - please let me know which option you prefer.

We generally allow three months as standard revision time. Should you foresee a problem in meeting this deadline, please let us know in advance to discuss an extension. As a matter of policy, competing manuscripts published during this period will not negatively impact on our assessment of the conceptual advance presented by your study. However, please contact me as soon as possible upon publication of any related work to discuss the appropriate course of action.

When preparing your letter of response to the referees' comments, please bear in mind that this will form part of the Review Process File and will therefore be available online to the community. For more details on our Transparent Editorial Process, please visit our website: <https://www.embopress.org/page/journal/14602075/authorguide#transparentprocess>. Please also see the attached instructions for further guidelines on preparation of the revised manuscript.

Please feel free to contact me if have any further questions regarding the revision. Thank you for the opportunity to consider your work for publication, and I look forward to discussing your revision with you.

With best regards,

Ieva

Ieva Gailite, PhD
Senior Scientific Editor
The EMBO Journal
Meyerohofstrasse 1
D-69117 Heidelberg
Tel: +4962218891309
i.gailite@embojournal.org

We realize that it is difficult to revise to a specific deadline. In the interest of protecting the conceptual advance provided by the work, we recommend a revision within 3 months (30th Dec 2024). Please discuss the revision progress ahead of this time with the editor if you require more time to complete the revisions.

Referee #1:

This manuscript presents several intertwined stories that complicate readability. It is difficult for the reader to follow a cohesive narrative, and I would recommend the authors streamline the content. Parts of the manuscript, for instance the section on ChvT and the hyperpore, could be omitted for clarity. Simplifying the presentation will help the reader focus on the key findings. That said, the experimental work is solid, and the genetics experiments are particularly well-executed. The wealth of information in this study is impressive, and the findings are both important and suitable for publication in a high-impact journal like EMBO. With some refinement in the presentation, this manuscript would make a strong contribution to the field.

Specific Comments:

1. Cysteines near the Yjbl extremities: The flanking pairs of cysteines at the extremities of Yjbl are intriguing, particularly given that they remain in a reduced state despite being in the oxidizing environment of the periplasm. This raises important questions. How are these cysteines protected from oxidation? Addressing (or at least discussing) this question is relevant, especially because the authors use these cysteines to assess the surface exposure of the protein. Moreover, Yjbl's role in surface interactions between cells suggests potential surface exposure, yet this hypothesis was not thoroughly investigated. Also, why, in Figure 1A, is Yjbl depicted as oriented towards the periplasm? This contradicts the idea of surface exposure and should be clarified.
2. Use of OM-impermeable probes: The authors use MGM to assess cysteine accessibility on the surface. However, the claim that MGM cannot cross the outer membrane (OM) needs further validation. It would be prudent to include control experiments using well-characterized periplasmic proteins to ensure that MGM is indeed impermeable to the OM under the experimental conditions.
3. Clarification on Line 319: The authors mention that Zn induction is affected "in both directions." Could the authors clarify what is being induced? More detail is needed here to avoid ambiguity.
4. Clarification on Line 332: The sudden mention of potential Yjbl degradation is not well-introduced. The manuscript should provide more context on why degradation of Yjbl would occur and by what mechanisms. A more detailed introduction to this idea earlier in the manuscript would help the reader follow the logic.
5. Line 286-289: Fig2C, the authors mention a decrease of 76% (for Uzcs/yjbl) and 91% (for chvg/yjbl) compared to WT. I find this confusing, as the actual decrease appears minor, such as 9% for the chvg/yjbl mutant, for instance. Additionally, these differences are on comparisons between the mock WT condition and Zinc addition for the mutants; is this comparable? Finally, are these differences significant?

6. Line 444: Why BugA is no longer detected in the ChvG/yjbl mutant? Does chvG/chvI affect bugA transcription? How?

Minor comments/typos:

Line 409: Is Novobiocin an LptB activator?

Line 126: regulate coordinate...

Line 128: « that » twice

Line 192: « Garm-negative" instead of Gram-negative

Line 212: why is figure 3 mentioned before figure 2?

Line 106: Fig 1a is missing an arrow for inducing DjIA by UzcR, correct?

Line 317: "Prom" instead of from

Line 333: "and" mentioned twice

Line 345: missing a verb

Line 423: remove "above"

Line 424: 'decrease" in italic?

Line 426: Lack of a space "...µg/mLof"....

Line 430: BAC2 or BAC3 ?

Line 496: ope and/or a sbottleneck in folding/in

Line 500: we noteds

Line 539: may repoesent a common

Line 542: that are import substrates of

Line 543: be likened to a narrow-spectrum

Line 549: The current challenge is also understanding TBDR expression is regulated and enlarging our repertoire of how chemical

Referee #2:

Costafrolaz et al investigate the role of protein Yjbl in *C. crescentus* antibiotic sensitivity. Through extensive genetic analysis they provide evidence of pathways through yjbl that influence two envelope stress responses that affect outer membrane protein composition and antibiotic sensitivity. The data largely supports the authors' conclusions. A few clarifications would help interpretation of the data.

1. Is general membrane integrity influenced by deletion of yjbl? General changes in membrane permeability/fluidity can affect both antibiotic flux and outer membrane protein abundance. Showing that loss of yjbl does not affect general permeability with inclusion of detergent sensitivity, NPN assay, or some similar non-specific membrane stability assay would support that all observed effects are through proteome remodeling and not more general effects on membrane integrity.
2. The chip analysis expression of UzcR and ChvI from the xylose promoter. This would seem to disconnect them from normal expression control at their endogenous loci. Would the authors expect loss of yjbl to similarly effect UzcR and ChvI when expressed from the xyl promoter?
3. Significant comparison is made between stress responses of *E. coli* and *C. crescentus*. It would be helpful to also include discussion of the important differences in the OM lipid and protein composition of the bacteria, which would likely have important impacts on the conserved translation of yjbl effects to *E. coli* and other gram-negative bacteria.

Referee #3:

In this manuscript the authors provide insights into the function of secreted pentapeptide repeat proteins through the characterization of one such protein, Yjbl.

Overall, this is a very large body of work with meticulously planned experiments that prove the authors points. The introduction is exceptionally well written, really standing out in terms of richness of information and thorough review of the past literature in a manner that is accessible to broad scientific audiences. The same cannot be said for the results section; which is convoluted, unnecessarily complicated and written with a mission to convince the reviewer that the hypothesis stated by the authors is correct. Which is actually inappropriate.

The authors build upon previous findings with the currently presented work and in results section 1A-C they set up the premise very well. The authors provide the rationale of how their study led to the Identification of yjbl reasonably and provide the screens that led to the discovery of yjbl as a player. The data presented to identify ChvT as the conduit for VAN import into the cell is

solid. This is all good and the experimental design is robust for this part. The only minor comment here is to move the FDX data from the supplement to the main figure and move the alpha fold images to the supplement.

However, the rest of the data are presented in a convoluted fashion, with all the information presented at once with Figure 1D lumping info on *chvG* and *uzcS* all at once and not referring to *uzcS* or *R* until later in the document. This happens throughout and frankly it makes the work so inaccessible that it is majorly frustrating. Along the same lines, the authors make statements and then put in parentheses statements like "(see below)". This is most evident in the second data section with *chvR*. This section is very confusing as written and not easy to follow at all. "Please see below for TnSeq"... and so forth. As mentioned above, this gives the impression that the authors are trying to convince the reader that they are correct in a pre-assumed hypothesis rather than actually following their data. This again is evident with how the (uncontrolled) Northern blot is presented. No loading controls; no densitometry; no statistics; no mention of the fact that there is induction of *chvR* in the conditions that the authors claim there isn't.

Experimental issues include - 1) The description of how the authors ascribe regulation to the two-component systems. The addition of *P_{xyl}* promoter to drive the expression of the downstream RR for the data shown in Fig. 1D is not well explained. Have the authors grown their cells at what levels of xylose? And do they know whether disruption of *ChvG* silences the RR or does it actually over-activate it as has been reported for other two-component systems? This information is unclear and should be clarified. Additionally, there is no description of statistical analyses performed in figure 1D, or 2C, which is unacceptable. Northern blots have no loading controls. In 1D, you still see *chvR* transcript in the absence of *uzcS* and in the absence of *yjbl* and in both cases it is inducible by Zinc. Therefore, the notion that what they see is dependent on *ChvG* is flawed. Have the authors looked at the double *uzcS/chvG* mutant? I would not be surprised if *chvI* is phosphorylated by both HKs. Ditto on figure 2C. No controls. No description of statistical analyses. How about qPCR to measure steady-state transcript of both *chvR* and *acrA*?

The immunoblot in Figure 2C needs both loading controls AND densitometry. "Eyeballing" it is not sufficient.

There are pre-conceived notions on the function of the two TCSs where the authors assume that RRs bind the promoter only when phosphorylated. It would be good to have the phospho-inactive versions of these RRs to see what the authors get. It is also interesting that there are no *lacZ* profiles with the RR mutants and only the SK mutants. What do the authors see if they were to perform their *lacZ* analyses in the *uzcR* null or *star* backgrounds?

Figure S4B should be main figure.

Line 332 - How do the authors know and state that *Yjbl* is degraded?

Figure 4B Where are the loading controls? How does the reader know that this peptide is degraded, versus not being produced/translated? Where is the rest of the immunoblot?

The connection to TCS activation is circumstantial. The use of fusions is powerful, but I think it is critical to include the very relevant controls of double HK mutants or double RR mutants in their analyses. Moreover, how were the ChIP-seq analyses performed? Under what stress?? No mention of that. Again, this is partly due to how densely this is written. A re-organization of data would really be helpful here.

Minor - "Thus, modulation of *Yjbl* levels can affect Zn induction in both directions", means that xxx affects how Zn is induced. Which is not what the authors are trying to state. Please revise.

We thank the reviewers for their constructive comments, prompting us to refine the wording and flow in the results section. Importantly, we also bolstered the experimental evidence in our manuscript with the following additions:

1. Molecular modelling simulations revealing the asymmetric disposition of Yjbl in the LPS layer. This finding explains the differential thiol-reactivity of the two pairs of cysteines, one at each extremity of Yjbls, while revealing the role of the charged C-terminal tail in tethering Yjbl to the OM (Figure 6A-D). We also show that the N-terminal secretion signal is active when grafted onto metallo-beta-lactamase (lacking one) to confer doripenem resistance (Figure S6E). Lastly, we replaced the 3 terminal lysine residues with alanines to show that C-terminal tail of Yjbl is indeed important for function (Figure S6F).
2. We show that our newly isolated gain of function mutation (Q152R) in the response regulator UzcR is sufficient to induce firing at P_{chvR} (Figure S5G) and P_{bugA} (Figure 5D), even without Zn induction as determined by promoter probe assays and that, because of the concomitant synthesis of the BugA TBDR upon expression of UzcR-Q152R, cells are now sensitive to bacitracin. Inactivation of *bugA* in these cells mitigates the bacitracin sensitivity (Figure 5E, S6A).
3. We construct a new in-frame deletion in the *uzcR* to show that $\Delta uzcR$ and/or $\Delta chvI$ mutations, long with newly constructed $\Delta uzcR$ - $\Delta chvI$ double mutant block firing at the *chvR* promoter, even after Zn stress induction or upon inactivation of Yjbl (Figure S4D).
4. The antibiotic TBT1 (a first generation MsbA inhibitor) uncovered in our antibiotic library screen is also a potent inducer of the *acrAB-nodT* operon (Figure S5D).
5. Quantification of our *bugA::bugA-bla* and *chvT::chvT-bla* reporters using doripenem E(-strip)tests (Figure S5C), illustrating the differences in expression upon inactivation of Yjbl.
6. ChvT-hyperpore expression also renders (efflux minus) *E. coli* sensitive to fidaxomicin (Figure S2C).
7. Demonstration that *yjbl* mutants do NOT have a general permeability defect as shown by the fact that these mutants or MORE resistant to moenomycin and gamithromycin than *WT* cells (Figure S5E,F).
8. Permeability tests with NPN (not shown) and propidium iodine (PI, see rebuttal letter, page 4) showing no (major) permeability defect of *yjbl* cells versus *WT* cells.

Referee #1:

This manuscript presents several intertwined stories that complicate readability. It is difficult for the reader to follow a cohesive narrative, and I would recommend the authors streamline the content. Parts of the manuscript, for instance the section on ChvT and the hyperpore, could be omitted for clarity. Simplifying the presentation will help the reader focus on the key findings. That said, the experimental work is solid, and the genetics experiments are particularly well-executed. The wealth of information in this study is impressive, and the findings are both important and suitable for publication in a high-impact journal like EMBO. With some refinement in the presentation, this manuscript would make a strong contribution to the field.

The reviewer's points are well taken, and we have now tried to streamline and simplify the results section. The key findings are now more stressed in the discussion, at the expense of some of the technical details and specialized discussions.

With respect to the suggestion of omitting the ChvT and hyperpore section: we understand that reporting these studies, extends the manuscript, yet two important points compel us to

leave the section. First, ChvT and TBDRs along with their regulation are both the entry and end points of this study. And, importantly, the fact that two TBDRs confer sensitivity to vancomycin and bacitracin is unprecedented for gated TBDRs. The sufficiency experiments with the hyperpore present valuable evidence that the diameter of ChvT is indeed sufficiently wide to accommodate vancomycin, both in *Caulobacter* and in *E. coli*. Moreover, Reviewer 3 advised to move some of the TBDR data from the supplement into the main figures to make a stronger case for the TBDR section in new **Figure 1D**. We thank the Reviewers for her/his complement about the experiments and the potential of our work to make an important contribution to the field.

Specific Comments:

1. Cysteines near the Yjbl extremities: The flanking pairs of cysteines at the extremities of Yjbl are intriguing, particularly given that they remain in a reduced state despite being in the oxidizing environment of the periplasm. This raises important questions. How are these cysteines protected from oxidation? Addressing (or at least discussing) this question is relevant, especially because the authors use these cysteines to assess the surface exposure of the protein. Moreover, Yjbl's role in surface interactions between cells suggests potential surface exposure, yet this hypothesis was not thoroughly investigated. Also, why, in Figure 1A, is Yjbl depicted as oriented towards the periplasm? This contradicts the idea of surface exposure and should be clarified.

We are confident that the molecular modelling evidence can now provide answers to this important question. As can be seen in (new) **Figure 6A-C**, Yjbl is predicted to be embedded in the LPS with the N-terminal Cys pair buried within lipid A whereas the C-terminal Cys pair is in the environment of the O-antigen and more water exposed as quantified in **Figure 6D**. Moreover, the predicted membrane contacts for residues flanking the N-terminal Cys pair is higher than that of the C-terminal pair (**Figure 6B**). We interpret these atomistic simulations as strong supporting evidence that match our experimental results from the reactive thiol probing of surface-exposed Yjbl shown in (new) **Figure 6E**.

The modelling also reveals the C-terminal tail as a hook, reaching across the lipid A and PL layers (**Figure 6A**), possibly into the periplasm. Therefore, our revised cartoon in **Figure 1A** should reflect this asymmetric disposition of Yjbl.

Other (cell cycle-regulated) layers shown in Figure 1A (capsule>S-layer>O-antigen) may be a challenge for accessibility for direct interactions between cells via Yjbl attached to the rough LPS in *C. crescentus*. Potential cellular interactions controlled by Yjbl from the surface have not yet been investigated owing to possible interference with capsule/S-layer/O-antigen structures that may mask Yjbl.

2. Use of OM-impermeable probes: The authors use MGM to assess cysteine accessibility on the surface. However, the claim that MGM cannot cross the outer membrane (OM) needs further validation. It would be prudent to include control experiments using well-characterized periplasmic proteins to ensure that MGM is indeed impermeable to the OM under the experimental conditions.

Yes, we agree. We have toned down the original version of this statement, as it is not clear whether MGM really modifies Yjbl in the periplasm in EGTA-treated cells. At 5kDa, MGM should not be able to cross an intact OM. Note that the periplasm(ic protein content) of *Caulobacter* is poorly characterized, and we did not know of a suitable periplasmic reporter protein for Cys reactivity in the periplasm. Nonetheless, we attempted to react a modified version of the native metallo-beta-lactamase (Bla) of *C. crescentus* with MGM. The Bla version that we used was engineered to harbour extra cysteines at the C-terminus, but the

experiments were inconclusive because the Bla protein was lost during the washing steps. We suspect that unattached periplasmic content leaks out when the outer envelope layers are destabilized under EGTA treatment and cells are washed before MGM coupling. Because Yjbl-HA was not lost under these conditions, our experiments also indirectly imply that Yjbl is firmly associated to cells, which is consistent with the new MD simulations which suggest it to be LPS embedded. Considering the new MD simulations, it no longer matters whether MGM really enters the periplasm or not, because the differential thiol-reactivity with MGM can also be explained by the asymmetric disposition within the LPS, potentially protected by calcium ions that stabilize Lipid A in the LPS. It appears that chelation of these calcium ions with EGTA, may simply alter binding of Yjbl to lipid A, essentially de-protecting the other Cys pair, and enabling its reactivity towards MGM.

3. Clarification on Line 319: The authors mention that Zn induction is affected "in both directions." Could the authors clarify what is being induced? More detail is needed here to avoid ambiguity.

Fixed.

4. Clarification on Line 332: The sudden mention of potential Yjbl degradation is not well-introduced. The manuscript should provide more context on why degradation of Yjbl would occur and by what mechanisms. A more detailed introduction to this idea earlier in the manuscript would help the reader follow the logic.

Since Zn stress and inactivation of *yjbl* both trigger UzcSR signalling, it follows that Zn might induce the removal of Yjbl within two hours of Zn exposure. Therefore, loss of Yjbl by proteolysis is a likely possibility and we hope that this logic come out now clearly in the revised manuscript.

5. Line 286-289: Fig2C, the authors mention a decrease of 76% (for *Uzcs/yjbl*) and 91% (for *chvg/yjbl*) compared to WT. I find this confusing, as the actual decrease appears minor, such as 9% for the *chvg/yjbl* mutant, for instance. Additionally, these differences are on comparisons between the mock WT condition and Zinc addition for the mutants; is this comparable? Finally, are these differences significant?

For Figure 2C the reference state for the LacZ activity in our comparisons between strains is the activity of the reporter in untreated WT cells. The Reviewer is presumably referring to the activity in the *yjbl* mutant (so 224% activity) and when these *yjbl* cells are devoid of UzcS or ChvG, then LacZ activity drops to 82% and 85%, i.e. a drop of 142% and 139%.

The other option would be to display the absolute activities in Miller units, instead of expressing it as relative percentage. However, day to day variation in Miller units makes this representation not ideal. To normalize for this day-to-day variation, the entire WT and mutant series is measured on one day and then the duplicate/triplicate measurements are done on another day(s).

In all our LacZ-based promoter probe assays, statistical significance calculated arithmetically is often validated even among strains exhibiting less than 20% difference. However, from > 20 years of experience, we know (empirically) that a > 20% difference in LacZ activity between strains/conditions (say for example +/- Zn for WT in Figure 2C) is required to translate into biologically meaningful significance or phenotypic traits that are robust for quantification or experimental follow-up by complementary approaches such as transcriptomics, Northern blots, immunoblotting, EOPs etc. It is not clear to us whether this is an idiosyncrasy of LacZ assays in *Caulobacter*, but after many years of LacZ assays this correlation is clearly there. In sum, we routinely only follow-up on activity differences of at least > 20%. If we show arithmetic significance to our graphs, even 10% changes will show

significance, and the reader may wonder why we did not follow up on those. We think that such phenotypes are too subtle for phenotypic follow-up.

It should also be noted that for these LacZ promoter probe assays are strongly promoter dependent, presumably of the dynamic activity range inherent to each promoter test case. Take for example P_{acrA} , that we know is (directly) repressed by TipR and (directly) activated by UzcR in response to Zn stress in *WT* cells (Figure 2C). Zn induction leads to only a 29% increase in LacZ measurements, but of course a strong induction is masked (dampened) by repression of TipR. If repression of TipR is eliminated, then the induction is larger. The dynamic range of induction say for UzcR-dependent promoters by Zn is clearly wider for promoters such as P_{bugA} for which there is no known repression. And finally, we point out that all promoter probe assays are done in triplicate.

6. Line 444: Why BugA is no longer detected in the ChvG/yjbI mutant? Does chvG/chvI affect bugA transcription? How?

In the previous version of the manuscript we showed that induction of P_{bugA} can occur by administering Zn or by inactivation of YjbI and in both cases it is strongly dependent on UzcS, but not ChvG (Figure 5B). These findings are consistent with the previous UzcSR-dependent transcriptomic analyses in response to Zn from Park et al (2017, 2019). Additionally, we now show in Figure S5E that expression of mutant UzcR-Q152R (but not *WT* UzcR) results in the following

- i) sufficiency of inducing P_{bugA} in the absence of Zn (Figure 5D);
 - ii) sufficiency of rendering cells susceptible to Bacitracin (2.5 $\mu\text{g}/\text{mL}$, Figure S6A);
- and
- iii) requirement of BugA for bacitracin sensitivity (Figure S6A).

Thus, UzcR can cause bacitracin sensitivity through induction of BugA, independently of Zn stress.

Minor comments/typos: thanks for noticing and taking the time to highlight typos, that we hope to have fixed now.

Line 409: Is Novobiocin an LptB activator? Yes, recently it was shown that novobiocin also enhances LptB according to PMID: 29746111 and 29135241, in addition to its role in trajecting GyrB

Line 126: regulate coordinate...

Line 128: « that » twice

Line 192: « Garm-negative" instead of Gram-negative

Line 212: why is figure 3 mentioned before figure 2?

Line 106: Fig 1a is missing an arrow for inducing DjlA by UzcR, correct? yes

Line 317: "Prom" instead of from

Line 333: "and" mentioned twice

Line 345: missing a verb

Line 423: remove "above"

Line 424: 'decrease" in italic?

Line 426: Lack of a space "... $\mu\text{g}/\text{mL}$ of"....

Line 430: BAC2 or BAC3 ?

Line 496: ope and/or a bottleneck in folding/in

Line 500: we noted

Line 539: may represent a common

Line 542: that are import substrates of

Line 543: be likened to a narrow-spectrum

Line 549: The current challenge is also understanding TBDR expression is regulated and enlarging our repertoire of how chemical

Referee #2:

Costafrolaz et al investigate the role of protein YbjI in *C. crescentus* antibiotic sensitivity. Through extensive genetic analysis they provide evidence of pathways through ybjI that influence two envelope stress responses that affect outer membrane protein composition and antibiotic sensitivity. The data largely supports the authors' conclusions. A few clarifications would help interpretation of the data.

1. Is general membrane integrity influenced by deletion of ybjI? General changes in membrane permeability/fluidity can affect both antibiotic flux and outer membrane protein abundance. Showing that loss of ybjI does not affect general permeability with inclusion of detergent sensitivity, NPN assay, or some similar non-specific membrane stability assay would support that all observed effects are through proteome remodeling and not more general effects on membrane integrity.

Our antibiotic library screen comparing the resistance profiles of *WT* versus *ybjI* mutant cells (Figure 4G), not only revealed which antibiotics that the *ybjI* mutant is hypersensitive to, but also which ones are less effective against *ybjI* cells versus *WT* cells. There is only a brief reference to this latter group of antibiotics in the text, which includes several macrolides (large molecules that can be excluded by an intact OM) such as solithromycin, but also others such as moenomycin for example. We have also now included data to verify the effect of moenomycin (Figure S5E). In other (unpublished) work, we have isolated a *C. crescentus* mutant with a general permeability defect towards large antibiotics akin to that of *E. coli imp/lptD4213*, *lptA* or *bamB* mutant cells. While this new *C. crescentus* mutant is sensitive to moenomycin, fidaxomicin and pleuromutilin(s), the fact that the *ybjI* mutant does not share this sensitivity profile, indicates that *ybjI* cells do NOT suffer from a general permeability defect.

Independently of the differential susceptibility argument, we have performed the requested NPN assays. It turns out that NPN is not suitable for *Caulobacter* the way it is used for *E. coli*, for reasons that are unclear. Perhaps NPN is retained at certain *Caulobacter* envelope structures (capsule, S-layer, O-antigen) or it is a potent efflux substrate of one (or several) of the at least four efflux pumps encoded in *Caulobacter*. However, in looking for alternative stains, we think that propidium iodine (PI) is suitable as a permeability probe for *Caulobacter* (20µM PI, measurements done in a Synergy H1 Biotech microplate reader two hours after the addition of PI, monitored by excitation at 535 nm and emission at 617 nm). In this case, we see no difference between *WT* and *ybjI* cells. In control experiments, we destabilized the LPS of *WT* cells with polymyxin, the envelope with EGTA or dissipated the PMF (and efflux) with CCCP. All these treatments resulted in increased PI staining of cells. These PI measurements suggest that there is no major permeability defect of *ybjI* cells.

Figure for reviewers removed

2. The chip analysis expression of UzcR and ChvI from the xylose promoter. This would seem to disconnect them from normal expression control at their endogenous loci. Would the authors expect loss of yjbl to similarly effect UzcR and ChvI when expressed from the xyl promoter?

Correct: the strains used for ChIP-Seq are *uzcR* or *chvI* are merodiploids, harbouring *xyl::P_{xyl}-HA-uzcR* and *xyl::P_{xyl}-HA-chvI* in *WT/Δbla* backgrounds that essentially uncouple expression of the HA tagged variants independent of the regulatory circuit. The comparison in the ChIP-Seq is between these cells and the derivatives harbouring the *yjbl::Tn(Gm^R)* mutation, therefore the ChIP-Seq data (Figure 3) indeed shows the consequences of loss of Yjbl when expression of HA-UzcR and HA-ChvI are expressed from P_{xyl}. In these *yjbl::Tn* cells, the HA-tagged proteins are expressed from P_{xyl} and therefore their synthesis is maintained, regardless of whether Yjbl is present or not.

3. Significant comparison is made between stress responses of *E. coli* and *C. crescentus*. It would be helpful to also include discussion of the important differences in the OM lipid and protein composition of the bacteria, which would likely have important impacts on the conserved translation of yjbl effects to *E. coli* and other gram-negative bacteria.

We have reduced the comparison/discussion of the *Caulobacter* and *E. coli* Rcs systems in the revised version in order to have more space available to highlight other general points. Currently, we feel it would be a stretch to suggest any relationship between *yjbl* function and LPS/OM composition in different genera, simply because we have insufficient knowledge about the role of Yjbl in other bacteria. Moreover, for *Caulobacter* the LPS/lipid composition is still poorly characterized, including the exact molecular nature of its lipid A (lack of phosphates, instead featuring galacturonic acid at these positions in *Caulobacter* instead of phosphates, PMID: 18387917) and its biosynthetic pathway, the O-antigen and its synthesis, but many alpha-proteobacteria including *Caulobacter* have other glycolipids such as ceramides/sphingolipids in the OM (PMID: 40533060, 35649364). We prefer to avoid biasing our discussions on the molecular nature lipid A by misleading/misinterpreting speculations without considering other OM components such as the ceramides in the model.

Referee #3:

In this manuscript the authors provide insights into the function of secreted pentapeptide repeat proteins through the characterization of one such protein, YjbI.

Overall, this is a very large body of work with meticulously planned experiments that prove the authors points. The introduction is exceptionally well written, really standing out in terms of richness of information and thorough review of the past literature in a manner that is accessible to broad scientific audiences. The same cannot be said for the results section; which is convoluted, unnecessarily complicated and written with a mission to convince the reviewer that the hypothesis stated by the authors is correct. Which is actually inappropriate. The authors build upon previous findings with the currently presented work and in results section 1A-C they set up the premise very well. The authors provide the rationale of how their study led to the Identification of yjbI reasonably and provide the screens that led to the discovery of yjbI as a player. The data presented to identify ChvT as the conduit for VAN import into the cell is solid. This is all good and the experimental design is robust for this part. The only minor comment here is to move the FDX data from the supplement to the main figure and move the alpha fold images to the supplement.

We agree and now show the FDX data moved to **Figure 1D**. With respect to the Alphafold views shown in **Figure 1B**, we prefer to maintain them for visualization of the Asp/Glu linings and the two Cys patches on the predicted surface of the quadrilateral helix. It also helps with understanding the MD simulations presented in Figure 6.

However, the rest of the data are presented in a convoluted fashion, with all the information presented at once with Figure 1D lumping info on chvG and uzcS all at once and not referring to uzcS or R until later in the document. This happens throughout and frankly it makes the work so inaccessible that it is majorly frustrating.

We understand that the comprehensive the LacZ bar graphs in mutants are very cumbersome for reviewers/readers, having to return to the data sets presented in earlier panels. Yet, data duplications in subsequent panels are also not ideal. It may be possible to address this issue with the extended view (EV) feature of the new format of the EMBO Journal in the published version after acceptance of the manuscript.

Along the same lines, the authors make statements and then put in parentheses statements like "(see below)". This is most evident in the second data section with chvR. This section is very confusing as written and not easy to follow at all. "Please see below for TnSeq"... and so forth.

We hope that the revised flow of the Results section now minimizes these inconveniences/confusions.

As mentioned above, this gives the impression that the authors are trying to convince the reader that they are correct in a pre-assumed hypothesis rather than actually following their data. This again is evident with how the (uncontrolled) Northern blot is presented. No loading controls; no densitometry; no statistics; no mention of the fact that there is induction of chvR in the conditions that the authors claim there isn't.

* This assessment is INCORRECT: the control for the Northern Blot had already been presented in the supplement of the original submission and in case this information was not

accessible to (or missed by) the reviewer, **Figure S3B** still shows the complete Northern, loading control (probing with 5S rRNA) and transcript quantification of the detected ChvR levels....

Experimental issues include - 1) The description of how the authors ascribe regulation to the two-component systems. The addition of P_{xyl} promoter to drive the expression of the downstream RR for the data shown in Fig. 1D is not well explained. Have the authors grown their cells at what levels of xylose? And do they know whether disruption of ChvG silences the RR or does it actually over-activate it as has been reported for other two-component systems? This information is unclear and should be clarified.

Our *chvG::P_{xyl}* insertional mutation disrupts the coding sequence of ChvG in the middle of *chvG* and places a P_{xyl} promoter downstream. Cells harbouring this *chvG* mutation were grown without xylose and this is now clearly stated in the legends. Moreover, we have included promoter-probe-based LacZ measurements with cells harbouring an in-frame deletion in *chvI* (Δ *chvI*) to consolidate the results with our *chvG* insertional mutant (**Figure S4D**). Note that the placement of a promoter to sustain downstream expression with our *chvG::P_{xyl}* mutant is akin to other insertional mutants such as transposon mutants. Transposons often carry outward facing promoters in the inverted repeats. In fact, this is also the case for the mariner (*himar1*-based) transposon Mar2xT7 (PMID: 16477005) that we use regularly in our studies (including this one) and that has a promoter reading outward that can, if appropriately placed, to maintain expression of essential distal open coding sequences (see for example PMID: 24415923 where we reported a Tn-Mar2x7 insertion in the essential promoter of *ftsN* which drives constant expression of FtsN in cells lacking the transcriptional activator of the FtsN promoter). But we also know from use of Tn- mar2xTz by Tn seq that it can insert into an operon that would otherwise not tolerate Tn insertions in the upstream cistron. This was the rationale for designing *chvG::P_{xyl}* because we had shown that the genes downstream of *chvG* can affect growth/fitness (see PMID: 30165530).

As *ChvI* is encoded upstream of *ChvG*, there is no interference on *ChvI* synthesis. Our *chvG::P_{xyl}* insertional mutation inactivates *ChvG* as indicated by the loss of P_{*chvR*}-lacZ activity when *ChvG* is disrupted in *yjbl::Tn* cells (**Figure 1C**). The same is true for the Δ *chvI* mutation (**Figure S4D**). Clearly, a *ChvG*-independent, but *UzcS*-dependent, path of activating P_{*chvR*} in response to Zn stress remains to be explored in future work.

Additionally, there is no description of statistical analyses performed in figure 1D, or 2C, which is unacceptable. Northern blots have no loading controls. In 1D, you still see *chvR* transcript in the absence of *uzcS* and in the absence of *yjbl* and in both cases it is inducible by Zinc. Therefore, the notion that what they see is dependent on *ChvG* is flawed. Have the authors looked at the double *uzcS/chvG* mutant? I would not be surprised if *chvI* is phosphorylated by both HKs.

* **INCORRECT** - despite the Reviewer's assertion, these data had been shown in the original submission reported in Figure 1D (now **Figure 1E**) reporting info on the Δ *uzcS*- Δ *chvG::P_{xyl}* double and the Δ *uzcS*- Δ *chvG::P_{xyl}*- *yjbl::Tn* triple mutant that seemed to have escaped the Reviewer's attention. The Northern blots previously had controls associated, shown in the supplement whereas the Inset in **Figure 1E** (previously Figure 1D) just showed the relevant band/signal of *ChvR* of the same Northern Blot. The reference to the complete Northern data had previously been detailed in the results section and in the Figure Legend of Figure 1 This critique is thus inappropriate and it is returned to the Reviewer.

There are absolutely no claims in our study that Zn induction is *chvG* dependent, only the overexpression of the genes of interest in absence of *yjbl*. Additionally, the zinc dependent

chvR expression in $\Delta uzcS$ cells is 133% of the *WT* activity, in comparison to 720% activity in the presence of *uzcSR*. This residual zinc induction of *chvR* is probably due to other regulators or enzymatic stresses and cannot less the principal role of the UzcRS TCS for zinc induction. It is worth to mention that this residual Zn induction has been observed in previous studies (PMID: 28035693, 30536755).

Ditto on figure 2C. No controls. No description of statistical analyses. How about qPCR to measure steady-state transcript of both *chvR* and *acrA*?

The immunoblot in Figure 2C needs both loading controls AND densitometry. "Eyeballing" it is not sufficient.

The blots in Figure 2D have now been quantified as requested. They reveal the relative increase in abundance of AcrA in *yjb1::Tn* mutant cells and the "reversal" of this effect by the *uzcR** mutations.

There are pre-conceived notions on the function of the two TCSs where the authors assume that RRs bind the promoter only when phosphorylated. It would be good to have the phospho-inactive versions of these RRs to see what the authors get. It is also interesting that there are no *lacZ* profiles with the RR mutants and only the SK mutants. What do the authors see if they were to perform their *lacZ* analyses in the *uzcR* null or star backgrounds?

Figure S4B should be main figure.

We now include the analyses in newly constructed $\Delta uzcR$ single and double mutants to the necessity. Conversely, and importantly, we use the gain-of function UzcR-Q152R mutation to show sufficiency of signal induction, resembling the effects of UzcSR signaling triggered by the *yjb1* mutation or by exposure to Zn.

Figure S4C is the NAL and Zn cumulative induction of P_{acrA} and P_{djlA} , is used for minor claims in our study. Thus, for more clarity and less dense figures, we prefer to keep it in the supplementary results section.

Line 332 - How do the authors know and state that Yjbl is degraded?

Figure 4B Where are the loading controls? How does the reader know that this peptide is degraded, versus not being produced/translated? Where is the rest of the immunoblot?

The connection to TCS activation is circumstantial. The use of fusions is powerful, but I think it is critical to include the very relevant controls of double HK mutants or double RR mutants in their analyses.

* **INCORRECT**- Double HK mutants had previously been shown already (in the original submission). The double RR mutant has now been added as request with newly constructed (unmarked) $\Delta uzcR$ mutation in *WT* and $\Delta chvI$ cells to measure LacZ activities, provided in Figure S4D. The $\Delta uzcR$ had to be re-made because of marker incompatibilities with the promoter probe plasmid.

The tagged Yjbl-proteins are expressed from P_{xyI} , and cells were grown iPYE with xylose, for constitutive expression of Yjbi-HA, independently of Yjbl, UzcR or ChvI (as for the strains used in the ChIP-Seq which also included derivatives with *yjb1::Tn* (GmR) insertion. Additionally, it is mentioned in the Experimental Procedures that the cells have been grown for 4h under xylose induction but only exposed to Zn or other stresses for 2h. The untreated control demonstrate that the protein is produced.

The blot concerning the MGM is shown is from the 14 to 70 kDa range, the rest of the membrane has not been imaged (cut blot) as the protein shift is not relevant as visible in the range shown. If the reviewer #3 is referring to the raw image, they will be deposited in the raw data folder.

Moreover, how were the ChIP-seq analyses performed? Under what stress?? No mention of that. Again, this is partly due to how densely this is written. A re-organization of data would really be helpful here.

As mentioned in the Experimental Procedures, for ChIP-Seq the cells were grown without stress, just in PYE with xylose for merodiploids with or without the *yjb1::Tn* (GmR) mutation. The legend now states that PYE containing xylose was used.

Minor - "Thus, modulation of YjbI levels can affect Zn induction in both directions", means that xxx affects how Zn is induced. Which is not what the authors are trying to state. Please revise.

This statement has been removed.

Dear Patrick,

Thank you for submitting a revised version of your manuscript. It has now been seen by all original reviewers, who are now generally satisfied with the revision and recommend acceptance of the manuscript. As requested by reviewer #3, and also to adhere to our editorial policies, please include the appropriate controls for Northern blots also in the main figures.

There now remain only a few editorial points that need to be addressed before I can extend official acceptance of the manuscript:

1. Please ensure that the manuscript section order is as follows: Title page - Abstract - Keywords - Introduction - Results - Discussion - Methods - Data Availability - Acknowledgements - Disclosure and Competing Interests Statement - References - Figure Legends - Table(s) - Expanded View Figure Legends.
2. Please check if the email provided for the co-author Simon Vallet (simon.vallet@unige.ch) is correct; emails sent to this address were returned to the sender.
3. In Author Checklist, please fill in column E for row 43 (availability of new materials and reagents).
4. Please rename the supplementary figures "Figure EV1" - EV5 and upload them as individual, high resolution figure files.
5. Please rename the supplementary tables into Dataset EV1 - EV6, remove their legends from the manuscript text and add them to each corresponding file in a separate tab/worksheet.
6. Please rename the movie files into Movie EV1-EV2 and zip each file with the corresponding legend. Please remove the legends from the manuscript text.
7. Please add a "Disclosure and competing interests statement" section (further info: <https://www.embopress.org/page/journal/14602075/authorguide#conflictsofinterest>).
8. Please check the order of the figure callouts, as they should be in numerical order: currently, Figure 3 is called out before Figure 2A, while Fig EV4D is called out before Fig EV3B.
9. Figure panel 5E is not mentioned in the manuscript text; please add the corresponding callout.
10. We require a Data Availability Section at the end of Materials and Methods. Please include a resolvable link to the transcriptomics dataset GSE269833 in this section. More information about the format of this section can be found here: <https://www.embopress.org/page/journal/14602075/authorguide#dataavailability>.
11. We noted that source data are missing for several main figure panels. Please fill in the source data checklist provided by our data editor Hannah Sonntag on October 2nd 2024 and submit with the next version. In case some source data cannot be provided, this should be indicated in the checklist with a brief reason provided.
12. During our standard image integrity check, we noted reuse of image sections between figure panels 2A, 2E, 5A and 55E (PYE column). If this was intentional, please indicate the reuse in the figure legends.
13. Our data editors have flagged the following issues in figure legends that need correcting:
 - Please define the error bars in the legends of figures 1E, 2C, 5B, C, D; S4 A-D; S5 D, G.
14. Papers published in The EMBO Journal are accompanied online by a 'Synopsis' to enhance discoverability of the manuscript. It consists of a short (1-2 sentences) summary of the findings and their significance and 3-4 bullet points highlighting key results. Please send us your synopsis text with the next revision.
15. Please note that it is The EMBO Journal policy for the transcript of the editorial process (containing referee reports and your response letters) to be published as an online supplement to each paper. If you should prefer removal of any referee-only figures included in the point-by-point response(s), e.g. because they may still be used for future publication or because they have been reproduced from published work by others, please do let us know immediately via response email.
More information is available here: https://www.embopress.org/transparent-process#Review_Process

With best wishes,

leva

leva Gailite, PhD
Senior Scientific Editor
The EMBO Journal
Meyerhofstrasse 1
D-69117 Heidelberg
Tel: +4962218891309

i.gailite@embojournal.org

We realize that it is difficult to revise to a specific deadline. In the interest of protecting the conceptual advance provided by the work, we recommend a revision within 3 months (11th Feb 2026). Please discuss the revision progress ahead of this time with the editor if you require more time to complete the revisions. Use the link below to submit your revision:

Referee #1:

The authors have addressed all of my previous concerns with care. The revised version is improved in both clarity and experimental depth; the new data strengthen the mechanistic model. I am satisfied with the revision and recommend the manuscript for publication.

Referee #2:

My comments have been addressed. No further comments.

Referee #3:

The revisions provided satisfy the reviewer's comments.

It is still preferred that all controls - and not just specific bands - are shown in the Northern blots.

The authors addressed the remaining editorial issues.

Dear Patrick,

Thank you for submitting the final revised version and addressing the remaining editorial points. I am now pleased to inform you that your manuscript has been accepted for publication in The EMBO Journal.

Before we forward your manuscript to our publishers, we would like to propose some edits in the manuscript title, abstract and synopsis, with the main goal of increasing its accessibility to our more general readership - please see the attached file. We have also prepared a short blurb that will accompany the title of your manuscript in our online system. Please take a look and let me know if any corrections or adjustments are needed.

Once we have received your approval of the textual edits, your manuscript will be processed for publication by EMBO Press. It will be copy edited and you will receive page proofs prior to publication. Please note that you will be contacted by Springer Nature Author Services to complete licensing and payment information.

You may qualify for financial assistance for your publication charges - either via a Springer Nature fully open access agreement or an EMBO initiative. Check your eligibility: <https://link.springer.com/journal/44318/how-to-publish-with-us>

If you have any questions, please do not hesitate to contact the Editorial Office. Thank you for your contribution to The EMBO Journal and congratulations on a nice study!

With best wishes,

leva

leva Gailite, PhD
Senior Scientific Editor
The EMBO Journal
Meyerhofstrasse 1
D-69117 Heidelberg
Tel: +4962218891309
i.gailite@embojournal.org

Please note that it is The EMBO Journal policy for the transcript of the editorial process (containing referee reports and your response letters) to be published as an online supplement to each paper. If you should prefer removal of any referee-only figures included in the point-by-point response(s), e.g. because they may still be used for future publication or because they have been reproduced from published work by others, please do let us know immediately via response email. More information is available here: <https://link.springer.com/partners/embo-press/editorial-policies#Peer%20review>
